# Extracellular fluid viscosity enhances cell migration and cancer dissemination

Kaustav Bera[1,2], Alexander Kiepas[1,2,14], Inês Godet[1,3,14], Yizeng Li[4,14], Pranav Mehta[1,2,14], Brent Ifemembi[1,2], Colin D. Paul[5], Anindya Sen[1,2], Selma A. Serra[6], Konstantin Stoletov[7], Jiaxiang Tao[8], Gabriel Shatkin[9], Se Jong Lee[1,2], Yuqi Zhang[1,2], Adrianna Boen[1], Panagiotis Mistriotis[10], Daniele M. Gilkes[1,3,11], John D. Lewis[7], Chen-Ming Fan[8], Andrew P. Feinberg[3,9,12], Miguel A. Valverde[6], Sean X. Sun[1,2,9,13] & Konstantinos Konstantopoulos[1,2,3,9 ✉]

Cells respond to physical stimuli, such as stiffness[1], fluid shear stress[2] and hydraulic pressure[3,4]. Extracellular fluid viscosity is a key physical cue that varies under physiological and pathological conditions, such as cancer[5]. However, its influence on cancer biology and the mechanism by which cells sense and respond to changes in viscosity are unknown. Here we demonstrate that elevated viscosity counterintuitively increases the motility of various cell types on two-dimensional surfaces and in confinement, and increases cell dissemination from three-dimensional tumour spheroids. Increased mechanical loading imposed by elevated viscosity induces an actin-related protein 2/3 (ARP2/3)-complex-dependent dense actin network, which enhances Na$^+$/H$^+$ exchanger 1 (NHE1) polarization through its actin-binding partner ezrin. NHE1 promotes cell swelling and increased membrane tension, which, in turn, activates transient receptor potential cation vanilloid 4 (TRPV4) and mediates calcium influx, leading to increased RHOA-dependent cell contractility. The coordinated action of actin remodelling/dynamics, NHE1-mediated swelling and RHOA-based contractility facilitates enhanced motility at elevated viscosities. Breast cancer cells pre-exposed to elevated viscosity acquire TRPV4-dependent mechanical memory through transcriptional control of the Hippo pathway, leading to increased migration in zebrafish, extravasation in chick embryos and lung colonization in mice. Cumulatively, extracellular viscosity is a physical cue that regulates both short- and long-term cellular processes with pathophysiological relevance to cancer biology.

Cell migration is essential for a variety of pathophysiological processes, such as development, tissue homeostasis, immune surveillance and cancer metastasis. Although mechanical forces arising from cell–substrate interactions and the surrounding fluid have been shown to regulate cell migration behaviour[6–8], the effect of physiologically relevant extracellular viscosities on cell function remains unclear. To date, most in vitro cell functional assays, including motility, are performed in medium with a viscosity close to that of water (0.7 centipoise (cP) at 37 °C). However, the viscosity of the interstitial fluid varies up to 3.5 cP (ref. [9]) and can be further augmented by macromolecules, such as mucins, secreted not only by resident epithelial cells in various tissues but also by tumour cells[10]. Primary tumour growth may compress lymphatic vessels and compromise drainage[11], leading to the accumulation of macromolecules over time. Elevated

degradation of extracellular matrix at tumour sites also exacerbates macromolecular crowding[12], which can further increase the viscosity of the interstitial fluid[13]. Notably, the physiological viscosity of whole blood varies between 4 and 6 cP, and can exceed 8 cP during pathological abnormalities[14].

Previous research has shown that supraphysiological viscosities (≥40 cP) increase the motility of carcinoma and normal cells on two-dimensional (2D) surfaces[15,16]. This is rather counterintuitive, as viscosity slows down the motion of particles within fluids. Nevertheless, key fundamental and translational questions remain unanswered, including how cells sense the physical cue of elevated, yet physiologically relevant, extracellular viscosity; whether elevated viscosity alters the cell phenotype and the underlying mechanisms of cell locomotion; how the cytoskeleton cooperates with ion channels and

[1]Department of Chemical and Biomolecular Engineering, Johns Hopkins University, Baltimore, MD, USA. [2]Institute for NanoBioTechnology, Johns Hopkins University, Baltimore, MD, USA. [3]Department of Oncology, The Sidney Kimmel Comprehensive Cancer Center, Johns Hopkins University School of Medicine, Baltimore, MD, USA. [4]Department of Biomedical Engineering, Binghamton University, SUNY, Binghamton, NY, USA. [5]Laboratory of Cell Biology, Center for Cancer Research, National Cancer Institute, National Institutes of Health, Bethesda, MD, USA. [6]Laboratory of Molecular Physiology, Department of Medicine and Life Sciences, Universitat Pompeu Fabra, Barcelona, Spain. [7]Department of Oncology, University of Alberta, Edmonton, Alberta, Canada. [8]Department of Embryology, Carnegie Institution for Science, Baltimore, MD, USA. [9]Department of Biomedical Engineering, Johns Hopkins University, Baltimore, MD, USA. [10]Department of Chemical Engineering, Auburn University, Auburn, AL, USA. [11]Cellular and Molecular Medicine Program, Johns Hopkins University School of Medicine, Baltimore, MD, USA. [12]Center for Epigenetics, Johns Hopkins University School of Medicine, Baltimore, MD, USA. [13]Department of Mechanical Engineering, Johns Hopkins University, Baltimore, MD, USA. [14]These authors contributed equally: Alex Kiepas, Inês Godet, Yizeng Li, Pranav Mehta. ✉e-mail: konstant@jhu.edu

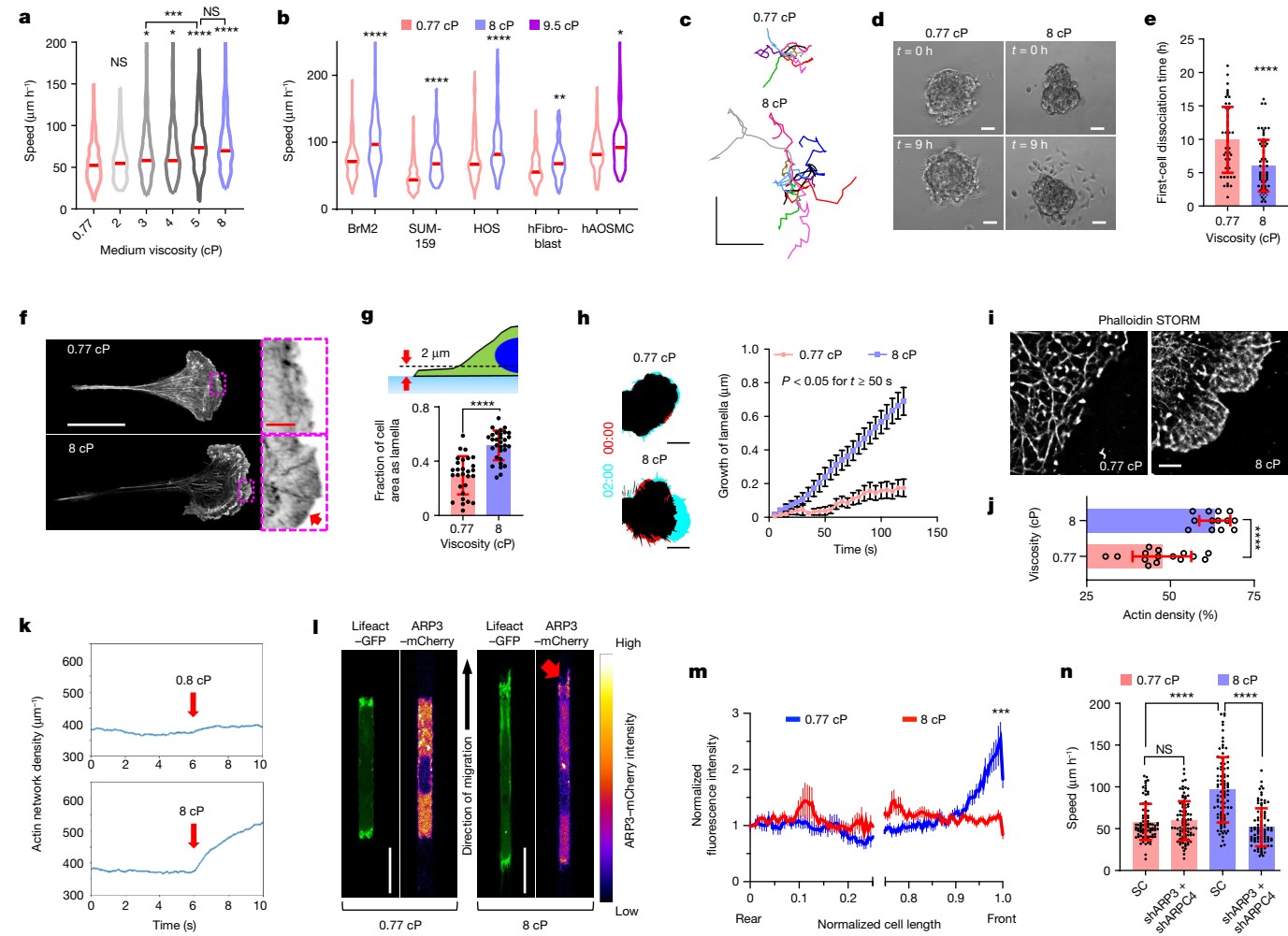

**Fig. 1 | Viscosity enhances cell migration and promotes an ARP2/3-mediated dense actin network at the leading edge. a,b,** Speeds of MDA-MB-231 cells (**a**) and other indicated cell types (**b**) inside confining channels at prescribed viscosities. The red lines represent the median of ≥69 cells from ≥3 experiments. **c,** Cell trajectories on 2D collagen-coated surfaces after 10 h. **d,** Cells disseminating from 3D spheroids. **e,** The time required for the first cell dissociation from each spheroid (n ≥ 53) from 3 experiments. **f,** Airyscan images of phalloidin stained cells on collagen-coated substrates. The red arrow indicates high F-actin staining along the cell edge. **g,** The fraction of cell-projected area with a Lifeact–GFP-rich lamella for n ≥ 28 cells from 3 experiments. **h,** The leading edge of Lifeact–GFP-expressing cells on collagen-coated surfaces at t = 0 min (red) and t = 2 min (cyan) (left). Right, leading-edge lamella growth in n ≥ 19 cells from 3 experiments. Data are the moving average ± s.e.m. P < 0.05 for all points t ≥ 50 s. Time is shown as min:s. **i,j,** STORM reconstruction (**i**) and density quantification (**j**) of F-actin for cells (n ≥ 13) on substrates from 2 experiments. **k,** The average actin density over time from 20 stochastic simulations. Viscous forces were applied at t = 6 s (red

arrow) and maintained until the end of the simulation. **l,** Confocal images of cells expressing Lifeact–GFP and ARP3–mCherry in confinement. The red arrow indicates high ARP3 intensity at leading-edge protrusions at 8 cP. **m,** The relative ARP3–mCherry intensity along normalized cell length in confined cells. Data are the moving average ± s.e.m. for n = 21 cells from 4 experiments. ***P < 0.001 for all comparisons at normalized cell length > 0.96. The x axis is discontinued between 0.25 and 0.75 to highlight differences at the cell edges. **n,** Confined migration speeds of SC versus *ARP3/ARPC4* double-knockdown cells (n = 90) from 3 experiments. For **e, g, j** and **n**, data are mean ± s.d. Unless otherwise indicated, statistical comparison was performed with respect to 0.77 cP. Statistical analysis was performed using Kruskal–Wallis tests followed by Dunn's test (**a** and **n**), Mann–Whitney U-tests (BrM2 only) or unpaired t-tests after log-transformation (other cells) (**b**), unpaired t-tests (**e, g** and **j**) and two-way analysis of variance (ANOVA) followed by Šidák's test (**h** and **m**). Scale bars, 250 μm (**c**), 50 μm (**d**), 25 μm (**f**, white), 3 μm (**f**, red), 10 μm (**h**), 2 μm (**i**), 20 μm (**l**). The cell model was MDA-MB-231 unless otherwise indicated. *P < 0.05, **P < 0.01, ***P < 0.001, ****P < 0.0001.

transporters to mediate efficient migration at elevated viscosities; whether faster motility observed in vitro translates to the in vivo setting and, if so, whether cell exposure to elevated viscosity affects cancer metastasis.

## Fluid viscosity enhances cell migration

To investigate the effects of increased extracellular fluid viscosity on cell function in vitro, we incorporated the amounts of 65 kDa methylcellulose into cell culture medium necessary to obtain media with viscosities ranging from 0.77 cP (0%) to 8 cP (0.6%) at 37 °C without

appreciably altering the osmolarity of the medium (Extended Data Fig. 1a,b). Using MDA-MB-231 breast cancer cells as a model, we found that the migration speed inside polydimethylsiloxane (PDMS)-based confining (width × height = 3.5 × 10 μm²) channels[17] (Extended Data Fig. 1c) increased with increasing extracellular viscosity, reaching a peak at 5–8 cP (Fig. 1a). Media with an elevated (8 cP) viscosity using alternative biologically inert macromolecules, such as dextran (500 kDa)[2] and polyvinylpyrrolidone K-90, also supported faster motility (Extended Data Fig. 1d–f), whereas low-molecular-mass dextran (6 kDa), used at similar molarity (~1.95 μM) to 500 kDa dextran, did not potentiate confined migration (Extended Data Fig. 1g). These data reveal that elevated

extracellular fluid viscosity enhances MDA-MB-231 cell migration in confinement independent of the nature of macromolecules. Enhanced migration speeds at elevated viscosities were also observed using different tumour cells (MDA-MB-231-BrM2 brain metastatic cells derived from breast cancer (hereafter BrM2)[18], SUM159 breast carcinoma[19] and human osteosarcoma (HOS)) and non-cancerous cells (normal human fibroblasts and human aortic smooth muscle cells (hAOSMCs)) (Fig. 1b).

To extend the relevance of our findings, we showed that elevated viscosity also enhanced random cell locomotion on 2D collagen-I-coated substrates (Fig. 1c and Extended Data Fig. 1h), and reduced the time required for 2D wound closure (Extended Data Fig. 1i) without altering cell proliferation relative to baseline viscosity (Extended Data Fig. 1j,k). Moreover, elevated viscosity accelerated the dissociation of MDA-MB-231 cells from three-dimensional (3D) breast cancer spheroids (Fig. 1d,e and Supplementary Video 1) and reduced the cell entry time into confining channels (Extended Data Fig. 1l). Cumulatively, our data identify extracellular fluid viscosity as a physical cue that regulates motility in a variety of physiologically relevant 2D and 3D settings.

## Viscous forces induce actin remodelling

Faster motility in confining channels is typically characterized by a cell phenotypic switch from amoeboid (or blebbing) to protrusive (or mesenchymal)[17]. Blebbing motile cells[17,20] display a pill-like morphology with membrane blebs at the cell poles (Extended Data Fig. 2a), and represent 80% of confined MDA-MB-231 and SUM159 cells at baseline viscosity (0.77 cP; Extended Data Fig. 2b,c). Elevated viscosity (8 cP) switches cells to a protrusive (mesenchymal) phenotype inside confining channels (Extended Data Fig. 2a–c). Breast cancer cells can migrate in confining channels at 0.77 cP albeit at a reduced velocity[21] when actin polymerization is abolished by a high dose (2 µM) of latrunculin A (LatA) (Extended Data Fig. 2d,e). By contrast, and consistent with the cell's mesenchymal mode of migration, LatA halts motility in confinement at 8 cP (Extended Data Fig. 2d,e), suggesting that viscosity mediates the mechanical loading of actin and its subsequent reorganization. Indeed, cells plated on 2D surfaces at 8 cP exhibited large F-actin-rich lamellipodia (Fig. 1f) and a markedly higher projected area as lamella relative to cells at 0.77 cP (Fig. 1g). Cells at 8 cP also displayed faster growth of leading-edge lamella (Fig. 1h, Extended Data Fig. 2f and Supplementary Video 2) and a lower occurrence of edge-retraction events compared with at 0.77 cP (Extended Data Fig. 2g,h). We also assessed actin dynamics by analysing the uninterrupted actin front growth and its retrograde flow in cells at 0.77 cP and 8 cP using β-actin fused to photoactivatable GFP (PA-GFP) and mRFP (Extended Data Fig. 2i and Supplementary Video 3). The uninterrupted actin front grew slower against higher viscosity (Extended Data Fig. 2j), consistent with the acquired kymographs (Extended Data Fig. 2h), in which the sharp spikes at 0.77 cP showed a rapid, but non-persistent, outward burst of the cell edge. Together with the slower actin retrograde flow at 8 cP (Extended Data Fig. 2k), these data suggest a denser actin structure at the cell front. This was confirmed using stochastic optical reconstruction microscopy (STORM) analysis of the actin cytoskeleton, which shows that cells at 8 cP exhibit a dense, highly branched network within a growing edge (Fig. 1i,j).

To verify our experimental findings that elevated extracellular viscosity promotes mechanical loading on a growing actin network, we adapted a stochastic model[22] that tracks lamellipodial network growth at the resolution of individual actin filaments[23]. We refined this model by incorporating the effects of extracellular fluid dynamics and viscous loading forces to quantitatively assess actin architecture (see the 'Theoretical methods' section in the Methods; Extended Data Fig. 3a). Without altering any parameter other than viscosity, we successfully recapitulated a denser actin network at 8 cP compared with at 0.8 cP (Fig. 1k and Extended Data Fig. 3a,b). These modelling data further verify that elevated extracellular viscosity is sufficient to alter the actin network at the leading edge of cells. The model also predicted a decrease in the growth speed of the actin network (Extended Data Fig. 3c–e), similar to experimental observations (Extended Data Fig. 2k).

Although the model predicted an overall increase in F-actin density (Fig. 1k and Extended Data Fig. 3a,b,f), more detailed evaluation revealed a spike in pointed ends compared with capped barbed ends immediately after the application of elevated viscous force (Extended Data Fig. 3g). This finding is indicative of an increase in the branched actin network as new actin filaments form with their pointed ends attached to mother filaments. Consistent with the established role of ARP2/3 in nucleating branched actin[24,25], cells at 8 cP exhibited a more intense ARP3 signal at their leading edge and, specifically, at the tips of protrusive filaments, as opposed to the relatively uniform distribution detected at baseline viscosity in cells on 2D surfaces and inside confinement (Fig. 1l,m and Extended Data Fig. 4a). Accordingly, ARP2/3 knockdown using short hairpin RNA (shRNA) targeting *ARP3* and/or *ARPC4* or the use of the ARP2/3-specific inhibitor CK666 suppressed confined migration at elevated, but not baseline, viscosities (Fig. 1n and Extended Data Fig. 4b–d). Importantly, ARP2/3 depletion reversed the protrusive morphology of confined cells at 8 cP to bleb-based (Extended Data Fig. 4e). In accord with the established crosstalk between ARP2/3 and focal adhesions[17], increased numbers and sizes of focal adhesions were detected at 8 cP (Extended Data Fig. 4f–h), and were primarily distributed at the front of the cell (Extended Data Fig. 4i–k), consistent with the localization pattern of ARP3 (Fig. 1l,m). Cumulatively, we demonstrate that ARP2/3-mediated actin branching occurs as a direct cell response to increased viscosity. We next investigated the pathway by which cells sense and adapt to elevated fluid viscosity.

## Viscosity induces NHE1-dependent cell swelling

In confinement, tumour cells migrate by taking up and discharging water at the cell poles[21], and also by pushing the column of water ahead of them, which generates hydraulic resistance[3,17]. Elevated extracellular viscosity increases the hydraulic resistance at the leading edge of confined cells. The two-phase model of cell migration—which treats the cytosol, which is essentially water and the actin network, as two fluid phases interacting with each other—predicts that high hydraulic resistance promotes cell migration through water uptake[26]. Thus, we examined whether hydraulic resistance due to elevated fluid viscosity increases water uptake, which may result in increased cell volume. Using confocal imaging[20], we demonstrate that elevated viscosity (8 cP) increased MDA-MB-231 cell volume by around 40% on unconfined 2D surfaces and inside confining channels (Fig. 2a). Given the well-established roles of NHE1 in regulatory volume increase[27] and confined migration[21], we examined its polarization pattern, activity levels and functional contribution to motility at different viscosities. Consistent with previous research[21], NHE1 was preferentially localized to the leading edge of confined cells at 0.77 cP (Fig. 2b,c). Interestingly, both NHE1 polarization at the cell anterior and NHE1 activity, monitored using the ratiometric pH sensor pHRed[28], were markedly increased at 8 cP (Fig. 2b–d). NHE1 depletion (Extended Data Fig. 5a–c) abrogated the viscosity-induced volume increase of cells on 2D surfaces (Extended Data Fig. 5d) and in confinement (Fig. 2e). To model the contributions of actin-based and NHE1-mediated (osmotic engine model) motility, we extended the two-phase actin–cytosol model by incorporating the distinct distribution patterns of focal adhesions and NHE1 detected at elevated and baseline viscosities (see the 'Theoretical methods' section in the Methods; Extended Data Fig. 5e). The model successfully recapitulated the higher polarization of F-actin at the cell anterior at 8 cP compared with at 0.77 cP (Extended Data Fig. 5f). Moreover, model predictions revealed that abrogation of NHE1-mediated ion flux exerted a higher inhibitory effect on the migration speed at 8 cP relative to at 0.77 cP, which was verified experimentally using shNHE1 (Fig. 2f). The critical role of NHE1 in confined migration was verified using two

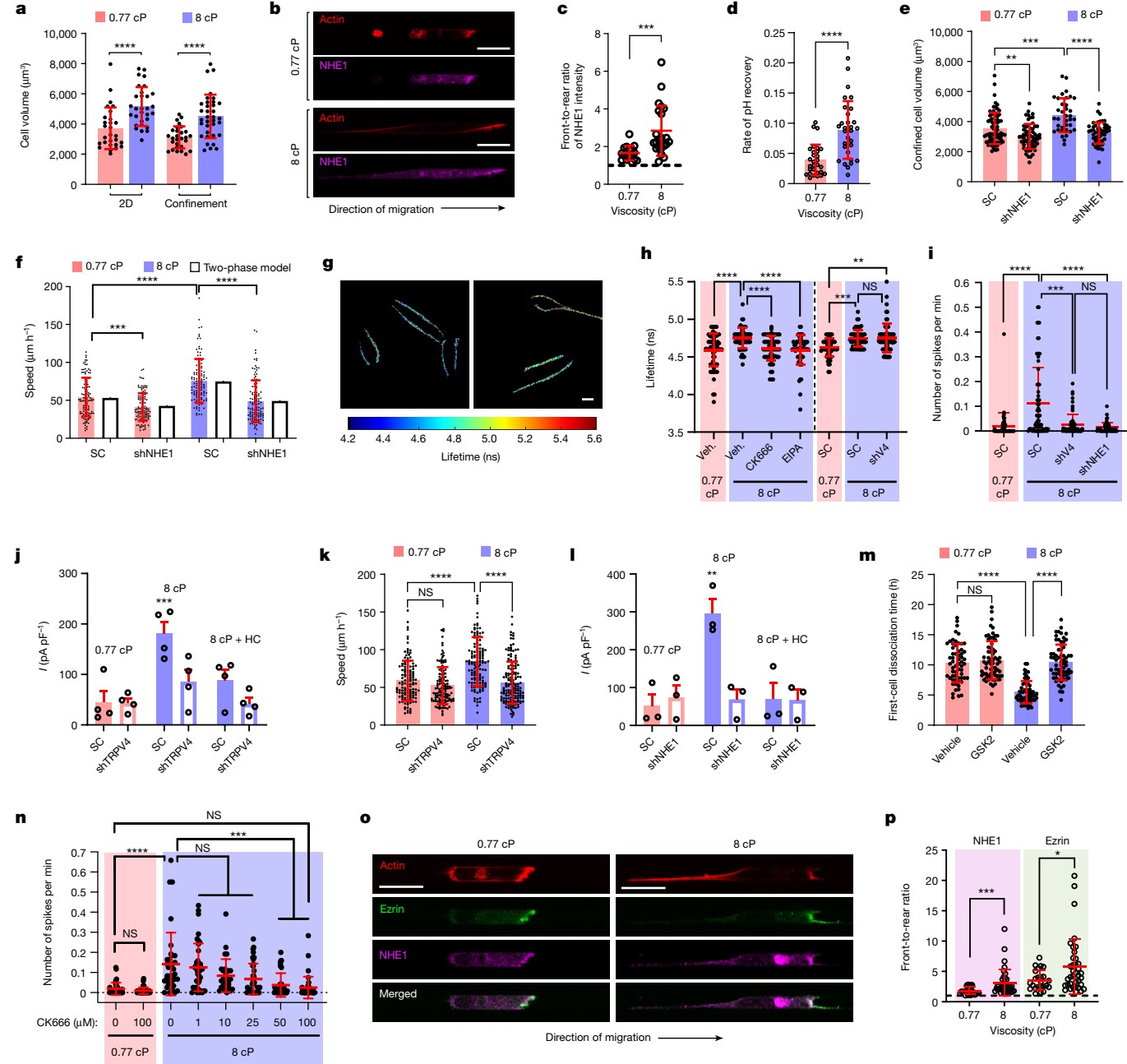

**Fig. 2 | Viscosity promotes NHE1-dependent cell swelling, which activates TRPV4 leading to calcium influx. a**, The volume of Lifeact–GFP-labelled MDA-MB-231 cells ($n \geq 28$) at the indicated viscosities from 3 experiments. **b**, Confocal images of confined MDA-MB-231 cells stained for NHE1 and phalloidin. **c**, Front-to-rear NHE1 intensity ratio in NHE1-immunostained cells ($n \geq 16$) from 3 experiments. **d**, The rate of pH recovery in pHRed-expressing MDA-MB-231 cells ($n \geq 29$) from 5 experiments. **e**, The volume of SC and shNHE1 Lifeact–GFP-tagged MDA-MB-231 cells ($n \geq 36$) from 3 experiments. **f**, Confined migration speeds of SC and shNHE1 MDA-MB-231 cells ($n \geq 113$) from 3 experiments compared with the two-phase model predictions. **g**, The elevated Flipper-TR lifetimes in MDA-MB-231 cells on 2D surfaces indicate high membrane tension. **h**, The membrane tension in wild-type MDA-MB-231 cells after treatment with vehicle (veh.), CK666 or EIPA, and in SC or shTRPV4 cells ($n \geq 58$) from 3 experiments. **i**, The number of calcium flashes in SC, shTRPV4 or shNHE1 MDA-MB-231 cells ($n \geq 52$) on 2D surfaces from 3 experiments. **j,l**, TRPV4 currents

(**l**) in SC- and shTRPV4 (**j**) or shNHE1 (**l**) MDA-MB-231 cells ($n \geq 3$) with or without the TRPV4 inhibitor HC-067047 from ≥3 experiments. **k**, Confined migration speeds of SC and shTRPV4 MDA-MB-231 cells ($n \geq 129$) from 3 experiments. **m**, The time required for the first cell dissociation from each spheroid ($n \geq 57$) after treatment with vehicle control or the TRPV4 inhibitor GSK 2193874 (GSK2) from 3 experiments. **n**, The number of calcium flashes in MDA-MB-231 cells ($n \geq 29$) treated with the ARP2/3 inhibitor CK666 from 2 experiments. **o**, Confocal images of confined MDA-MB-231 cells stained for NHE1, ezrin and phalloidin. **p**, Front-to-rear NHE1 or ezrin intensity ratio from immunostained cells ($n \geq 24$) from 2 experiments. Data are mean ± s.d. (**a**, **c**–**f**, **h**, **i**, **k**, **m**, **n** and **p**) and mean ± s.e.m. (**j** and **l**). Statistical analysis was performed using unpaired $t$-tests after log-transformation (**a**, **c** and **d**), one-way ANOVA followed by Tukey's test after log-transformation (**e** and **m**), Kruskal–Wallis tests followed by Dunn's test (**f**, **h**, **i**, **k** and **n**), one-way ANOVA followed by Holm–Šidák's test (**j** and **l**) and Mann–Whitney $U$-tests (**p**). Scale bars, 20 μm (**b** and **o**) and 10 μm (**g**).

distinct shNHE1 sequences (Extended Data Fig. 5g) and pharmacological inhibition in different cell lines (Extended Data Fig. 5h). Previous research showed that NHE1 works in coordination with aquaporin 5

(AQP5) to mediate water uptake in confinement[21]. Accordingly, AQP5 depletion was equally effective in reducing confined migration as *NHE1* knockdown (Extended Data Fig. 5i).

## Interplay of NHE1, membrane tension and TRPV4

When cells are exposed to mechanical perturbations, force balance at the cell surface shows that cell membrane tension changes, which can lead to membrane channel activation[7], thereby enabling cells to sense external cues and respond by mediating calcium influx through mechano/osmo-sensitive ion channels (MOSICs). Consistent with this notion, cells that were subjected to elevated viscosity (8 cP) displayed a marked increase in membrane tension (Fig. 2g,h), as observed for cells exposed to hypotonic solutions (Extended Data Fig. 5j), and in the number of calcium spikes (Fig. 2i and Extended Data Fig. 5k). Moreover, both cell impermeable (BAPTA) and permeable (BAPTA AM) calcium chelating agents abolished the viscosity-induced enhancement of cell migration (Extended Data Fig. 5l,m), thereby suggesting the potential involvement of MOSICs in this process. Possible $Ca^{2+}$-permeable MOSICs expressed in MDA-MB-231 cells[3] include PIEZO1 and PIEZO2, TRPM7 and TRPV4. Knockdown of *PIEZO1* or *PIEZO2* did not reduce the enhancement of cell motility at elevated viscosity (Extended Data Fig. 5n,o). CRISPR–Cas9-based knockout of *TRPM7*, a key mechanosensor of hydraulic pressure[3,17] and shear stress[2], also did not block the enhanced migration at 8 cP (Extended Data Fig. 5p). Similarly, 2-aminoethoxydiphenyl borate (2-APB), which blocks TRPC1, TRPC6 and TRPM7, and restricts the release of store-operated $Ca^{2+}$, did not exert an inhibitory effect on cell motility (Extended Data Fig. 5q). Consistent with the effect of fluid viscosity on ciliated epithelial cells[29,30], cells exposed to 8 cP displayed increased whole-cell TRPV4 current and calcium oscillations that were abolished by shTRPV4 or treatment with the TRPV4 inhibitor HC067047 (Fig. 2i,j and Extended Data Fig. 6a–c). Although our intracellular calcium concentration measurements did not provide precise localization of the calcium signal, TRPV4-generated calcium influx has been reported to act either locally[31] or away from where it is generated[32]. In accordance with the calcium-dependent effect of viscosity on confined cell migration (Extended Data Fig. 5l,m), *TRPV4* knockdown was sufficient to abrogate the viscosity-induced enhancement of cell motility in confinement (Fig. 2k), thereby illustrating that TRPV4 is the key MOSIC activated by elevated extracellular viscosity. The critical role of TRPV4 was verified using a second target sequence for shTRPV4 (Extended Data Fig. 6a,d) and the specific pharmacological inhibitor GSK2193874 (Extended Data Fig. 6e). To generalize our findings, we show that inhibition or depletion of TRPV4 in SUM159, HOS, U87 and BrM2 cells reduced confined migration at 8 cP to the levels at baseline viscosity (Extended Data Fig. 6f). Taken together, elevated viscosity promotes NHE1-dependent cell swelling, which leads to increased membrane tension that triggers calcium influx through the MOSIC TRPV4. In line with the proposed model, NHE1 inhibition abrogated viscosity-mediated increases in membrane tension (Fig. 2h), whole-cell TRPV4 currents (Fig. 2l and Extended Data Fig. 6g) and calcium spikes (Fig. 2i and Extended Data Fig. 6c). By contrast, reducing the expression of β1-integrin, which has been proposed to participate in the mechanical activation of TRPV4[33], did not reduce calcium spikes at 8 cP (Extended Data Fig. 6h).

To further establish that TRPV4 activation is downstream of NHE1-mediated cell swelling, we first demonstrated that the synthetic TRPV4 agonist GSK1016790A (1 and 15 μM) was sufficient to restore cell migration speeds in NHE1-depleted cells at elevated viscosity (Extended Data Fig. 7a,b). Moreover, TRPV4 activation induced calcium flashes at baseline viscosity that were not blocked by shNHE1 (Extended Data Fig. 7c). Concurrent inhibition of NHE1 and TRPV4 functions was equally effective as individual interventions in suppressing cell migration (Extended Data Fig. 7d) and cell volume (Extended Data Fig. 7e) at elevated viscosity, thereby implying that they function in the same pathway. Importantly, shTRPV4 did not suppress the NHE1-dependent increases in cell volume (Extended Data Fig. 7e) and membrane tension (Fig. 2h) at 8 cP. As an orthogonal approach, we increased cell volume at 0.77 cP using 2× diluted (141 mOsm) hypotonic medium (Extended Data

Fig. 7f). Hypotonic shock was sufficient to induce calcium flashes in wild-type and scramble control (SC) cells (Extended Data Fig. 7g,h), and these calcium flashes were abolished by *TRPV4* knockdown (Extended Data Fig. 7h). These findings highlight the critical role of TRPV4 in mediating calcium flux in response to NHE1-mediated cell volume and membrane tension increases induced by elevated viscosity. To extend the physiological relevance of our findings, we demonstrate that TRPV4 inhibition by GSK2193874 (15 μM) markedly delayed cell dissociation from MDA-MB-231 spheroids at 8 cP without having an effect at 0.77 cP (Fig. 2m).

We next investigated whether intrinsic load adaptation of actin due to elevated viscosity is responsible for NHE1 localization, NHE1-dependent cell volume and membrane tension increases, and TRPV4-mediated calcium signalling. ARP2/3 inhibition blocked viscosity-induced increases in cell volume, membrane tension and calcium activity (Fig. 2h,n and Extended Data Fig. 7i), whereas shTRPV4 did not alter the highly branched actin network within a growing cell edge at 8 cP (Extended Data Fig. 7j). We subsequently focused on ezrin—an actin-binding protein that anchors NHE1 to the cell membrane[34]. Consistent with previous research on 2D surfaces[34], immunofluorescence analysis of confined cells co-stained with ezrin and NHE1 revealed significant colocalization at all viscosities (Fig. 2o). Interestingly, elevated relative to baseline viscosity induced higher polarization of both NHE1 and ezrin at the cell leading edge (Fig. 2o,p). Ezrin inhibition at 8 cP with NSC668394 (NSC6) blocked calcium flashes (Extended Data Fig. 7k) and phenocopied the confined migration of shNHE1 cells (Extended Data Fig. 7l). Importantly, no additive inhibitory effect on confined cell migration was observed in shNHE1 cells that were treated with ezrin inhibitor (Extended Data Fig. 7m). Taken together, elevated viscosity imposes mechanical loading, which triggers the formation of a denser actin network at the cell leading edge at which ezrin enrichment promotes NHE1 polarization and NHE1-dependent cell swelling. This leads to increased membrane tension that, in turn, activates TRPV4 and regulates calcium influx.

## TRPV4 enhances cell contractility

Given the association of TRPV4 with the regulation of the RHOA–ROCK–myosin II pathway[35,36], we examined the spatial localization of RHOA activity and its potential involvement in cell motility at elevated viscosities. Using a Förster resonance energy transfer (FRET)-based RHOA-activity biosensor[2,20], we found that cells on collagen-I-coated 2D surfaces exhibited higher RHOA activity at 8 cP compared with at 0.77 cP (Fig. 3a,b and Extended Data Fig. 8a–c). Notably, markedly higher RHOA activity levels were detected at the leading-edge lamella of 2D cells at 8 cP (Fig. 3a,b and Extended Data Fig. 8d). The viscosity-induced increase in RHOA was regulated by TRPV4, as TRPV4 depletion reduced RHOA activity to baseline levels (Fig. 3c). This is further substantiated by data showing that knockdown of *NHE1*, which encodes the upstream activator of TRPV4, or calcium chelation through BAPTA abolished the increase in RHOA activity at 8 cP (Extended Data Fig. 8e,f).

In confinement, cells at 0.77 cP displayed intense RHOA activity at their poles (Extended Data Fig. 8g,h), consistent with their blebbing phenotype (Extended Data Fig. 2a–c). Notably, cells at 8 cP, which predominantly displayed a protrusive morphology, exhibited elevated RHOA activity primarily at their leading edge (Extended Data Fig. 8g,h). We verified these findings using a live reporter of RHOA activity that contains GFP fused to the GTP–RHOA-binding anillin homology domain (GFP–AHD)[37] (Fig. 3d–f and Supplementary Video 4). Consistent with NHE1 enrichment at the cell anterior at 8 cP (Fig. 2b,c), inhibition of NHE1 by EIPA[21] (Fig. 3e) or its downstream effector TRPV4 by GSK2193874 (Fig. 3f) reverts the spatial localization of active RHOA at 8 cP to that at 0.77 cP where it is distributed evenly at both poles of the cell. The RHOA–ROCK axis activates myosin-II-based contractility, which generates high regional intracellular pressure[17] that can enable cells to overcome the

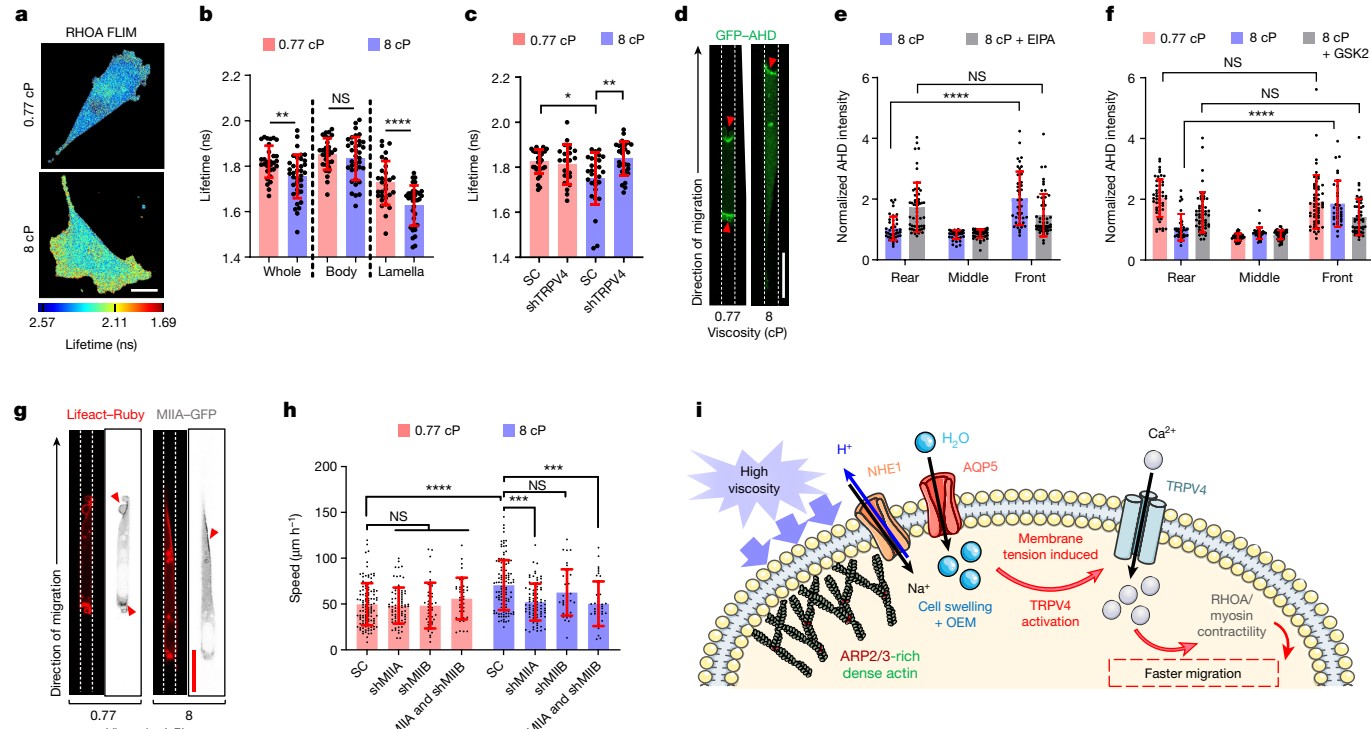

**Fig. 3 | TRPV4-mediated activation of RHOA–ROCK–myosin II contractility.**
**a**, The lifetimes of the RHOA activity biosensor in MDA-MB-231 cells on a 2D surface at the indicated viscosities. **b**, The subcellular distribution of RHOA activity in $n \geq 30$ cells on a 2D surface from 4 experiments. **c**, RHOA activity in SC and shTRPV4 MDA-MB-231 cells ($n \geq 21$) on a 2D surface from 3 experiments. **d**, Confocal images of GFP–AHD-expressing MDA-MB-231 cells in confinement. The red arrowheads indicate regions of active RHOA. **e**, GFP–AHD intensity in different segments of confined MDA-MB-231 cells ($n \geq 33$) at 8 cP in the presence of vehicle control or NHE1 inhibitor from $\geq 3$ experiments. **f**, GFP–AHD intensity in different segments of confined MDA-MB-231 cells ($n \geq 33$) after treatment with vehicle control or the TRPV4 inhibitor GSK 2193874 (GSK2) from

3 experiments. The intensity in each segment was normalized to the mean intensity of the entire cell in **e** and **f**. **g**, Confocal images of MIIA–GFP-expressing and Lifeact–Ruby-expressing MDA-MB-231 cells migrating in confinement. The red arrowheads indicate regions of intense MIIA localization. **h**, The confined migration speeds of SC and *MIIA* and *MIIB* single- or double-knockdown MDA-MB-231 cells ($n \geq 38$) from 2 experiments. Data are mean ± s.d. **i**, Schematic of the proposed viscosity-sensing pathway. OEM, osmotic engine model. The schematic in **i** was created using Servier Medical Art. Statistical analysis was performed using unpaired *t*-tests (**b**), Kruskal–Wallis tests followed by Dunn's test (**c** and **h**) and two-way ANOVA followed by Tukey's test (**e** and **f**). Scale bars, 20 μm (**a**, **d** and **g**).

elevated resistance during migration at 8 cP. Live-cell imaging of myosin IIA (MIIA)–GFP (Fig. 3g) and immunostaining for phosphorylation (Ser19) of myosin light chain 2 (pMLC) (Extended Data Fig. 8i,j) showed that myosin II activity is elevated at the leading edge of confined cells migrating at 8 cP. Moreover, Y27632 (10 μM), which inhibits ROCK, or blebbistatin (50 μM), which inhibits myosin ATPase activity, abrogated the viscosity-induced enhancement of confined cell motility (Extended Data Fig. 8k). *MIIA*, but not *MIIB*, knockdown[3] reduced motility at 8 cP to basal levels (Fig. 3h), thereby identifying MIIA as the primary effector of actomyosin contractility at 8 cP. The expression level of *MIIC* was negligible in MDA-MB-231 cells (Extended Data Fig. 8l), and the role of MIIC in cell motility was therefore not tested.

To investigate whether the heightened RHOA activity and actin reorganization at the leading edge of cells at 8 cP coordinate to enhance cell motility, we tested how inhibition of the RHOA–ROCK–myosin II axis alters leading-edge lamella growth and retrograde actin flow. Double knockdown of *MIIA* and *MIIB* or ROCK inhibition by Y27632 did not affect leading-edge lamella growth at either elevated or basal viscosities relative to the appropriate controls (Extended Data Fig. 8m,n). Live-cell imaging of MIIA–GFP revealed the presence of intense myosin II puncta in the lamella of cells at 8 cP, consistent with their locally increased RHOA activity, which is in distinct contrast to the rather uniform myosin IIA signal distribution at 0.77 cP (Extended Data Fig. 8o). These myosin II puncta, which are indicative of higher actomyosin coupling, disappeared after cell treatment with the ROCK inhibitor Y27632 (Extended Data Fig. 8p). Importantly, an analysis of actin dynamics showed that

Y27632 or shMIIA reduced retrograde actin flow by 50% (Extended Data Fig. 8q,r). This level of reduction is consistent with previously reported data[38] and indicates that there are reduced myosin-II-mediated pulling forces on the leading-edge actin network. The TRPV4 inhibitor GSK2193874 had similar effects to Y27632 (Extended Data Fig. 8q), thereby providing further support for the role of TRPV4 in upregulating RHOA-dependent actomyosin contractility at 8 cP. Consistent with data on leading-edge lamella growth (Extended Data Fig. 8m,n), TRPV4 or ROCK inhibition or shMIIA had no appreciable effect on the rate of uninterrupted protrusion growth of cells at 8 cP (Extended Data Fig. 8s,t). As a control, inhibition of ARP2/3 function suppressed both retrograde actin flow and uninterrupted protrusion growth, consistent with the role of ARP2/3 in nucleating branched actin networks (Extended Data Fig. 8q,s).

It is well established that classic blebbing (amoeboid) versus protrusive (mesenchymal) phenotypes have divergent requirements of actomyosin contractility[20,39]. Elevated contractility typically leads to the collapse of the actomyosin cytoskeleton into cellular blebs[39]. However, here we observed that cells display a predominantly protrusive phenotype at 8 cP (Extended Data Fig. 2a–c), yet they exhibit elevated RHOA–ROCK–myosin II activity at the cell leading edge. This counterintuitive phenomenon is explained by the elevated number of focal adhesion sites at 8 cP (Extended Data Fig. 4f–k), which can anchor stress fibres and promote protrusions at the cell leading edge. Indeed, depletion of β1-integrin (Extended Data Fig. 9a), which is known to reduce focal adhesions[40], supported a blebbing phenotype irrespective

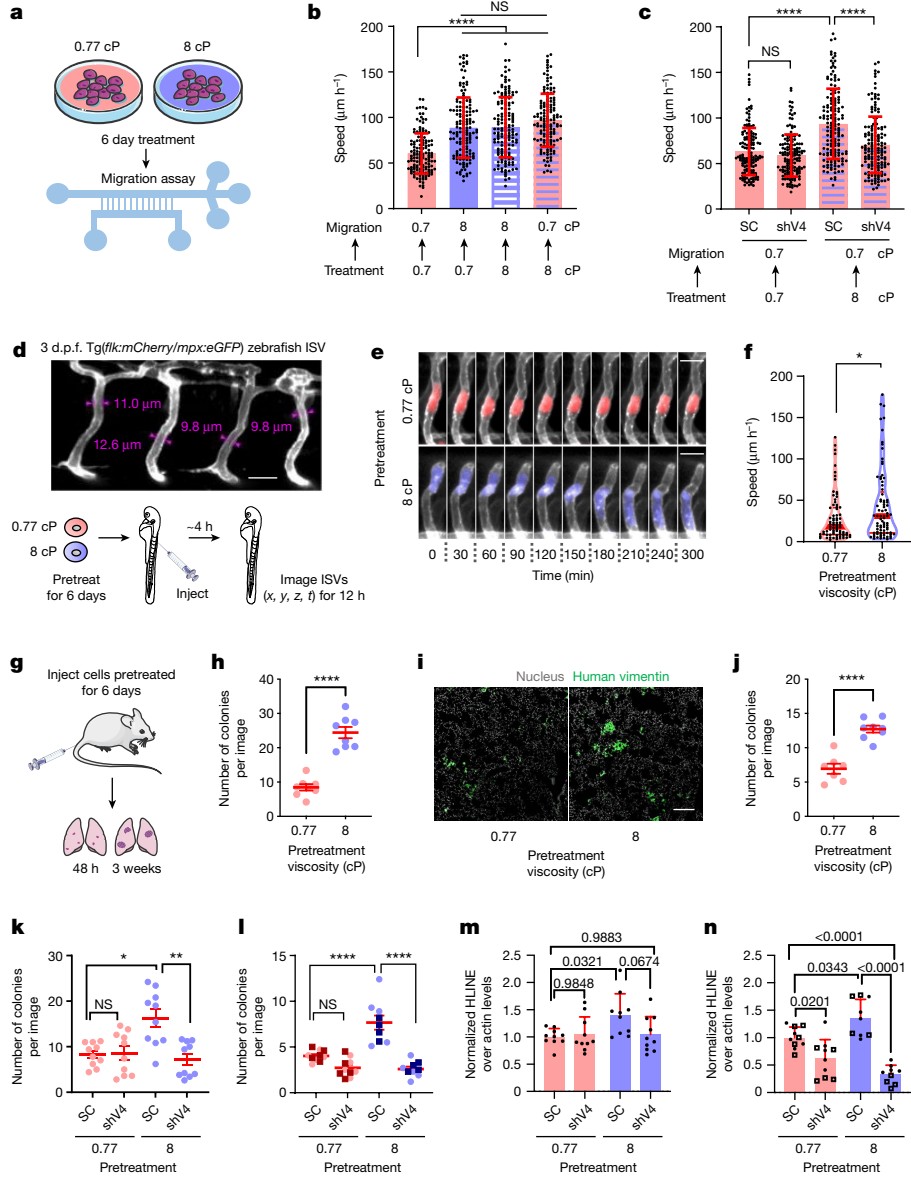

**Fig. 4 | MDA-MB-231 cells preconditioned to elevated viscosity exhibit enhanced migration, extravasation and lung colonization. a**, Illustration of cell preconditioning at the indicated viscosities. **b**, Confined migration speeds of preconditioned cells (0.77 or 8 cP for 6 days) resuspended at the indicated migration viscosity. Data are mean ± s.d. for $n ≥ 140$ cells from 3 experiments. **c**, The confined migration speeds of preconditioned SC or shTRPV4 cells allowed to migrate at 0.77 cP. Data are mean ± s.d. for $n ≥ 146$ cells from 3 experiments. **d**, Confocal image of 3 day post-fertilization (d.p.f.) zebrafish ISVs with measurements of vessel width (top). Bottom, experimental design of migration studies in zebrafish. **e,f**, Time-lapse confocal images (**e**) and average speeds (**f**) of preconditioned cells ($n ≥ 77$) inside ISVs from 3 experiments. The red lines indicate the median (thick) and quartiles (thin). **g**, The experimental design of mouse tail-vein experiments. **h**, The number of human vimentin-positive colonies in the lungs 48 h after injection. Data are mean ± s.e.m. for 8 mice per group from 2 experiments. **i,j**, Confocal images of lung sections (**i**) and quantification of human vimentin-positive metastatic colonies (**j**) 3 weeks after injection. Data are mean ± s.e.m. for a total of ≥7 mice per group from 2 experiments. **k,l**, The number of human vimentin-positive metastatic colonies in the lungs 48 h (**k**) and 3 weeks (**l**) after injection. Data are mean ± s.e.m. for ≥9 mice per group from 2 experiments. The squares represent experiments with PVP as the medium additive. **m,n**, qPCR detection of human DNA in the lungs of mice 48 h (**m**) or 3 weeks (**n**) after injection. Data are mean ± s.d. for ≥9 mice per group. Squares are from experiments with PVP. Statistical analysis was performed using Kruskal–Wallis tests followed by Dunn's test (**b** and **c**), Mann–Whitney $U$-tests (**f**), unpaired $t$-tests (**h** and **j**), one-way ANOVA followed by Tukey's test (**l**–**n**) and one-way ANOVA followed by Tukey's test on log-transformed data (**k**). Scale bars, 20 μm (**d**), 30 μm (**e**) and 200 μm (**i**). The schematics in **a**, **d** and **g** were created using Servier Medical Art.

of the extracellular viscosity (Extended Data Fig. 9b) and greatly suppressed confined migration at 8 cP (Extended Data Fig. 9c). By contrast, β1-integrin knockdown did not alter confined migration at baseline viscosity in agreement with previous research[41]. These data are consistent with the dual role of MIIA in promoting contractility and actin disassembly[42] and highlight that MIIA-dependent contractility is a means to promote not only bleb-based migration but also mesenchymal migration.

Although cells can readily migrate in confinement at 0.77 cP even after LatA-mediated F-actin disruption[21], LatA halts motility inside confining channels at 8 cP without altering NHE1 or ezrin front-to-rear polarization relative to the vehicle control or NHE1 activity (Extended Data Fig. 9d–f). This finding suggests that the osmotic engine alone is not sufficient to overcome the elevated hydraulic resistance and support motility at 8 cP. This is corroborated by data showing that overexpression of NHE1–GFP could partially rescue the motility of

LatA-treated MDA-MB-231 cells at 8 cP (Extended Data Fig. 9g), presumably by hyperactivating the osmotic engine model.

In summary, we demonstrate that elevated extracellular viscosity causes mechanical loading at the cell leading edge, which induces intrinsic actin remodelling that facilitates NHE1 recruitment to the cell membrane through its binding partner ezrin. NHE1 enrichment at the cell anterior increases cell volume through water uptake, as postulated by the osmotic engine model[21], which leads to increased membrane tension that in turn triggers the activation of TRPV4 to mediate calcium entry and upregulate downstream RHOA-mediated contractility. The coordinated actions of the osmotic engine model, the ARP2/3-mediated denser actin network and a thicker cortex supported by active RHOA at the cell leading edge enhance motility at elevated extracellular viscosities (Fig. 3i).

## TRPV4- and Hippo-mediated viscous memory

In view of the in vitro data showing that elevated viscosity enhances cell dissemination from 3D spheroids and cell migration on 2D and in confinement, we hypothesized that local increases in extracellular viscosity at the primary tumour site[5,13] confer escaping tumour cells with metastatic advantage. All in vitro assays so far were performed using cells cultured at baseline viscosity (0.77 cP) and acutely subjected to elevated viscosity, which is sufficient to bestow cells with an enhanced migratory propensity. Interestingly, cells that were preconditioned for 6 days at 8 cP and were allowed to migrate either at 8 cP or even at 0.77 cP also exhibited an enhanced migratory potential (Fig. 4a,b), which was abolished by TRPV4 depletion (Fig. 4c). Similarly, cells preconditioned at 2 cP for 6 days retained equally fast motility after switching the medium viscosity back to 0.77 cP (Extended Data Fig. 10a). These data collectively suggest that cells can sense and retain memory of the extracellular viscosity to which they were exposed at any point in their metastatic cascade.

To examine whether elevated viscosity induces TRPV4-dependent mechanical memory through transcriptional regulation, we performed a blinded RNA-sequencing (RNA-seq) analysis to compare the transcriptomes of SC cells subjected to baseline or elevated (8 cP) viscosity for 6 days as well as shTRPV4 cells exposed to elevated viscosity. Principal component analysis (PCA) indicated that the first and second principal components (PC1 and PC2) together accounted for 91% of the variation among these datasets (Extended Data Fig. 11a). Paired comparisons revealed numerous differentially expressed genes (DEGs, false-discovery rate (FDR) < 0.05) between SC cells at 8 cP compared with at 0.77 cP or shTRPV4 cells at 8 cP (Extended Data Fig. 11b). By analysing these DEGs using ingenuity pathway analysis, we identified the Hippo pathway as the most inhibited in SC cells subjected to 8 cP relative to 0.77 cP or compared with shTRPV4 cells at 8 cP (Extended Data Fig. 11c). YAP and TAZ are key effectors of the Hippo signalling pathway that activate downstream transcriptional programs in response to mechanical cues[43]. Interestingly, *YAP1* and *TEAD2* are upregulated in SC cells at 8 cP, whereas shTRPV4 restores their expression to baseline viscosity levels (Extended Data Fig. 11d). To confirm the involvement of Hippo pathway signalling in establishing mechanical memory to viscosity, we first demonstrated that cell exposure to 8 cP increased the nuclear translocation of YAP within the first 4 h relative to exposure to 0.77 cP (Extended Data Fig. 11e). Second, YAP inhibition by verteporfin (0.1 μM) abrogated the development of memory induced by elevated viscosity, as evidenced by the lack of preconditioned cells to display enhanced motility (Extended Data Fig. 11f). Notably, YAP inhibition did not affect motility at baseline viscosity (Extended Data Fig. 11f). Collectively, these data suggest that cells pre-exposed to elevated viscosity acquire TRPV4-dependent mechanical memory through transcriptional control of the Hippo pathway.

To establish the in vivo relevance of viscosity-induced faster cell migration, MDA-MB-231 breast cancer cells were cultured either at 8 cP (preconditioned) or 0.77 cP (naive) for 6 days, and resuspended at baseline viscosity before injection into the dorsal aorta of 3 day old zebrafish embryos[44]. Cell movement in the narrow intersegmental vessels (ISVs) was tracked using confocal microscopy (Fig. 4d,e). Consistent with in vitro findings, preconditioned relative to naive cells moved with a significantly higher speed (Fig. 4f) and persistence (Extended Data Fig. 10b–d) inside the ISVs of the fish embryos.

## Viscous memory promotes cancer dissemination

To determine whether the enhanced in vivo migratory propensity of tumour cells preconditioned at elevated viscosity translates to higher extravasation and tissue colonization, we used the chick embryo and mouse tail-vein injection models[45]. MDA-MB-231 cells, cultured at 3 cP (preconditioned) or 0.77 cP (naive) for 6 days, were resuspended at baseline viscosity and injected into the chick chorioallantoic membrane (CAM) vasculature. Preconditioned cells displayed a higher extravasation potential in vivo compared with naive cells (Extended Data Fig. 10e). Similarly, a markedly higher number of preconditioned (8 cP) relative to naive (0.77 cP) tumour cells colonized the lungs of mice 48 h after tail-vein injection (Fig. 4g,h), as evidenced by quantification of single human vimentin-positive MDA-MB-231 cells[45] (Extended Data Fig. 10f). The single-cell colonies detected at 48 h ultimately led to the formation of more metastatic foci in the lungs after 3 weeks (Fig. 4g,i,j). These observations suggest that preconditioned relative to naive cells were more successful at extravasating into the lungs, resulting in a greater metastatic burden in mice 3 weeks after inoculation. Consistent with the critical role of TRPV4 in conferring increased migratory propensity of preconditioned cells in vitro, TRPV4 depletion abolished the enhanced potential of preconditioned relative to naive cells to colonize the lung tissue in vivo at both the 48 h and 3 week time points (Fig. 4k,l). To validate our confocal microscopy measurements on lung colonization, DNA was extracted from the lungs and human DNA content was evaluated using quantitative PCR (qPCR) with primers specific for human long interspersed nuclear elements[19] (hLINE). qPCR confirmed the enhanced ability of preconditioned cells to form both micro- and macrometastases in vivo (Fig. 4m,n). *TRPV4* knockdown eliminated the enhancement in viscosity-induced single-cell seeding to the lungs at 48 h with a tendency to suppress the seeding of preconditioned, but not naive, cells (Fig. 4m). This intervention also markedly diminished the metastatic burden in both naive and preconditioned cells at the 3 week time point (Fig. 4n).

Here we illustrate that extracellular fluid viscosity is a physical cue that facilitates faster motility and cell dissociation from 3D tumour spheroids in vitro as well as extravasation and cancer dissemination in animal models. Considering that physiologically relevant fluid viscosities in vivo are higher than that of water along with the fact that cells can develop viscosity memory, this research opens a new path for controlling the metastatic potential of cancer cells. It will also be interesting to investigate whether extracellular viscosity also affects other physiologically relevant cellular processes such as morphogenesis.

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

## Methods

### Experimental methods

**Cell culture and reagents.** Human MDA-MB-231 adenocarcinoma (ATCC), MDA-MB-231 BrM2[18] (MDA-MB-231 cells isolated from brain metastasis were a gift from J. Massagué), HOS (ATCC), U87 (ATCC) and HEK293 (ATCC) cells and dermal fibroblasts (Coriell Institute) were grown in DMEM (Life Technologies/Gibco) supplemented with 10% heat-inactivated FBS (Gibco) and 1% penicillin–streptomycin (10,000 U ml[−1]; Gibco). hAOSMCs (PromoCell) were grown in smooth muscle cell growth medium (PromoCell). SUM159 cells were obtained from D. Wirtz (Johns Hopkins University) and were grown in Ham's F-12 medium (Gibco) containing 5% FBS, 1% penicillin–streptomycin (10,000 U ml[−1]; Gibco), 1 μg ml[−1] hydrocortisone (Sigma-Aldrich) and 5 μg ml[−1] insulin (Sigma-Aldrich). Cells were maintained in an incubator at 37 °C and 5% $CO_2$ and passaged at 60–90% confluency every 3 to 5 days. Cells were regularly checked for mycoplasma contamination by PCR using the following primer pairs: forward, 5′-GGGAGCAAACAGGATTAGATACCCT-3′; reverse, 5′-TGCACCATCTGTCACTCTGTTAACCTC-3′.

In select experiments, cells were treated with the following pharmacological agents (purchased from Sigma-Aldrich unless otherwise stated) and corresponding vehicle controls: 2-APB (Calbiochem, 100 μM), CK666 (100 μM), latrunculin A (2 μM), NSC 668394 (Calbiochem, 10 μM), Y-27632 (10 μM), blebbistatin (50 μM), BAPTA (50 μM), GSK2193874 (15 μM), GSK1016790A (1 or 15 μM), BAPTA AM (Invitrogen, 25 μM), EIPA (40 μM), HC-067047 (1 μM) and verteporfin (0.1 μM).

**Preparation and characterization of media of varying viscosities.** Methylcellulose stock solution (3%) in IMDM was purchased from R&D Systems, and media of desired viscosities were prepared by diluting the stock solution with appropriate amounts of IMDM (Life Technologies/Gibco) containing 10% heat-inactivated FBS (Gibco) and 1% penicillin–streptomycin (10,000 U ml[−1]; Gibco). As an alternative means of increasing viscosity, Dextran 500 kDa (Spectrum Chemical) and PVP (K-90; Sigma Aldrich; molecular mass, 360 kDa) were used as suspension in appropriate basal medium. In select experiments, Detran 6 kDa (Alfa Aesar) was used as a control to Dextran 500 kDa, as low-molecular-mass (6 kDa) dextran at the same molarity does not change viscosity appreciably[2]. Cannon–Fenske capillary viscometers were used to measure the viscosity of the resulting medium at 37 °C. Osmolarity was measured using freezing-point depression with a 3205 Single-Sample Osmometer (Advanced Instruments). To ensure the successful entry of the viscous medium into the microchannels, red fluorescent beads of 0.5 μm diameter (FluoSpheres) were dispersed uniformly in the medium, and their flow through the devices was imaged by epifluorescence microscopy.

During medium preparation, up to 0.6% or 92 μM of 65 kDa methylcellulose was used. Thus, osmolarity variations due to addition of methylcellulose can be only up to 100 μOsm, which is insignificant relative to the osmolarity range of basal media that is 3,000× higher (~300 mOsm).

**2D assays.** In wound-healing assay, cells were allowed to grow to confluence on collagen-I-coated (20 μg ml[−1]; collagen I rat tail; Thermo Fisher Scientific) tissue culture plastic plates (Falcon). After reaching confluency, a scratch in the cell monolayer was created using a 200 μl pipette tip with a plastic ruler as a guide. In 2D single-cell motility assays, cells were plated at 5–8% confluency on collagen-I-coated (20 μg ml[−1]) tissue culture plastic plates (Falcon). After cell spreading or after wound creation, the culture medium was replaced with that of the appropriate viscosity, and the acquisition of time-lapse images started 30 min after medium change. In select 2D assays involving high-resolution imaging, cells were plated on collagen-I-coated (20 μg ml[−1]) dishes (Cellvis or IBIDI).

**Microfluidics device fabrication and cell seeding.** PDMS-based microfluidics devices containing an array of parallel microchannels of specific dimensions (200 μm in length, 10 μm in height, and 3.5 μm or 10 μm in width) were fabricated as described previously[20,46]. A laser profilometer was used to verify the height and width of microchannels. Microchannels were coated with collagen I (20 μg ml[−1]) before medium loading and cell seeding. All cell types except for hAOSMCs and human dermal fibroblasts were tested inside narrow ($3.5 \times 10$ μm$^2$) channels. hAOSMCs and human dermal fibroblasts were examined inside $10 \times 10$ μm$^2$ channels.

Microfluidics devices were loaded with medium of appropriate viscosities by filling the inlet wells with 100 μl of medium 15 min before adding cells. Cells were detached from culture flasks or plates using 0.05% trypsin-EDTA (Gibco), followed by centrifugation at 300 g for 5 min and resuspension in medium of the specified viscosity at a concentration of $5 \times 10^6$ cells per ml. After removing the medium from the inlets and outlets of the device, 20 μl of cell suspension was added into the cell-seeding inlet to create a pressure driven flow of cells into the cell-seeding channel. Cells were allowed to adhere and spread outside the microchannel entrances for at least 20 min before filling all inlet wells to 100 μl with medium at the appropriate viscosities. Devices were incubated at 37 °C and 5% $CO_2$ before imaging. Cells were not subjected to any chemotactic stimulus or medium flow during the migration assay. In experiments with pharmacological agents, the medium in the devices was replaced with that containing the drug of consideration or its respective vehicle control once cells had fully entered the microchannels. By imaging the flow of fluorescent beads dispersed in 8 cP medium, we confirmed their successful entry into the microchannels (Supplementary Video 5).

**Cell migration tracking and analysis.** Cells were imaged every 2 to 20 min for 12 to 20 h on an inverted Nikon Eclipse Ti microscope (Nikon) with automated controls (NIS-Elements; Nikon) and a ×10/0.45 NA ph1 objective using time-lapse microscopy. In select experiments, cells were imaged using fluorescein isothiocyanate filters. For experiments with NHE1-GFP-transfected cells, a Nikon A1 confocal microscope (Nikon) with ×20 or ×60 objective was used. During the course of the experiments, cells were maintained on a stage-top incubator (Tokai Hit) at 37 °C and 5% $CO_2$. The A1 confocal system also had a temperature-controlled cage.

Live-cell videos were exported to ImageJ (National Institutes of Health), and the MTrackJ plugin[47] was used for cell motility tracking. In single-cell 2D migration assays, cells that were not in contact with any neighbours were chosen and manually tracked. Cell tracking was performed until the end of the video unless a neighbouring cell touched the tracked cell, in which case tracking was stopped. In microfluidics experiments, cell paths were recorded from the time of complete entry into the microchannel until contact was made with the end of the microchannel. A custom MATLAB (MathWorks) script was used to analyse cell migration speed and velocity from the cell paths. Cell entry time was calculated by manually evaluating the time elapsed between leading and trailing edge entry into a microchannel. Dividing or apoptotic cells were excluded from analysis in both 2D and microchannel migration. Wound-healing assays were analysed using ImageJ by drawing a line indicating the migrating fronts on two sides of the scratch wound. Wound closure was recorded when both apposing fronts touched each other.

For cell migration phenotype classification, live cells expressing Lifeact–GFP or fixed cells stained with Alexa Fluor 488 phalloidin (1:100; Invitrogen) and Hoechst (1:2,000; Invitrogen) were observed using an inverted Nikon Eclipse Ti microscope (Nikon) using a ×40 air objective 4 to 8 h after channel entry. Cell phenotype under confinement was tabulated manually, as reported previously[47].

**Spheroid assays.** Spheroids with MDA-MB-231 cells were generated as described previously[48]. In brief, growth-factor-reduced Matrigel

(Corning) was diluted 1:3 with cold DMEM supplemented with 10% heat-inactivated FBS and 1% penicillin–streptomycin. Then, 50 μl of the diluted Matrigel was transferred to each well of a 96-well plate (Falcon) and allowed to polymerize at 37 °C and 5% $CO_2$ for 1 h. Cells were collected from culture flasks, resuspended in diluted Matrigel and were gently dispensed (2,000 cells per 50 μl) on top of the previously polymerized Matrigel in each well. After incubation at 37 °C and 5% $CO_2$ for 1.5 h, 100 μl of DMEM with 10% heat-inactivated FBS and 1% penicillin–streptomycin was added into each well. The medium was replaced every 2 days until spheroids were collected for experiments (typically ~10–15 days later). Spheroids were collected after dissolving the Matrigel in cold DMEM and centrifuged at 2,655$g$ for 5 min. After removing the supernatant, the spheroids were resuspended in either 0.77 or 8 cP medium and plated onto collagen-I-coated (20 μg ml$^{-1}$) 24-well plates. After 60 min, the spheroids were imaged every 20 min for at least 20 h on an inverted Nikon Eclipse Ti microscope with automated controls (NIS-Elements; Nikon) with a ×10/0.45 NA ph1 objective using time-lapse microscopy. During the course of the experiments, the spheroids were maintained on a stage-top incubator at 37 °C and 5% $CO_2$. First-cell dissociation times were acquired using the NIS Element Software (Nikon) by manually recording the time required for the first cell to fully detach from each intact spheroid.

**Cloning, lentivirus preparation, transduction and transfection.** To generate MDA-MB-231 cells with stable knockdown of *ARP3* and/or *ARPC4*, or *TRPV4*, the pLKO.1 puro (plasmid 8453; Addgene; a gift from B. Weinberg) backbone was used. The following sequences were subcloned into the pLKO.1 puro backbone: Non-targeting scramble sequence: 5′-GCACTACCAGAGCTAACTCAGATAGTACT-3′; shARP3 sequence 1: 5′-GTAACACCAAACATGATTATA-3′; shARPC4 sequence 1: 5′-GCTGAAGAGTTCCTTAAGAAT-3′; shITGB1 sequence 1: 5′-TGCCTACTTC TGCACGATGT-3′; shTRPV4 sequence 1: 5′-TCTTGGTAACAAACTTGGT-3′; and shTRPV4 sequence 2: 5′-TGCTCCTATGGAGTCACATAA-3′.

Whereas shTRPV4 sequence-1 was used in Extended Data Fig. 6, in the rest of the manuscript, sequence-2 was used. For *AQP5* knockdown, the following targeting sequence was subcloned into the pLVTHM (plasmid 12247; Addgene; a gift from D. Trono) backbone: shAQP5 sequence 1: 5′-ACGCGCTCAACAACAACACAA-3′.

For *NHE1* depletion, the following sequences were subcloned into both the pLKO.1 puro and pLVTHM backbones and used with SC sequences subcloned into the respective backbones: shNHE1 sequence 1: 5′-GACAAGCTCAACCGGTTTAAT-3′; and shNHE1 sequence 2: 5′-CCAATCTTAGTTTCTAACCAA-3′.

Whereas the shNHE1 sequences were used individually in Extended Data Fig. 5, both sequence-1 and sequence-2 were used in combination in all of the other experiments. MDA-MB-231 cells stably transduced with shMIIA and shMIIB[3] were used for migration assays. For actin retrograde flow analysis, the shMIIA sequence was cloned into the pLKO.1 puro backbone. After every cloning, sequence integrity and orientation were verified by Sanger Sequencing at the JHU Genetic Resources Core Facility.

For lentivirus production, HEK293T/17 cells were co-transfected with psPAX2, pMD2.G and the lentiviral plasmid of interest. Then, 48 h after transfection, the lentivirus was collected and concentrated using centrifugation. Wild-type MDA-MB-231 cells at 60–80% confluency were incubated for 24 h with 100× virus suspension and 8 μg ml$^{-1}$ of Polybrene transfection reagent (Millipore Sigma). To maintain stable knockdown, cells transduced with the pLKO.1 puro backbone were cultured in medium containing 0.5 μg ml$^{-1}$ puromycin (Gibco).

**Live-cell reporters.** The pGP-CMV-GCaMP6s (40753; a gift from D. Kim and the GENIE Project), FUGW-pHRed (65742; a gift from M. Tantama), pLentiRhoA2G (40179; a gift from O. Pertz), pLenti.PGK. LifeAct-GFP.W and pLenti.PGK.LifeAct-Ruby2.W (51010 and 51009 respectively; gifts from R. Lansford) plasmids were purchased from

Addgene and used for lentiviral cell transduction. The GFP-AHD plasmid was offered by M. A. Glotzer[49], and was cloned into the lentiviral backbone modified from pLV-EF1a-IRES-Puro (plasmid 85132; Addgene; a gift from T. Meyer) using BamHI and NotI. The following sequences were used to amplify GCaMP6s: 5′-CAAATGTGGTATGGCTGATTATG-3′ and 5′-GTACGCGTCACCATGGGTTCTCATCATC-3′. The PCR product was subsequently inserted into pLV-EF1a-IRES-Puro using MluI and NotI. pEYFP-C1 (Clontech)[2] was cloned into pLV-EF1a-IRES-Puro using the following sequences and NotI and HpaI restriction sites: 5′-GTGCGGCCGCGTCGCCACCATGGTGAGC-3′ and 5′-CTGTTAACTCAC TTGTACAGCTCGTCCATGC-3′. The aforementioned plasmids were used to prepare lentiviral particles for stable cell transduction.

Arp3-pmCherryC1 (27682; a gift from C. Merrifield) and β-actin-mRFP-PAGFP (62382; a gift from G. Charras and T. Mitchison) were purchased from Addgene. The NHE1-GFP plasmid was a gift from J. Orlowski. The above-mentioned plasmids were used for transient transfections. About 60–80% confluent MDA-MB-231 cells were transfected using Lipofectamine 3000 reagent according to the manufacturer's recommendations. To enrich the population of cells expressing Arp3-pmCherryC1, β-actin-mRFP-PAGFP or NHE1-GFP, cells were selected with 0.5 mg ml$^{-1}$ G418 (Corning) and sorted. Cells expressing Myosin IIA-GFP were generated as previously described[3].

**Lamellipodial actin staining and imaging.** To stain for F-actin, a previously described protocol[50] was used for cells plated on collagen-I-coated glass bottom dishes (Cellvis). In brief, when cells reached about 20% confluency, their medium was replaced with that of the desired viscosity and they were incubated for 2–6 h at 37 °C. Cells were then washed once with PBS containing $Ca^{2+}$/$Mg^{2+}$ (Gibco), fixed and extracted for 1 min using 0.25% glutaraldehyde and 0.5% Triton X-100 in cytoskeleton buffer (150 mM NaCl, 10 mM MES buffer, 5 mM EGTA, 5 mM glucose, 5 mM $MgCl_2$ in water; pH 6.8) and then fixed for 5 min in 2% glutaraldehyde in cytoskeleton buffer. The samples were either processed immediately or stored with 2% glutaraldehyde in cytoskeleton buffer at 4 °C until subsequent staining. The background autofluorescence from glutaraldehyde was quenched with 1% $NaBH_4$ for 15 min before blocking with 5% bovine serum albumin (BSA) for 1 h. Cells were stained with rhodamine or Alexa Fluor 488 phalloidin (1:100; Invitrogen) and Hoechst (1:2,000; Invitrogen). High-resolution images were acquired using the Zeiss LSM800 Confocal with AiryScan super-resolution module with an $x,y$ resolution of 120 nm and a $z$-resolution of 350 nm.

**STORM sample preparation, imaging and analysis.** Samples were prepared, fixed and extracted as described in the 'Lamellipodial actin staining and imaging' section. Next, fixed samples were treated with 0.2% $NaBH_4$ for 7 min, and three times with 0.5% glycine in PBS for 10 min, before blocking with 5% BSA solution containing 0.1% Triton X-100 (Sigma-Aldrich) for 1 h. The samples were subsequently stained overnight at 4 °C with Alexa Fluor Plus 647 Phalloidin (1:400; Invitrogen). On the day of imaging, the following solutions were prepared: buffer A (10 mM Tris at pH 8.0 containing 50 mM NaCl), buffer B (50 mM Tris at pH 8.0 containing 10 mM NaCl and 10% glucose) and GLOX (14 mg glucose oxidase (Millipore Sigma) and 0.85 mg catalase (Millipore Sigma) dissolved in 200 μl buffer A and 50 μl distilled $H_2O$). The samples were washed five times with PBS containing 0.05% Triton X-100 for 5 min. Immediately before imaging, fresh STORM imaging buffer (7 μl GLOX solution, 7 μl 2-mercaptoethanol and 690 μl buffer B) was prepared and added to the dishes. A glass coverslip was gently placed on top to seal off the samples from atmospheric oxygen, and the dishes were mounted on the scope.

Images were acquired on a Nikon Eclipse Ti-E (Nikon) microscope equipped with a ×100/1.49 NA Plan-Apo objective, 647 nm fibre laser operated at 500 mW and Evolve 512 EMCCD camera (Photometrics). A total of 50,000 cropped images (128 × 128 px$^2$) were captured in

NIS-Elements (Nikon) at 120 fps. TIRF was used to limit fluorescence to a thin region of the sample and the Nikon Perfect Focus System was used to reduce $z$-drifts. STORM images were reconstructed and visualized using Picasso software[51]. Localizations were filtered by the point spread function width (sx = 2) and height (sy = 2) and localization precision in $x$ (lpx = 2) and $y$ (lpy = 2) in camera pixels. For analysing actin density, the reconstructed images were exported to ImageJ and the fibres were transformed into binary masks using the Auto Local Threshold function of ImageJ, using the Phansalkar algorithm. The edge of the cells was manually traced and the percentage of ON pixels over total pixels was calculated within 1.5 µm from the cell edge.

**Cell volume and lamella area measurements.** Cell volume measurements were performed as described previously[20]. Cells stably expressing Lifeact–GFP were imaged using a Nikon A1 confocal microscope with a ×60 oil-immersion objective within 2–6 h after the replacement of the medium with that of the desired viscosity. Images were acquired at 1,024 × 1,024 px resolution, and confocal slices were spaced at 0.5 µm apart. A 488 nm laser was used to image the cells and obtain their boundaries from the Lifeact–GFP signal. Throughout the course of the experiments, cells were maintained at 37 °C and 5% $CO_2$ inside a stage-top incubator and cage (Tokai Hit).

Cell volume was measured from $z$-stacks using a custom MATLAB script described previously[20]. Out-of-focus $z$-planes were removed. Images from each analysed focal plane were processed using the binary thresholding function in MATLAB to remove noise. The cell boundary was detected in each image using the Canny Edge Detector operator. The edge was dilated and refilled as to obtain the cross-sectional area of the cell from each slice. The volume was calculated by multiplying the $z$-slice interval by the average area from two adjacent slices and integrating all of the values. In select experiments, cell volume was analysed using volume measurement wizards of NIS-Elements v.5.02.01 (Nikon) and Imaris v.9.7.0 (Bitplane).

In addition to measuring the projected area of the entire cell, we assessed the area of the part of the cell above a critical height from the substrate. A critical height of 2 µm was chosen after taking into consideration the typically reported lamella thickness of 1 µm. Notably, this threshold eliminated possible $z$-aberrations from confocal imaging. The projected lamella area was calculated by subtracting the projected area of the cell above the critical height from the overall projected cell area. To normalize the measurements with respect to the cell size, data were reported as projected area of lamella over projected area of the entire cell.

**Lamella growth imaging and analysis.** Cells stably expressing Lifeact–GFP or Lifeact–Ruby were plated on collagen-I-coated glass-bottom dishes (Cellvis) at about 20% confluency. After treating the cells with medium of the desired viscosities for 2–6 h, time-lapse images were acquired every 5 s for 110 s or 2 min with a Nikon A1 confocal microscope equipped with a ×60 oil-immersion objective. Cells were maintained at 37 °C and 5% $CO_2$ inside a stage-top incubator and cage.

Time-lapse images were imported into ImageJ and the growing front of the cells was manually extracted. Subsequently, a custom macro was used to identify the instantaneous cell area and edge length (or perimeter). The lamella growth at every instance ($i$), was calculated by using the ratio of area change over the last perimeter, using the formula:

$$\text{Growth}(i) = \frac{\text{Area}(i) - \text{Area}(i-1)}{\text{Perimeter}(i-1)}$$

A heat map summarizing instantaneous leading-edge lamella growth was prepared by combining all of the instances from cells at a particular viscosity.

**Actin retrograde flow imaging and analysis.** For measuring actin retrograde flow, MDA-MB-231 cells stably expressing β-actin-mRFP-PAGFP

were plated on collagen-I-coated glass-bottom dishes (Cellvis). Cells were imaged using the FRAP capabilities of a Nikon A1 confocal microscope (Nikon) with a ×60 oil-immersion objective. The direction of cell motility was manually identified by visual inspection over 3 min, and then a rectangular region of interest (ROI; 60.75 × 7.25 µm²) was drawn with its length aligned perpendicular to the cell leading edge. A point in the cytoplasm was exposed to ultraviolet light (405 nm) at 20% intensity for 500 ms, and then imaged at GFP and RFP channels, every 100 ms or 400 ms for 30 s after ultraviolet illumination.

For analysis of retrograde flow, the time-lapse videos were imported into ImageJ. RFP and GFP channels were individually extracted, converted to binary and smoothed twice. Next, a kymograph was drawn perpendicular to the leading edge for the image with GFP signal, and the angle subtended by the retrograde flow of the photo-converted PAGFP molecule was used to estimate the flow rate with respect to the laboratory frame. The uninterrupted leading edge growth rate was quantified by measuring the slope generated by the cell leading edge in the kymograph before a retraction event occurred. The image from the RFP channel was used to locate the cell edge before photoactivation.

**Measurement of epifluorescent FRET-based RHOA activity.** Cells expressing the RhoA2G biosensor were plated onto collagen-I-coated glass-bottom dishes. Within 2–6 h of exposing cells to medium of the desired viscosity, images at CFP (excitation: 430/24, emission: 470/24), YFP (excitation: 500/20, emission: 535/30) and FRET (excitation: 430/24, emission: 535/30) settings were acquired. For RHOA-activity quantification, the mean pixel ratio of FRET over CFP was measured after subtraction of the background signal from the corresponding channels[20,52]. The image in the YFP channel was used for tracing the cell boundary. Imaging and quantification were performed using Nikon Elements. For control experiments with FBS (10%) activation, cells were serum-starved for 24 h before adding medium containing 10% FBS, and the images were acquired within 0.5–2 h.

**FLIM analysis of membrane tension and RHOA activity.** For membrane-tension measurements, cells, plated on collagen-I-coated glass-bottom dishes and incubated for 3–4 h in media of the specific viscosities in the absence or presence of the indicated drugs, were stained with the live-cell membrane tension probe Flipper-TR (Spirochrome, 0.5 mM) and imaged immediately thereafter. Confocal fluorescence lifetime imaging microscopy (FLIM) analysis of live cells stained with Flipper-TR or stably expressing the RhoA2G sensor was performed as described previously[20,47] using a Zeiss LSM 780 microscope and a PicoQuant system consisting of a PicoHarp 300 time-correlated single-photon counting (TCSPC) module, two-hybrid PMA-04 detectors and a Sepia II laser control module. For Flipper-TR imaging, a 485 nm laser line was used for excitation with 600/50 band-pass detector unit. Cells were maintained at 37 °C and 5% $CO_2$ during imaging inside a PeCon environment chamber and stage.

**FLIM reconvolution, image segmentation and segmentation quantification.** The FLIM data were processed as described previously[20,47] using SymPhoTime 64 (PicoQuant) software. In brief, a customized script was used for the calculation of the internal response function from 100 data points with no smoothing. After binning the data to ensure at least 500 photons per binned pixel, a cell-specific threshold was applied to eliminate out-of-cell fluorescence. The fluorescence decays were fit into every binned pixel using the three-exponential reconvolution. For Flipper-TR quantification, the longest lifetime ($\tau_1$) with the higher fit amplitude was recorded for each cell according to the manufacturer's recommendations. For RHOA-FLIM segmentation quantification, the intensity-weighted fluorescence lifetime averages ($\tau_{AI}$) were measured using SymPhoTime 64 in different areas of the cells. Data from the nuclear region were excluded during FLIM segmentation using a free-form ROI because the RHOA FRET sensor was excluded

from the nucleus, resulting in low and variable photon counts in the centre of the cells. The entire cell without the nucleus is defined as 'whole', whereas the lamella region at cell edge is identified from DIC images (Extended Data Fig. 8d). The 'body' of the cell comprises of the remainder of the cell without the lamella and the nucleus.

**Calcium dynamics imaging and quantification.** Cells transiently or stably expressing GCaMP6s were plated on collagen-I-coated glass coverslips with PDMS walls[3] and were grown to 60–80% confluency. Within 5 min of the addition of the indicated stimulus, cells were imaged every minute for 80–120 min with a ×20 objective on a Nikon A1 confocal microscope using the 488 laser and the GFP filter settings. The GFP signal of representative cells was manually traced in ImageJ over time by measuring the mean intensity of a circular ROI inside the cell. Calcium spikes were identified as instances with greater than 2× intensity over the baseline levels.

**Osmotic shock experiments.** For application of osmotic shocks, hypotonic media were prepared and their osmolarity was measured as described previously[20]. Medium solutions were prepared by diluting basal IMDM medium (292 mOsm) twofold (2×, resulting in 141 mOsm) or four-fold (4×, 71 mOsm) with ultrapure water. For FLIM measurements, cells were serum-starved for 24 h before measuring RHOA activity within 5–30 min after application of the osmotic shock. Cell volume measurements were also completed within 50 min after shock.

**Immunofluorescence and image analysis.** For pMLC, Ki-67 and YAP1 staining, cells were fixed with 4% paraformaldehyde (Thermo Fisher Scientific), permeabilized in 0.1% Triton X-100 and blocked for 1 h at room temperature with 5% BSA solution containing 2% (for YAP1) or 5% normal goat serum (Cell Signaling) and 0.1% Triton X-100. Next, cells were incubated at 4 °C with an anti-pMLC (Ser19) antibody (raised in rabbit; Cell Signaling; 3671; 1:100), anti-YAP antibody (raised in mouse; clone 63.7; Santa Cruz Biotechnology; sc-101199; 1:50) or anti-Ki-67 antibody (raised in mouse; 8D5; Cell Signaling, 9449; 1:800) overnight. For NHE1 and ezrin staining, cells were fixed with 4% paraformaldehyde, permeabilized in 0.1% Triton X-100 and blocked for 1 h at room temperature with 1% BSA containing 0.1% Triton X-100. Subsequently, cells were incubated at 4 °C with an anti-NHE1 antibody (raised in mouse; 54; Santa Cruz Biotechnology; sc-136239; 1:50) or anti-ezrin antibody (raised in rabbit; Cell Signaling; 3145S; 1:200) overnight. After washing three times with PBS, samples were incubated for 1 h at room temperature with Alexa Fluor 488 goat anti-rabbit immunoglobulin G (IgG) H+L (Invitrogen; A11034; 1:200), Alexa Fluor 488 goat anti-mouse immunoglobulin G (IgG) H+L (Invitrogen; A11029; 1:200) or Alexa Fluor Plus 647 goat anti-mouse immunoglobulin G (IgG) H+L (Invitrogen; A32728; 1:100). All antibodies were prepared in the respective blocking buffer. Nuclei and actin were also stained with Hoechst 33342 (Invitrogen; 1:2,000) and rhodamine phalloidin (Invitrogen; 1:200), respectively. Samples were imaged using a ×60 oil-immersion objective on a Nikon A1 confocal system. Samples stained with only secondary antibodies were used as negative controls.

The spatial distribution of pMLC and ARP3 was evaluated by drawing a rectangle along the entire length of the cell and recording the average intensity across the rectangle's width using the Plot Profile function in ImageJ. Raw intensity data from each cell were compiled in Excel, and the moving average was calculated after normalizing the intensity with respect to the lowest signal for the cell. For quantification of NHE1 and ezrin polarization, free-form regions at the front and rear edges of the cells were manually drawn, and their mean intensity values were recorded. The front-to-rear ratio was calculated from the mean intensity values for each cell. For evaluation of the percentage of Ki-67-positive cells, $2,396 × 1,917 \ \mu m^2$ images were exported to ImageJ, and the total number of nuclei identified by DAPI as well as the number of Ki-67-positive cells were enumerated using the Find

Maxima function on the respective fluorescent channels. To quantify the nuclear-to-cytoplasmic ratio of YAP1, the mean fluorescence intensity of YAP1 within a manually drawn ROI inside the nucleus was divided by the mean intensity of the same ROI placed outside the nucleus.

**qPCR analysis.** Scramble control, shTRPV4 or shAQP5 cells were grown to 95% confluency and their total RNA was isolated using Direct-zol RNA isolation kit (Zymo Research) according to manufacturer's recommendations. Reverse transcription and qPCR were performed using standard techniques described previously[19] using the following primer sets: *TRPV4*: F, 5′-CCCGTGAGAACACCAAGTTT-3′ and R, 5′-GTGTCCTCATCCGTCACCTC-3′; *AQP5*: F, 5′-CAGCTGGCACTCTGC ATCTT-3′ and R, 5′-TGAACCGATTCATGACCACC-3′; *MIIA*: F, 5′-ATCCTGG AGGACCAGAACTGCA-3′ and R, 5′-GGCGAGGCTCTTAGATTTCTCC-3′; *MIIB*: F, 5′-GCTGATGGCAACTCTCCGAAAC-3′) and R, 5′-CTTCCAGGACA CCATTACAGCG-3′); *MIIC*: F, 5′-CAGCCGTCAAATGCAAACCGAG-3′ and R, 5′-TTGCCTCTGTCGTCACCTTCTC; 18S: F, 5′-CAGCCACCCGAGATT GAGCA-3′ and R, 5′-TAGTAGCGACGGGCGGTGTG-3′.

**Western blotting.** Western blots were performed as described previously[19,20,47] using NuPage 4–12% gels and the following antibodies. Uncropped, original blots are provided in Supplementary Fig. 1.

The following primary antibodies were used: anti-TRPV4 antibody (raised in mouse; 1B2.6; Millipore Sigma; MABS466; 1:1,000), anti-ARP3 (mouse; FMS338; Abcam; ab49671; 1:5,000), anti-ARPC4 (rabbit; Abcam; ab217065; 1:2,000), anti-integrin β1 antibody (rabbit; Cell Signaling; 4706S; 1:1,000) and anti-NHE1 (raised in mouse; 54; Santa Cruz Biotechnology; sc-136239; 1:200). GAPDH was used as a loading control (rabbit; 14C10; Cell Signaling; 2118S; 1:5,000).

The following secondary antibodies were used: anti-mouse IgG, HRP-linked antibody (Cell Signaling; 7076S; 1:1,000) and anti-rabbit IgG, HRP-linked antibody (Cell Signaling, 7074S; 1:1,000).

**Electrophysiological recording.** Ionic currents were recorded in the whole-cell patch-clamp mode[32]. Patch pipettes were filled with a solution containing 140 mM N-methyl-D-glucamine chloride, 1 mM $MgCl_2$, 5 mM EGTA, 10 mM HEPES, 4 mM ATP and 0.1 mM GTP (300 mOsm l⁻¹, pH 7.3). The bath solution contained 125 mM NaCl, 1.5 mM $MgCl_2$, 1 mM EGTA, 10 mM HEPES and adjusted to 305 mOsm l⁻¹ (with D-mannitol) and pH 7.36. Cells were held at 0 mV and ramps from −100 mV to +100 mV (400 ms) were applied at a frequency of 0.2 Hz. Ramp data were acquired at 10 kHz and were low-pass filtered at 1 kHz using Axon p-Clamp software. Experiments were performed at room temperature.

**NHE1 activity measurement.** To measure NHE1 activity, two different probes and methods were used. A methodology described in ref. [53] was used, which quantifies the rate of pH recovery in cells stably expressing the ratiometric pH sensor pHRed[28] after intracellular acidosis due to brief exposure to $NH_4Cl$. The second methodology used was the quantification of intracellular pH in SC and shNHE1 cells loaded with the pHrodo red-AM (Molecular Probes; P35372) as well as the measurements of changes in pHrodo signal in cells exposed to the NHE1 inhibitor EIPA. MDA-MB-231 cells expressing pHRed were excited at 561 nm and 405 nm laser lines, and emission was collected using a 600/50 detector. Intracellular pH is inversely proportional to the ratio of intensities at 561 nm over 405 nm ($R = I_{561}/I_{405}$).

Cells expressing pHRed were plated on collagen-I-coated glass-bottom dishes (Cellvis) at 20% confluency. After 2–6 h exposure to media of desired viscosities, cells were incubated with DMEM supplemented with 10% heat-inactivated FBS and 1% penicillin–streptomycin for 10 min. Next, cells were imaged every 30 s for 2 min on a Nikon A1 confocal microscope with a Plan Apo ×60 objective. Thereafter, the medium was gently aspirated and replaced with DMEM containing 10% heat-inactivated FBS, 1% penicillin–streptomycin and 15 mM $NH_4Cl$. Cells were then imaged every 30 s for 4 min before

replacing $NH_4Cl$-containing medium with DMEM supplemented with 10% heat-inactivated FBS and 1% penicillin–streptomycin. Imaging was continued every 30 s for another 5 min. For analysis, the boundary around each cell was automatically traced at every time point using the ROI detection function in NIS Element (Nikon), and the ratiometric intensity values over time were exported. GraphPad Prism (GraphPad) was used to plot the time-dependent variation in intracellular pH and linear curves were fit to the pH recovery phase of each cell. The slope from the fitted line was used to quantify the rate of proton efflux or NHE1 activity where the rate of pH recovery is -d$R$/d$t$.

Intracellular pH was measured using pHrodo Red AM according to the manufacturer's instructions. In brief, cells grown on 13 mm coverslips were exposed to 1 ml of bathing solution containing a dilution of 1 µl of 5 mM pHrodo Red AM in 10 µl of PowerLoad 100× concentrate at 37 °C for 30 min. Cells were washed twice with isotonic solution and allowed to settle for 5 min at room temperature in a handmade perfusion chamber. Video microscopic measurements of pHrodo Red AM fluorescence were obtained using an Olympus IX70 inverted microscope with a ×20 objective (Olympus). ROIs of GFP-positive cells were selected before running the experiment using a 488 nm excitation filter. To monitor pHrodo Red signal, cells were excited at 572 nm using a xenon arc lamp coupled to computer-controlled filter wheel (Lambda Series, Sutter Instrument). Fluorescence images were collected by an ORCA Flash4plus camera (Hamamatsu Photonics) after being passed through eGFP/mCherry dual-emission filter (Chroma Technology) using the HCI software (Hamamatsu Photonics). Images were acquired every 5 s after 65 ms exposure time. Basal fluorescence levels were recorded for 2 min in isotonic solution and the medium was then exchanged to isotonic medium containing 10 µM EIPA for 10 min. Subsequent intraexperimental calibration of the probe was performed using a commercially available intracellular pH calibration buffer kit (Molecular Probes; P35379) according to the manufacturer's instructions. In brief, after each experiment, 2–3 ml of 7.5, 6.5, 5.5 and 4.5 pH calibration standards containing 10 µM valinomycin and 10 µM of nigericin were gently perfused for 3–5 min. An average of the last ten data points of each standard measurement were plotted and fitted to a standard linear curve. A linear trend equation was used to extrapolate the experimental pH values.

**Cell exposure to medium of different viscosities.** In some experiments, cells were plated in tissue culture flasks/plates in medium of various viscosities. The medium was replaced every alternate day, and cells were passed to a fresh flask/plate after reaching 90% confluency. Cells were maintained in medium of the desired viscosity for 6 days before being tested in in vitro and in vivo assays.

**RNA-seq and analysis.** Total RNA was extracted and purified using the Zymo Research Quick-DNA/RNA MiniPrep Plus Kit (Zymogen) according to the manufacturer's recommendations. Strand-specific mRNA libraries were generated using the NEBNext Ultra II Directional RNA library prep Kit for Illumina (New England BioLabs, E7760), and mRNA was isolated using the poly(A) mRNA magnetic isolation module (New England BioLabs, E7490). Preparation of libraries was performed according to the manufacturer's protocol (New England BioLabs, 2.2 05/19). Input was 1 µg and the samples were fragmented for 15 min for an RNA insert size of ~200 bp. The following PCR cycling conditions were used: 98 °C for 30 s; followed by 8 cycles of 98 °C for 10 s and 65 °C for 75 s; and a final extension of 65 °C for 5 min. Stranded mRNA libraries were sequenced on an Illumina NovaSeq instrument, SP flowcell using 100 bp paired-end dual-indexed reads and 1% PhiX control. All of the samples had over 39 million reads.

We mapped the reads to the human GRCh38.p13 genome (hg38) using the HISAT2 package[54], annotated each gene using Ensemble104 and totalled the number of exon reads for each pair using the HTSEQ package. The human GRCh38.p13 genome (hg38) was obtained from

Ensembl. The DESeq2.r package was used (in R, v.4.0) to normalize and compare the reads for each gene between the samples of interest, with respective $P$ values for each gene per comparison and PCA score for each sample (Extended Data Fig. 11a). We selected differentially expressed genes with $P \le 0.05$ between SC samples at 0.77 cP versus 8 cP and between SC versus shTRPV4 at 8 cP. These data were presented in volcano plots (Extended Data Fig. 11b) and used for pathway analysis using ingenuity pathway analysis (Qiagen) (Extended Data Fig. 11c). The $P$ values for the differentially expressed genes were generated by the DESeq2.r package (assuming a negative binomial model and corrected for FDR). Fisher's exact method was used to calculate the significance of pathway enrichment from ingenuity pathway analysis with a threshold of $P \le 0.05$. Sample preparation and RNA-seq analysis were performed in a blinded manner.

**Zebrafish husbandry.** Animal studies were conducted in K. Tanner's laboratory under protocols approved by the National Cancer Institute and the National Institutes of Health Animal Care and Use Committee. Sexually mature zebrafish used for breeding were maintained at 28.5 °C under a 14 h–10 h light–dark cycle. Transgenic Tg(*mpx:eGFP/ flk:mCherry*) zebrafish[55,56] were obtained from natural spawning, kept at 28.5 °C until the time of cell injection, and maintained in fish water (60 mg Instant Ocean sea salt (Instant Ocean) per litre of deionized water). To inhibit melanin formation and maintain optical transparency, embryos were transferred to fish water supplemented with *N*-phenylthiourea (PTU; Millipore Sigma) between 18–22 h after fertilization. PTU water was prepared by dissolving 16 µl of PTU stock (7.5% w/v in DMSO) per 40 ml of fish water. Water was replaced daily. Injected fish were maintained at 33 °C after injection and during intravital imaging.

**Zebrafish circulatory injections.** Before injection, cells were washed three times with PBS and incubated for 20 min at 37 °C at a concentration of $2 \times 10^6$ cells per ml in PBS containing 1 µM CellTracker Deep Red (Thermo Fisher Scientific). Stained cells were then washed with PBS and resuspended to a concentration of $1 \times 10^6$ cells per 20 µl in PBS for injection.

For injection to the circulation, 3 days post-fertilization zebrafish were anaesthetized in 0.4% Tris-buffered tricaine and oriented in a lateral orientation on an agarose bed. Then, 2–5 nl of cell suspension was injected using a pulled-glass micropipette in the circulation and directed towards the tail through the dorsal aorta. Fish from a given clutch were randomly divided into experimental groups before injection, and experiments were blocked so that treated and control cells were injected into larvae from the same clutch for three independent clutches. Injected fish were screened between 1–4 h after injection to check for successful dissemination of cells through the circulatory system, and were then imaged as described below. Fish lacking cells in the circulation were euthanized. Gonad differentiation to determine sex had not been completed by the time points used in these experiments.

**Intravital microscopy of cell trafficking.** Zebrafish larvae were anaesthetized and immobilized in a lateral orientation in 1% (w/v) low-gelling-temperature agarose (Millipore Sigma) dissolved in fish water approximately 4 h after cell injection. To enable high-resolution confocal imaging, fish were laterally oriented in coverglass-bottom chamber slides (Nunc Lab-Tek Chambered #1.0 Borosilicate Coverglass slides, Thermo Fisher Scientific). PTU water supplemented with 0.4% buffered tricaine was then added to the imaging chamber to keep the larvae anaesthetized over the course of the experiment.

One-photon, confocal 2D images were acquired at a resolution of 512 × 512 pixels, which were stacked to acquire 3D images. Images were obtained on the Zeiss 780 or 880 confocal microscope. For each larva, 1–2 confocal $z$-stacks centred on the intersegmental vessels were acquired every 10 min for ~50 frames at 2 µm axial steps to

image a total depth of ~100 μm. Images were acquired with a Zeiss ×20 EC Plan-Apochromat, 0.8 NA objective. The samples were simultaneously excited with 488 nm light from an argon laser, 561 nm light from a solid-state laser and 633 nm light from a HeNe633 solid-state laser. The zebrafish larva was maintained at 33 °C for the course of imaging on a stage top incubator.

**Intravital cancer cell tracking.** Time-lapse microscopy images were exported to ImageJ for analysis. Images were analysed only if blood flow was visually confirmed throughout the larvae. To adjust for fish growth during imaging, images were first registered by phase-correlation-based translational registration using the Correct 3D Drift plugin[57], with the vasculature of the fish used as a topographical reference. The multi-time scale computation, subpixel drift correction and edge enhancement options were enabled during registration. Cancer cells were tracked in 3D in registered images for up to 12 h from the registered image stacks using the TrackMate plugin for ImageJ[58]. Tracks were visually inspected for accuracy over the entire acquisition period and were manually edited to ensure that point-to-point tracks were generated for the entire time that a cell moved through the intersegmental vessels.

Cell tracks ($x,y,t,z$) were exported to MATLAB for analysis. For each cell, the frame-to-frame speed was calculated by dividing the displacement of the cell by the time interval between frames. An average speed for that cell over the course of imaging was then calculated by averaging these frame-to-frame speeds. The persistence of migration was calculated by dividing the net displacement of a given cell over the course of tracking by the total point-to-point distance travelled. Speed and persistence values were calculated only for cells remaining in the intersegmental vessels for at least 4 frames, and values were calculated over a maximum of 71 frames. The average speed of cell movement through the intersegmental vessels for a given larva was calculated from mean cell speeds.

**Ex ovo chick embryo cancer cell extravasation model.** Fertilized white leghorn chicken eggs were acquired from the University of Alberta Poultry Research Centre and maintained at 38 °C. Embryos were isolated from their shells after 4 days of incubation and maintained under shell-less conditions in a covered dish placed in an air incubator at 38 °C and 60% humidity[2]. A protocol described previously[59] was adopted to image and quantify tumour cell extravasation in chick embryos, which were randomly divided into experimental groups before injection. In brief, $25$–$50 \times 10^3$ YFP-tagged MDA-MB-231 cells, preconditioned for 6 days at either 3 cP or 0.77 cP (naive cells), were suspended in ice-cold PBS and injected intravenously into the CAM vasculature. Tumour cell extravasation was assessed by intravital confocal imaging 5 h after cell injection. Then, 15 min before intravital imaging, lectin-649 was injected into the CAM vasculature for visualization of blood vessels. At least 16 animals were used for each condition for 3 independent experiments. All of the procedures were approved by University of Alberta Institutional Animal Care and Use Committee (IACUC).

**Tail-vein injection into mice.** The protocol followed for intravenous injection of MDA-MB-231 cells was previously described[45]. In brief, female NOD-SCID Gamma (NSG) mice (aged 5 to 7 weeks) were obtained from the Johns Hopkins animal core facility, and were distributed into groups to obtain a random weight distribution across conditions. Injections were performed in a blinded manner, without knowledge of cell pretreatment conditions or identity of shRNA-mediated modifications. Before injection, the mice were warmed for 5–10 min with an overhead heat lamp to dilate the tail veins. MDA-MB-231 cells ($5 \times 10^5$) suspended in 200 μl of normal saline were injected. To assess the extent of micro- and macro-metastasis in the lungs, mice were euthanized 48 h or 3 weeks after inoculation, respectively. Lungs were inflated with optimal cutting temperature (OCT) compound (Fisher Healthcare) and excised

for both image and qPCR analysis. Animal studies were conducted with all relevant ethical regulations outlined in protocols approved by the Johns Hopkins University Animal Care and Use Committee. For the 3 week lung colonization experiments, mice were observed for any potential humane experimental end-point signs, including respiratory distress, hunched posture, poor grooming and loss of weight according to IACUC approved protocols. No humane end-point signs were observed during any experiment. Animal rooms were maintained at 30–70% relative humidity and a temperature of 18–26 °C with a minimum of 10 room air changes per hour. Cages were changed once a week. Mice were fed a diet containing low fibre (5%), protein (20%) and fat (5–10%).

**Metastatic burden analysis by imaging.** Excised lungs inflated with OCT were formalin-fixed, saturated in 30% sucrose at 4 °C overnight, embedded in OCT compound, flash-frozen in liquid nitrogen and sectioned (15 μm thick) using a cryotome CM1100 (Leica). The sections were mounted onto Superfrost Plus Microscope Slides (Thermo Fisher Scientific), permeabilized with 1% Triton X-100 for 10 min and blocked with 2% BSA for 30 min. The sections were stained for human vimentin (raised in mouse; O91D3; BioLegend; Alexa Fluor 647 conjugated; 677807; 1:200 dilution) for 1.5 h followed by DAPI (1:1,000) incubation for 15 min and mounted for imaging with anti-fade solution (90% glycerol, 20 mM Tris, pH 8.0, and 0.5% $N$-prolyl gallate). Background fluorescence was reduced with 0.1% Sudan Black treatment for 25 min.

Tissue-mounted slides were imaged using a Nikon A1 confocal microscope with a ×10/0.45 NA air objective. Five or ten areas of $1,272 \times 1,272$ μm$^2$ each were randomly selected for samples collected 48 h or 3 weeks after injection, respectively. Images were exported to ImageJ and vimentin-bearing metastatic sites were annotated for manual counting. Micrometastases after 48 h were enumerated as any vimentin-positive single cell, whereas macrometastases after 3 weeks were defined as vimentin-positive clusters containing >5 nuclei.

**Lung colonization analysis by qPCR.** qPCR analysis of hLINE[19,60] was performed in triplicate and the average value per mouse was reported. In brief, two small lobes of the excised lungs were cut and flash-frozen in liquid nitrogen. For DNA extraction, the lung tissue was lysed in lysis buffer (1 M Tris, pH 8.0, 5 M NaCl, 0.5 M EDTA, 10% Tween-20, 10% NP-40 and 40 μg of proteinase K), homogenized and incubated at 55 °C. After 2 h, proteinase K was deactivated at 95 °C for 2 min. DNA was then extracted by phenol–chloroform, precipitated with 2.5 mM NaCl and 100% ethanol (prechilled at −20 °C), and washed with 70% ethanol. DNA samples were resuspended in pure water after air drying. qPCR was performed using the primer pair for hLINE: forward, 5′-TCACTCAAAGCCGCTCAACTAC-3′; reverse, 5′-TCTGCCTTCATTTCGTTATGTACC-3′. A primer pair detecting both mouse and human beta actin sequences in the DNA samples (forward, 5′-ACGAGGCCCAGAGCAAGAGA-3′; reverse, 5′-GCCACACGCAGCTCATTGTAG-3′) was used as baseline to normalize the expression level of human DNA (measured by hLINE) over the total amount of DNA input (measured by mouse and human actin).

**Statistics and reproducibility.** Data represent the mean ± s.d. of ≥3 experiments (reflecting independent biological replicas) with ≥30 cells analysed per condition per experiment, and data points denote values from each cell, unless otherwise specified. Shapiro–Wilk tests were used for normality testing in cases in which the number of data points was between 3 and 8. For samples with >8 data points, the D'Agostino–Pearson omnibus normality test was used to determine whether data were normally distributed. Datasets with Gaussian distributions were compared using two-tailed Student's $t$-tests and one-way ANOVA followed by Tukey's post hoc test. For log normal distribution, statistical comparison was made after logarithmic transformation of the data followed by unpaired two-tailed Student's $t$-tests or one-way ANOVA

followed by Tukey's post hoc test. For non-Gaussian distributions, nonparametric Mann–Whitney $U$-tests were used comparing two conditions, and comparisons for more than two groups were performed using Kruskal–Wallis tests followed by Dunn's multiple-comparison test. Select experiments were analysed using two-way ANOVA followed by Šidák's or Tukey's multiple-comparisons test. Analysis was performed using GraphPad Prism 7, 8 or 9 (GraphPad Software). $P < 0.05$ was considered to be statistically significant; $*P < 0.05$, $**P < 0.01$, $***P < 0.001$, $****P < 0.0001$. The exact sample size, number of replicates, $P$ values and statistical tests performed are provided in Supplementary Information 5–19.

In Fig. 1, images are representative of 3 (Fig. 1d,h,l) or 2 (Fig. 1f,i) independent biological replicas. In Fig. 2, images are representative of 3 (Fig. 2b,g) or 2 (Fig. 2o) independent biological replicas. In Fig. 3, images are representative of ≥3 (Fig. 3a,d) or 2 (Fig. 3g) independent experiments. In Fig. 4, images are representative of 3 (Fig. 4e) or 2 (Fig. 4i) independent experiments.

## Theoretical methods

**Stochastic model of actin network.** A stochastic 2D model of actin-based lamellipodia protrusion was constructed in Python v.3.8 using frameworks and public code established in ref. [23]. As actin filaments at the cell leading edge undergo elongation, branching and capping events at the barbed ends following a Poisson process, the actin network grows against the plasma membrane tension, and exerts a force that determines the protrusion velocity of the network leading edge. The model used previously determined force–velocity relationships recapitulating not only the Brownian ratchet model of polymerization but also other models such as the tethered ratchet or end-tracking motors. While the published model examined the effects of transient changes in membrane tension on leading edge network organization, we incorporated dynamic forces acting on the actin network due to extracellular fluid of a defined viscosity into the current model. We introduced viscous resistive force into the model using a linear-viscous dashpot[61] where viscous loading forces on the leading edge scaled linearly with extracellular viscosity. The total force per unit width at the leading edge can be described as

$$F = F_0 + F_\eta, \tag{1}$$

where, $F_0$ and $F_\eta$ are forces due to membrane tension and the viscous external fluid, respectively. The viscous force at every iteration ($i$) can be written as

$$F_\eta = k \times \eta \times v_{i-1}, \tag{2}$$

where, $k$ (=100,000) is a dimensionless scaling constant, $\eta$ is the extracellular viscosity and $v_{i-1}$ is the edge velocity from the last iteration.

Using viscosity values tested experimentally, the effects of viscous loading in actin network formation was modelled.

## Two-phase model for confined cell migration

**Model description.** A two-phase, actin–water-osmosis-coupled cell migration model[26,62] was used to explain the cell behaviour observed experimentally at different extracellular viscosities. The two phases refer to the F-actin network and the cytosol. G-actin is in the cytosol phase. The model also includes ion diffusion and flux. Active ion fluxes arise from ion pumps on the cell membrane. Without loss of generality, we assume that the ions are electroneutral. The steady-state model is briefly described below.

To model cell migration in confined channels, we use a 1D coordinate system to represent the cell. For a cell of length $L$, we let $x \in [0,L]$ represent the cell domain, which is attached to the moving cell. The cytosol phase, with pressure $p$ and velocity $v_c$, can be considered to be an incompressible fluid and the pressure gradient should be balanced by the friction between the cytosol and the actin network. These two conditions can be written as

$$-\frac{dp}{dx} - \eta\theta_n(v_c - v_n) = 0, \ \frac{dv_c}{dx} = 0, \tag{3}$$

where $\eta$ is the coefficient of interfacial friction between the two phases; $\theta_n$ and $v_n$ are the concentration and velocity of the actin network phase, respectively. At the cell boundary, water flows across the cell membrane due to the chemical potential difference of water between the inside and outside of the cell[7]. If we assume the water flux, $J_{water}$, to be positive in the inward direction, then the boundary condition for the cytosol phase is

$$v_c - v_0 = -J_{water}^f \ \text{at} \ x = L; \ v_c - v_0 = J_{water}^b \ \text{at} \ x = 0, \tag{4}$$

where $v_0$ is the steady-state velocity of the cell. The superscript notes f and b indicate quantities associated with the front and rear (or back) of the cell, respectively. The water flux is calculated from

$$J_{water}^{f(b)} = -\alpha^{f(b)}\left[\left(p^{f(b)} - p_*^{f(b)}\right) - RT\left(c^{f(b)} - c_0^{f(b)}\right)\right], \tag{5}$$

where $\alpha$ is the permeability coefficient of water, $c$ is the concentration of ion, and $RT$ is the ideal gas constant multiplied by the absolute temperature. The subscript 0 indicates the extracellular environment. $p_*$ is the hydraulic pressure experienced by the cell, which is different from the hydraulic pressure at the end of the channel, $p_0$, owing to hydraulic resistance developed in the channel. From the analysis of a generic pipe flow, the pressure experienced by the cell can be expressed as

$$p_*^f = p_0^f + d_g^f(v_0 - J_{water}^f), \ p_*^b = p_0^b - d_g^b(v_0 + J_{water}^b), \tag{6}$$

where $d_g$ is the coefficient of external hydraulic resistance, which depends on the geometry of the channel and viscosity of the extracellular medium.

The actin network phase also has a pressure, $\sigma_n$, arising from actin swelling. Without loss of generality, we can assume that the passive swelling pressure is proportional to the concentration of the actin network, that is, $\sigma_n = k_{\sigma_n}\theta_n$, where $k_{\sigma_n}$ is a constant. The force from focal adhesions provides an effective body force on the actin network. The force balance on the actin network is

$$-\frac{d\sigma_n}{dx} + \eta\theta_n(v_c - v_n) - \eta_{st}\theta_n v_n = 0, \tag{7}$$

where $\eta_{st}$ is the strength of focal adhesions. Actin polymerization typically occurs at the front of the cell, whereas actin depolymerization occurs within the cytoplasm. We assume that $\gamma$ is a constant rate of actin depolymerization. As F-actin and G-actin convert interchangeably, the mass conservations of the two actin species are described by

$$\frac{d}{dx}(v_n\theta_n) = -\gamma\theta_n, \ \frac{d}{dx}(v_c\theta_c) = D_{\theta_c}\frac{d^2\theta_c}{dx^2} + \gamma\theta_n, \tag{8}$$

where $\theta_c$ and $D_{\theta_c}$ are the concentration and diffusion coefficient of G-actin, respectively. At the front end of the cell, the flux boundary condition for F-actin and G-actin are related to the rate of actin polymerization, $J_{actin}$, and the boundary condition for F-actin and G-actin is

$$\theta_n(v_0 - v_n) = J_{actin}, \ \theta_c(v_0 - v_c) = -J_{actin}, \ \text{at} \ x = L, \tag{9}$$

where $J_{actin} = J_{actin}^f \theta_c/(\theta_{c,c} + \theta_c)$ and $J_{actin}^f$ and $\theta_{c,c}$ are constants. As actin depolymerization occurs throughout the cytoplasm, the flux boundary conditions for F-actin and G-actin are zero at the rear of the cell. At the time scale we are interested in, the total amount of F-actin and G-actin remain a constant so that $\int_0^L (\theta_n + \theta_c) dx = L\theta_*$, where $\theta_*$ is the average concentration of actin.

The diffusion–convection equation for ions is

$$\frac{\mathrm{d}}{\mathrm{d}x}(v_c c) = D_c \frac{\mathrm{d}^2 c}{\mathrm{d}x^2}, \tag{10}$$

where $D_c$ is the diffusion coefficient of ions. The flux at the boundary is a combination of passive, $J_{c,\mathrm{p}}$, and active, $J_{c,\mathrm{active}}$, fluxes. The passive flux follows the concentration difference of the ion, that is, $J_{c,\mathrm{p}}^{\mathrm{f(b)}} = -k_{\mathrm{sol}}^{\mathrm{f(b)}}(c^{\mathrm{f(b)}} - c_0^{\mathrm{f(b)}})$, where $k_{\mathrm{sol}}$ is the permeability of the passive ion flux. The negative sign arises from the model convention that all inward fluxes are considered positive. The active ion flux is prescribed in the model.

Frictional forces exist between the cell's lateral membrane and the channel wall. The magnitude of the frictional force can be expressed as $F_f = \xi v_0$, where $\xi$ is a friction coefficient. This force is in the opposite direction of cell migration. By considering all the external forces acting on the cell, the force balance at the cellular level is

$$-(p_0^{\mathrm{f}} - p_0^{\mathrm{b}}) - (d_g^{\mathrm{f}} + d_g^{\mathrm{b}})(v_0 - J_{\mathrm{water}}^{\mathrm{f}}) - \eta_{\mathrm{st}}\int_0^L \theta_n v_n \mathrm{d}x - F_f = 0. \tag{11}$$

The system is solved by considering all of the equations together.

**Model specifications and parameters.** From experiments, we observed a polarized distribution of focal adhesions, myosin II activity (pMLC) and NHE1 at elevated viscosities. The distributions of focal adhesions and pMLC are highly correlated (Extended Data Figs. 4i and 8j), which is consistent with the idea that myosin II activity facilitates the maturation of focal adhesions[63,64]. We therefore assume that the distribution of focal adhesions occurs as a result of the distribution of myosin II, which is not explicitly included in the model. The polarized distribution of NHE1 (Fig. 2c) is also a part of the model input.

The experimental data suggest that the distribution of focal adhesions is non-uniform across the cell length (Extended Data Fig. 4i). For cells at 0.77 cP medium (low viscosity (LV)), the distribution of focal adhesions is high at both ends and low in the middle of the cell. We use a quadratic function

$$\bar{\eta}_{st,\mathrm{LV}} = 4(1 - b_{\mathrm{LV}})(\bar{x} - 1/2)^2 + b_{\mathrm{LV}} \tag{12}$$

to describe the normalized distribution of focal adhesions in low viscosity medium (Extended Data Fig. 5e), where $\bar{x}$ is the normalized coordinate of the cell length, and $b_{\mathrm{LV}} \in [0,1]$ is the minimum value of the profile at $\bar{x} = 1/2$. For cells in 8 cP medium (high viscosity (HV)), the distribution of focal adhesions is high at the leading edge and low at the trailing edge of the cell. Therefore, we use a different function

$$\bar{\eta}_{st,\mathrm{HV}} = (1 - b_{\mathrm{HV}})\bar{x}^4 + b_{\mathrm{HV}} \tag{13}$$

to describe the normalized distribution of focal adhesion for high viscosity medium (Extended Data Fig. 5e), where the minimum value $b_{\mathrm{HV}}$ occurs at $\bar{x} = 0$. Experimental data suggest that $b_{\mathrm{LV}} = 0.3$ at 0.77 cP and $b_{\mathrm{HV}} = 0.25$ at 8 cP (Extended Data Fig. 4i).

The non-normalized local strength of focal adhesions is expressed as $\eta_{st,i} = \eta_{st,i}^0 \bar{\eta}_{st,i}$ where $i \in \{\mathrm{LV},\mathrm{HV}\}$ and $\eta_{st,i}^0$ is the coefficient of focal adhesions, which varies with the extracellular viscosity. To approximate the strength of focal adhesions, we combined the distribution of focal adhesion size and number for all cells in each group (Extended Data Fig. 4j) and divided them by the corresponding number of cells to obtain a normalized focal adhesion strength map along the cell length. From experimental data (Extended Data Fig. 4j), the maximum local focal adhesion area at 8 cP is about 8× larger than that at 0.77 cP. We therefore assume $\eta_{st,\mathrm{HV}}^0 = 8\eta_{st,\mathrm{LV}}^0$ The value of $\eta_{st,\mathrm{LV}}^0$ is estimated from literature[62,65] with the consideration that cells under confinement experience a relatively low density of focal adhesions compared with cells in 2D[66,67].

For confined cell migration in a 1D channel, the coefficient of hydraulic resistance is estimated from a pipe flow[68] as $d_g = 12\mu l_0/w^2$, where $\mu$ is the viscosity of the extracellular medium, $l_0$ is the effective channel length and $w$ is the smallest dimension of the cross-sectional geometry, which is 3.5 µm in this work. While the length of cells varies with viscosity by virtue of their volume alteration, the channel length ($L_0 = 200$ µm) and cross-sectional area (35 µm²) are constants across both viscosities. For this maths model, we assume that the cell occupies the entire cross section of the channel and therefore the cell lengths can be calculated from the measure of average cell volume in confinement (Fig. 2a): for 0.77 cP, the cell length is $L = 85$ µm, and for 8 cP, the cell length is $L = 125$ µm. The effective channel length is calculated by $l_0 = L_0 - L$. On the basis of these parameters, we can then calculate the coefficient of hydraulic resistance for each given viscosity $\mu$. The viscosity of the medium not only increases the hydraulic resistance experienced by a cell, but also increases the friction between the cell and channel. Thus, we use a higher frictional coefficient for 8 cP medium ($\xi_{\mathrm{HV}} = 900$ Pa s µm⁻¹) than 0.77 cP ($\xi_{\mathrm{LV}} = 180$ Pa s µm⁻¹).

The polarized ion fluxes are determined by the distribution of membrane ion channels. We use $J_{c,\mathrm{active}}^{\mathrm{b}}$ and $J_{c,\mathrm{active}}^{\mathrm{f}}$ to represent active ion fluxes at the back (trailing edge) and front (leading edge) of a cell, respectively. For polarized cells, $|J_{c,\mathrm{active}}^{\mathrm{f}}| \neq |J_{c,\mathrm{active}}^{\mathrm{b}}|$ and we let the ratio $\gamma = |J_{c,\mathrm{active}}^{\mathrm{f}}/J_{c,\mathrm{active}}^{\mathrm{b}}|$ be the same as the polarization ratio of NHE1. Thus, for 0.77 cP medium, $\gamma_{\mathrm{LV}} = 1.67$, and for 8 cP medium, $\gamma_{\mathrm{HV}} = 2.84$ (Fig. 2c). In the model, we fix $J_{c,\mathrm{active}}^{\mathrm{b}}$ and let $J_{c,\mathrm{active}}^{\mathrm{f}} = -\gamma J_{c,\mathrm{active}}^{\mathrm{b}}$. When NHE1 is inhibited, $J_{c,\mathrm{active}}^{\mathrm{f}} = J_{c,\mathrm{active}}^{\mathrm{b}} = 0$.

The remaining parameters are identical at both viscosities and are listed in Supplementary Information 4.

### Reporting summary

Further information on research design is available in the Nature Research Reporting Summary linked to this article.

## Data availability

The main data supporting the results of this study are available within the Article, Extended Data Figs. 1–11 and Supplementary Information. RNA-seq data are available at the National Center for Biotechnology Information Gene Expression Omnibus, under accession number GSE203651. Source data are provided with this paper.

## Code availability

The code used for data analysis and stochastic actin network simulation is available at GitHub (https://github.com/KKLabJHU). The code for the two-phase model is available at GitHub (https://github.com/sxslabjhu/).

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

**Acknowledgements** We thank R. Zhao for providing select image analysis methods and helpful discussions; R. Law for the GFP–AHD virus; G. Szep for discussion of stochastic model of actin network; M. Rosen for assistance with tail-vein injections; M. A. Glotzer for the gift of the GFP-AHD plasmid; and J. Orlowski for the gift of the NHE1-GFP plasmid. Zebrafish experiments were supported by the Intramural Research Program of the National Institutes of Health, the National Cancer Institute. Some of the schematics were created using cartoons adapted from Servier Medical Art licensed under a Creative Commons Attribution 3.0 Unported License (https://smart.servier.com). This work was supported in part by R01 CA257647 (to K.K. and D.M.G.), R01 GM134542 (to S.X.S. and K.K.), NSF 2045715 (to Y.L.), R01 AR071976 (to C.-M.F. and J.T.), R01 AR072644 (to C.-M.F. and J.T.) and R01 CA054358 (to A.P.F.), the Spanish Ministry of Science, Education and Universities through grants RTI2018 099718-B-100 (to M.A.V.) and an institutional "Maria de Maeztu" Programme for Units of Excellence in R&D and FEDER funds (to M.A.V.), and postdoctoral fellowships from the Fonds de recherche du Quebec—Nature et technologies and the Natural Sciences and Engineering Research Council of Canada (to A.K.). The opinions, findings and conclusions, or recommendations expressed are those of the authors and do not necessarily reflect the views of any of the funding agencies.

**Author contributions** K.B. conceptualized and designed the study, performed and analysed most of the in vitro experiments, analysed in vivo mouse data, developed stochastic model of actin network (with B.I.) and wrote the manuscript. A.K., P. Mehta, B.I., A.S., S.J.L., Y.Z. and A.B. performed selected in vitro experiments, analysed data and edited the manuscript. I.G. and D.M.G. performed and analysed the in vivo mouse experiments, provided critical input and edited the manuscript. C.D.P. performed and analysed the in vivo zebrafish experiments, provided critical input and edited the manuscript. K.S. and J.D.L. performed and analysed the chick embryo experiments, provided critical input and edited the manuscript. J.T., G.S., C.-M.F. and A.P.F. performed the RNA-seq experiments and the corresponding data analysis, provided critical input and edited the manuscript. S.A.S. and M.A.V. performed patch-clamp experiments and intracellular pH measurements, analysed data, designed select experiments, provided critical input throughout the manuscript and edited the manuscript. P. Mistriotis designed select experiments (with K.B.), supervised the generation of stable cells used in this study, provided critical input and edited the manuscript. Y.L. and S.X.S. developed the two-phase mathematical model, provided critical input and edited the manuscript. K.K. conceptualized, designed and supervised the study, and wrote the manuscript.

**Competing interests** The authors declare no competing interests.

**Additional information**
**Correspondence and requests for materials** should be addressed to Konstantinos Konstantopoulos.

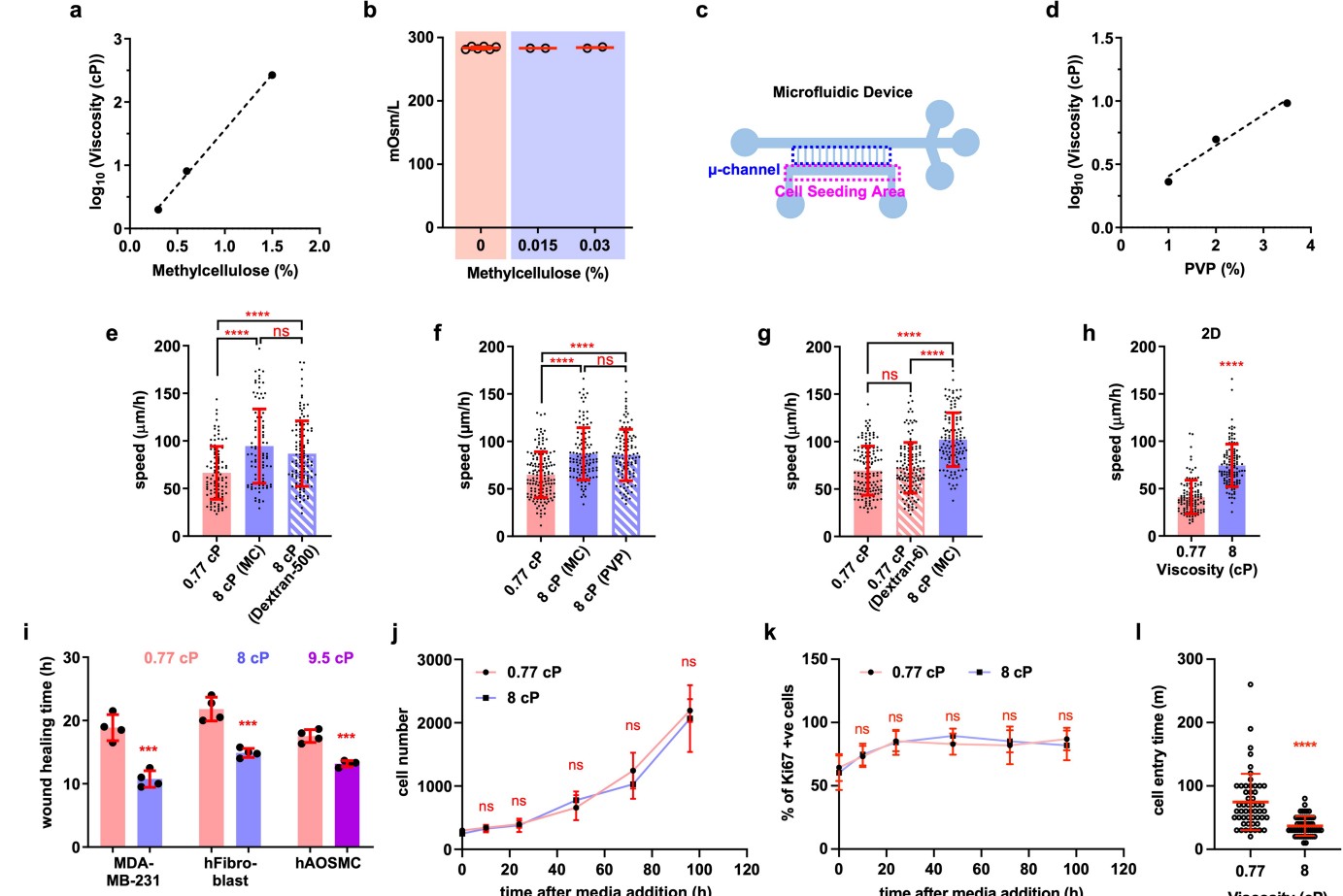

**Extended Data Fig. 1 | Measurement of extracellular fluid viscosity and its effects on cell migration and proliferation. a**, Dependence of medium viscosity on addition of prescribed concentrations of methylcellulose stock solution. **b**, Osmolarity measurements at different concentrations of methylcellulose. Data represent mean ± s.d for 0% methylcellulose or mean ± range for 0.015% and 0.03% methylcellulose. **c**, Schematic of microfluidic device used to assess confined cell migration. **d**, Dependence of medium viscosity on addition of prescribed concentrations of Polyvinylpyrrolidone K 90 (PVP). Data represent mean ± s.d. from ≥3 independent measurements with best fit line in (**a**) and (**d**). s.d. bars are smaller than the size of the symbols in some cases. **e,f,g**, Confined migration speed of MDA-MB-231 cells in media prepared using different macromolecules. Data represent the mean ± s.d. for

$n \geq 101$ cells from ≥2 experiments. **h**, Migration speed of MDA-MB-231 cells on 2D collagen-I-coated surfaces. Data represent the mean ± s.d. for $n \geq 112$ cells from 5 experiments. **i**, Wound healing time of indicated cell lines at 0.77 cP and 8 cP or 9.5 cP. Data represent the mean ± s.d. from 4 experiments. **j**, Total number of MDA-MB-231 cells and **k**, percentage of cells with Ki67 positive nuclei on 2D collagen-I-coated surfaces after addition of media of prescribed viscosities. Data represent the mean ± s.d. for $n > 200$ cells per ROI imaged, for ≥6 ROIs from 2 experiments. **l**, Cell entry time in confining channels at the indicated viscosities. Data represent the mean ± s.d. for $n = 52$ cells from 3 experiments. Tests performed: Kruskal-Wallis followed by Dunn's multiple comparisons (**e–g**), unpaired t-test (**i**), and after log transformation of data (**h,l**), and two-way ANOVA followed by Šidák's multiple comparisons (**j,k**).

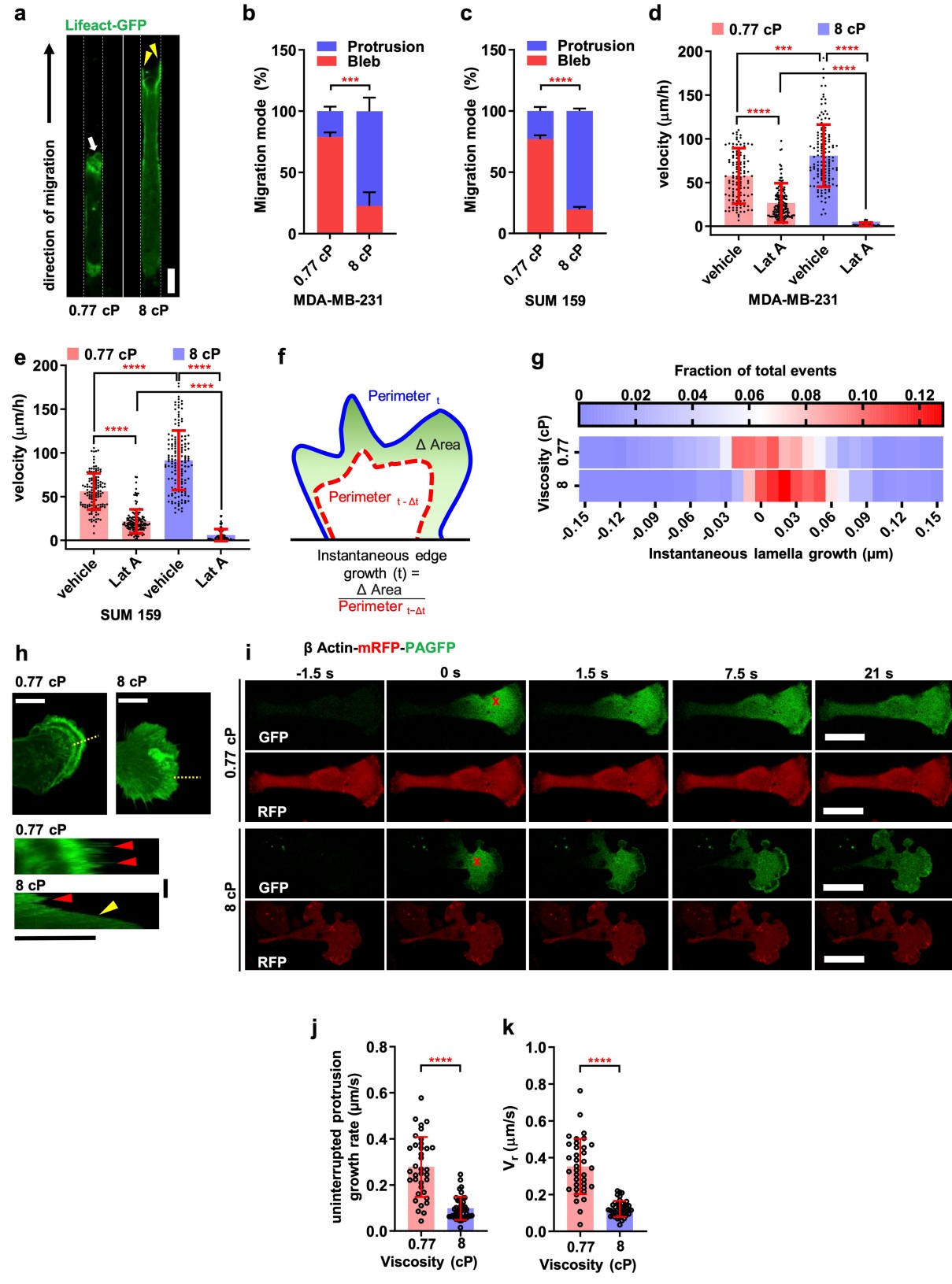

**Extended Data Fig. 2** | See next page for caption.

**Extended Data Fig. 2 | Effects of extracellular viscosity on cell phenotype, migration velocity and actin retrograde flow. a**, Representative confocal images of Lifeact-GFP-tagged MDA-MB-231 cells inside confining channels displaying a predominantly blebbing or protrusive phenotype at 0.77 cP versus 8 cP, respectively. Blebs are indicated by white arrow, while yellow arrowheads point to leading edge protrusions. Scale bar: 5 μm. **b**, Percentage of MDA-MB-231 cells and **c**, SUM159 cells migrating in confining channels with blebbing versus protrusive phenotypes at 0.77 cP and 8 cP. Data represent mean ± s.e.m. for $n \geq 20$ cells per experiment from 4 (**b**) or 6 (**c**) experiments. **d**,**e**, Confined migration velocity of MDA-MB-231 (**d**) and SUM159 (**e**) cells in the presence of vehicle control or LatA (2 μM). Data represent the mean ± s.d. for $n \geq 57$ cells from 3 experiments. **f**, Schematic of image processing strategy used to calculate instantaneous edge growth of Lifeact-GFP-expressing cells on 2D. **g**, Heatmap summarizing relative occurrences of instantaneous leading-edge lamella growth of Lifeact-GFP-expressing MDA-MB-231 cells on 2D substrate. Blue to red colour scale indicates low to high number of fractional occurrences. Negative growth indicates reduction of cell area or retraction, while positive growth indicates forward protrusion. Data summarized from > 450 events in 20 cells per condition imaged over 3 experiments. **h**, (*Top*) Snapshots of confocal micrographs of Lifeact-GFP-expressing MDA-MB-231 cells on 2D at 0.77 cP and 8 cP. The dashed yellow lines are used for kymographs at the bottom. At 0.77 cP, the leading edge extends and contracts rapidly as indicated by the "spikes" in the kymograph (red arrowheads). At 8 cP, the leading edge has slow yet persistent growth (yellow arrowhead) with occasional retraction events (red arrowhead). White bars: 10 μm; for kymographs, black horizontal bar: 5 μm and vertical bar: 30 s. **i**, Representative time-lapse confocal image sequence of PA-GFP fused actin dynamics after it is activated at the interior of cells on 2D. Red "X" symbols indicate points of UV excitation. Scale bar: 25 μm. **j**,**k**, Uninterrupted protrusion growth rate (**j**) and retrograde actin flow rate (**k**) in β-actin-mRFP-PAGFP-expressing MDA-MB-231 cells on 2D at 0.77 and 8 cP. Data are mean ± s.d. for $n \geq 36$ cells from 3 experiments. Tests performed: Two-way ANOVA followed by Šidák's multiple comparisons (**b**,**c**), Kruskal-Wallis followed by Dunn's multiple comparisons (**d**,**e**), Mann Whitney test (**j**), and unpaired t-test (**k**). Images are representative of 3 (**a**,**h**,**i**) experiments.

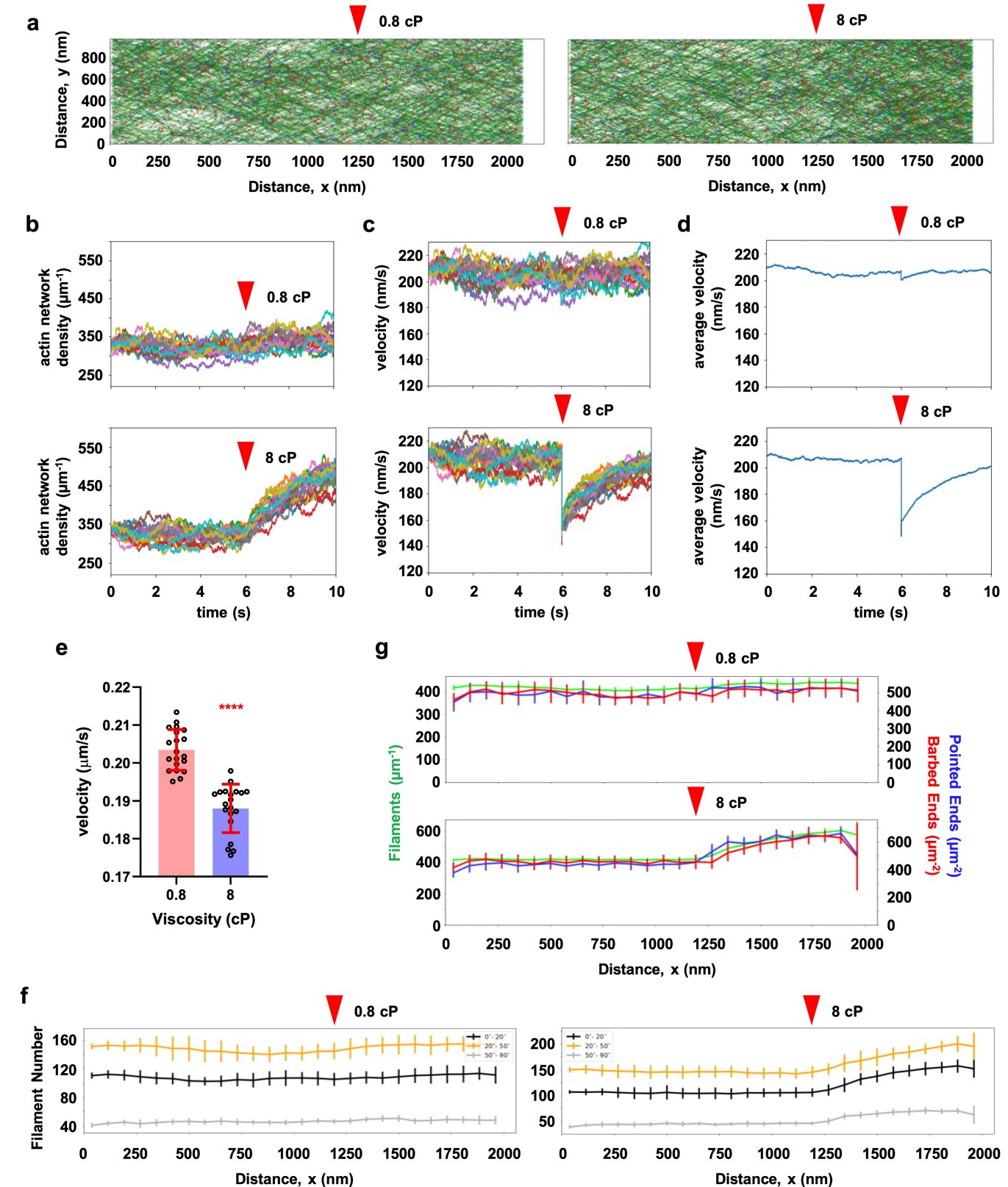

**Extended Data Fig. 3** | See next page for caption.

**Extended Data Fig. 3 | Stochastic model predictions on actin network architecture. a**, Representative visualization of stochastic model results with actin filaments shown in green, barbed ends as red dots, and pointed ends as blue dots. To reach steady state, the simulation was allowed to run for 6 s during which the network grew only against membrane tension from 0 to -1250 nm as indicated by the red arrowheads. Subsequently, the network grew against the membrane tension plus the indicated viscous forces (0.8 or 8 cP). **b**, Actin network density over time of individual simulation runs. **c**, Growth velocity of the actin network front over time. Each line (**b**,**c**) represents individual simulation runs. **d**, Growth velocity of the actin network front over time, averaged from 20 individual simulation runs per condition in (**c**). **e**, Time-averaged actin network front growth velocity over 4 s following the application of viscous forces. **f**, Temporal variation in density of filaments in three different angle bins. Zero degrees represent filaments perpendicular to network edge while 90 degrees are parallel to the edge. Red arrowhead indicates time instant of viscous force application. The filament density at each angle bins is flat at 0.8 cP after turning on the viscous forces. At 8 cP, there is an increase in filament density across all the bins indicating a denser actin network at the cell edge facing the viscous resistance. **g**, Actin filament, capped barbed end, and pointed end density quantified at two different viscous forces. After application of the 8 cP viscous force, there is a sharp rise of pointed ends and increase in filament density, presumably resulting from increased ARP2/3-mediated nucleation. In **e**, **f** and **g**, data are mean ± s.d. from 20 runs per condition. Unpaired t-test was used in **e**.

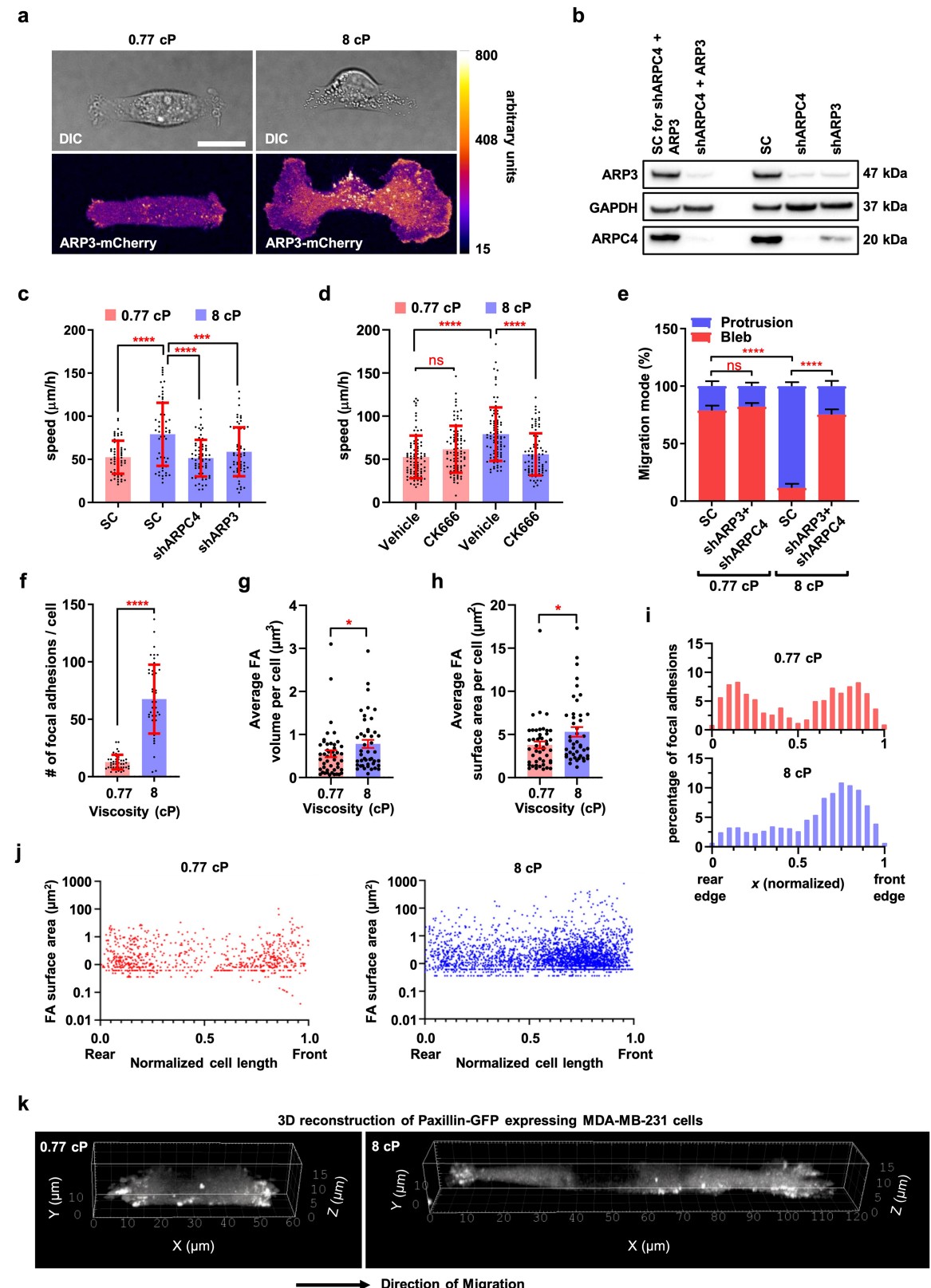

**Extended Data Fig. 4** | See next page for caption.

**Extended Data Fig. 4 | The spatial localization and role of ARP2/3 in cell phenotype and migration as well as the distribution of focal adhesions in confinement at 0.77 and 8 cP. a**, Confocal images of ARP3-mCherry-tagged cells on 2D. Lower panel shows ARP3 intensity as a heatmap. Scale bar: 20 μm. **b**, Representative western blot from cells transduced with shRNA sequences to ARP3 and/or ARPC4 or a noncoding scramble control sequence. For gel source data, see Supplementary Fig. 1. **c,d**, Confined migration speeds of SC, shARPC4 or shARP3 cells (**c**) or wild-type cells under CK666 or vehicle control treatment (**d**). Data are mean ± s.d. for $n = 62$ cells (**c**) or $n = 88$ cells (**d**) from 2 experiments. **e**, Percentage of SC, dual ARP3/ARPC4-KD cells with blebbing versus protrusive phenotypes inside confining channels. Data are mean ± s.e.m. for $n \geq 20$ cells per experiment from 3 experiments. **f**, Number of focal adhesions per cell migrating in confinement, as quantified from cells expressing paxillin-GFP.

Data are mean ± s.d. **g,h**, Average focal adhesion volume (**g**) or surface area (**h**) per cell migrating in confinement at different viscosities. Data are mean ± s.e.m. **i**, Percentage of focal adhesions spatially distributed along normalized cell length for cells migrating in confinement at prescribed viscosities. Data in (**f–i**) are for $n \geq 44$ cells per condition from 3 experiments. **j**, Spatial distribution of focal adhesion areas in cells migrating in confinement at 0.77 cP and 8 cP for 50 and 44 cells, respectively. **k**, 3D reconstructed confocal images of paxillin-GFP in cells migrating in confinement at 0.77 or 8 cP. Tests performed: One-way ANOVA followed by Tukey's multiple comparisons (**c**), Kruskal-Wallis followed by Dunn's multiple comparisons (**d**), two-way ANOVA followed by Tukey's multiple comparisons (**e**), Mann Whitney test (**f**), and unpaired t-test on log transformed data (**g,h**). Images are representative of 2 (**a**) or 3 (**k**) biological replicas. Cell model: MDA-MB-231.

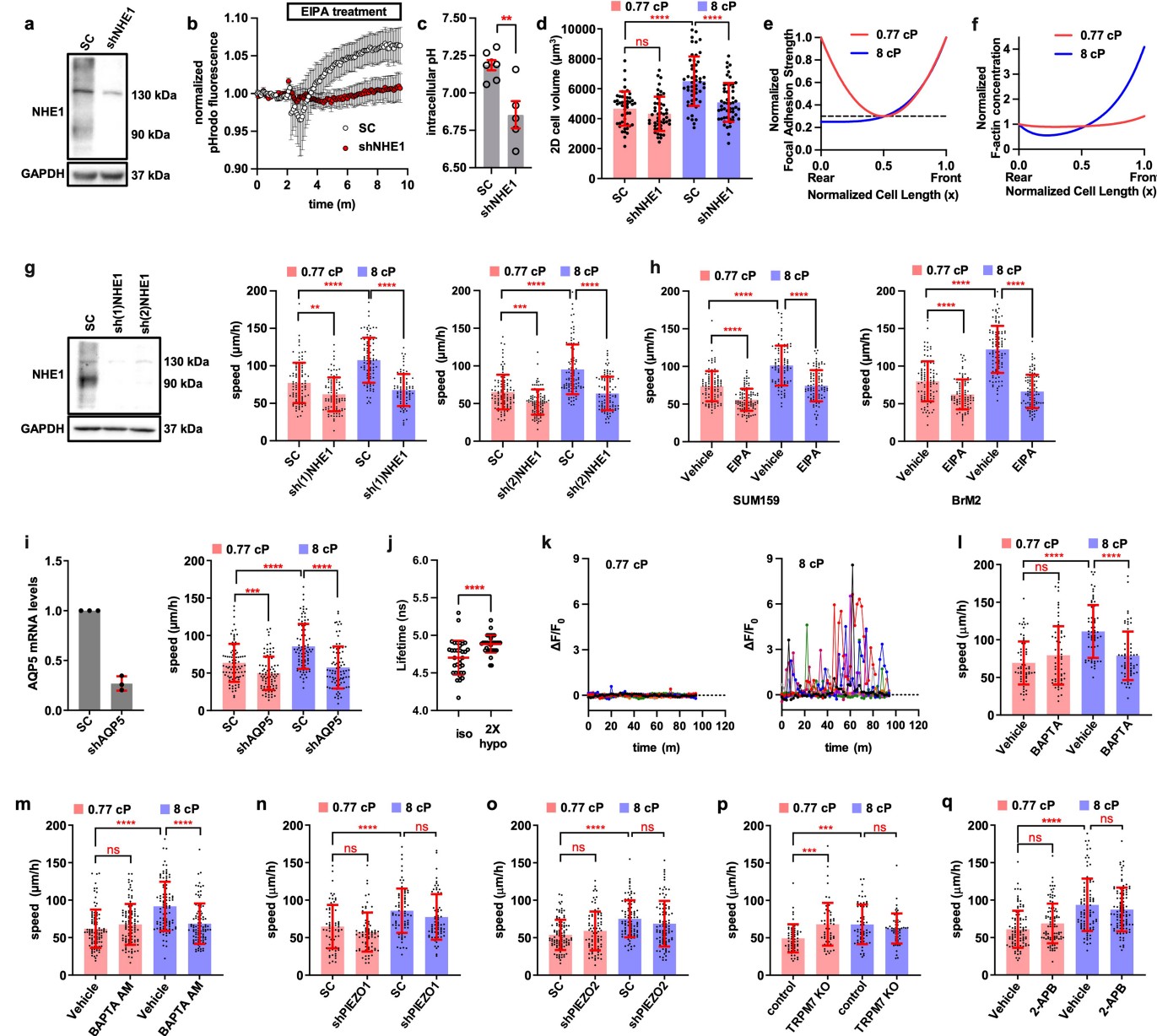

**Extended Data Fig. 5 | The roles of NHE1, AQP5 and MOSICs in cell mechanotransduction in response to extracellular fluid viscosity.**
**a**, Representative western blot of SC and shNHE1 cells. Image is representative of 3 independent protein isolations. **b**, Normalized changes in pHrodo fluorescence in response to EIPA in shControl and shNHE1 cells. EIPA causes acidification in SC cells (increased pHrodo fluorescence) but not in shNHE1 cells, consistent with an already reduced proton extrusion capacity of the shNHE1 cells. Data are mean ± s.e.m. for n≥3 cells from 3 experiments. **c**, Intracellular pH values (mean ± s.e.m.) measured in cells loaded with pHrodo and calibrated using solutions with different pH, verifying diminished proton extrusion in shNHE1 cells. Data represent n≥5 cells from 3 experiments. **d**, Cell volume of Lifeact-GFP-tagged SC and shNHE1 cells on 2D at 0.77 cP and 8 cP from n≥48 cells from 5 experiments. **e**, Spatial distribution of focal adhesions as an input to the two-phase model. At 8 cP, the distribution is higher at the leading than the trailing edge, as observed experimentally. **f**, Two-phase model predictions on the spatial distribution of F-actin. The total F-actin redistributes at 8 cP with higher network density at the cell leading edge. **g**, (*Left*) Representative western blot of SC and two distinct shRNA sequences (sh1 and sh2) against NHE1. Image is representative of 3 independent protein isolations. (*Right*) Confined migration speed of cells expressing SC, sh(1) or sh(2)NHE1 at the prescribed viscosities. Data are mean ± s.d. for n≥85 cells from 2 experiments.

**h**, Confined migration speed of SUM159 and BrM2 cells following vehicle or EIPA treatment at prescribed viscosities. Data are mean ± s.d. for n≥98 cells from 2 experiments. **i**, (*Left*) Relative AQP5 mRNA levels in SC and shAQP5 cells. Data are normalized to the levels of SC cells. Data are mean ± s.d. from 3 experiments. (*Right*) Confined migration speeds of SC and shAQP5 cells at prescribed extracellular viscosities. Data are mean ± s.d. for n≥89 cells from 2 independent experiments. **j**, Quantification of membrane tension in wild-type cells in response to isotonic (1X) or hypotonic (2X) media at 0.77 cP. Data are mean ± s.d. for n≥35 cells from 2 experiments. **k**, GCaMP6s activity after exposure to 0.77 or 8 cP. **l,m**, Confined migration speeds at prescribed viscosities in response to chelation of extracellular calcium via BAPTA (**l**) or the cell permeant calcium chelator BAPTA-AM (**m**). **n,o**, Confined migration speeds of SC and shPiezo1 (**n**) or shPiezo2 (**o**) brain metastatic MDA-MB-231 cells (BrM2) at prescribed viscosities. **p,q**, Confined migration speeds of control and TRPM7-KO cells (**p**) or wild-type cells under 2-APB or vehicle control treatment (**q**) at 0.77 and 8 cP. Data in **l-q** are mean ± s.d. for n≥60 cells from 2 experiments. Tests performed: unpaired t-test (**c,j**), one-way ANOVA followed by Tukey's multiple comparisons (**d**) and after log transformation of data (**i,l–o,q**), Kruskal-Wallis followed by Dunn's multiple comparison (**g,h,p**). For gel source data, see Supplementary Fig. 1. Cell model: MDA-MB-231 unless otherwise indicated.

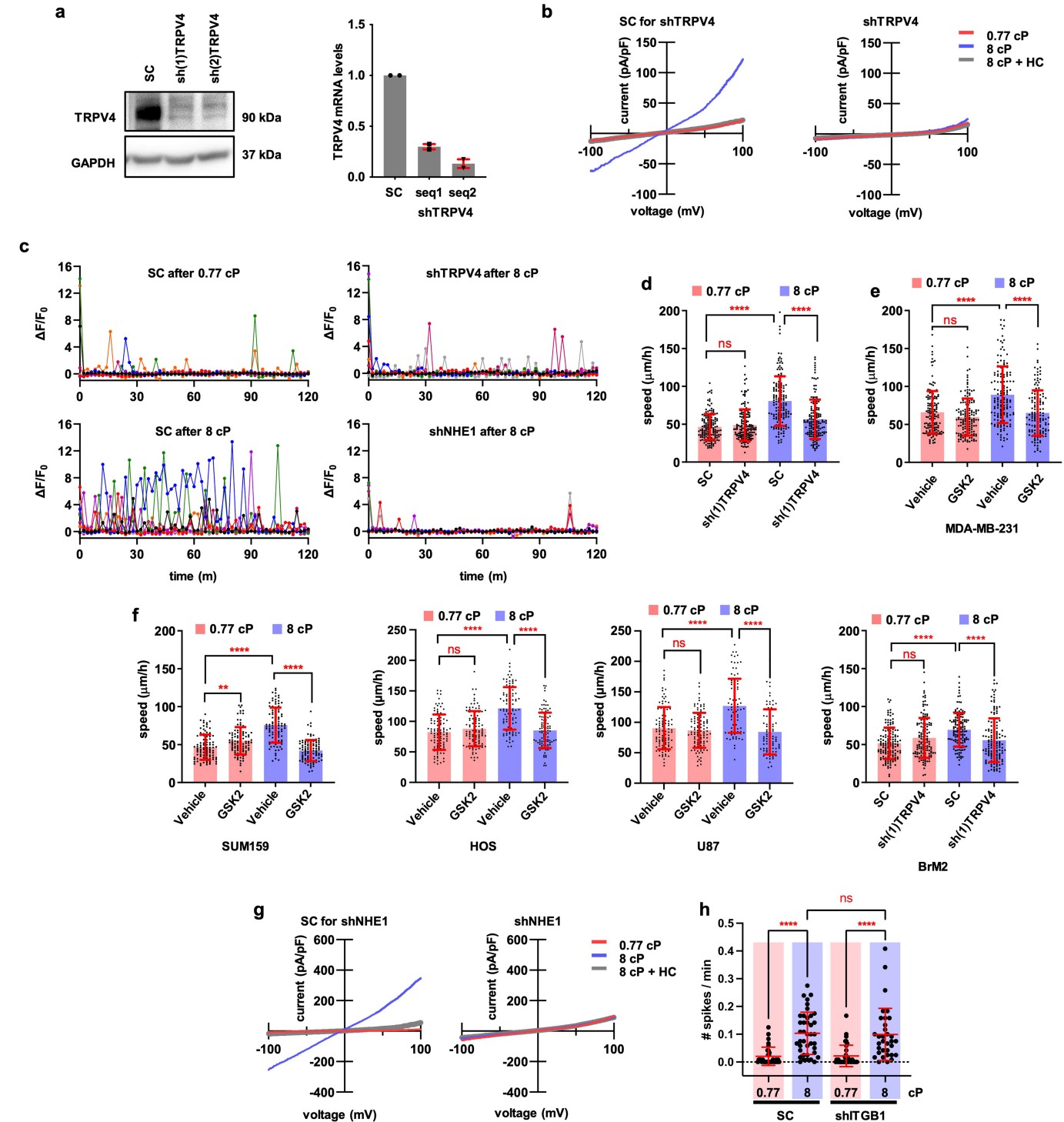

**Extended Data Fig. 6 | The role of TRPV4 in cell mechanotransduction in response to extracellular fluid viscosity. a**, (*Left*) Representative western blot of SC and shTRPV4 cells using two shRNA sequences. Image is representative of 2 independent biological replicas. (*Right*) Relative TRPV4 mRNA levels of SC and shTRPV4 cells. Data are normalized to the levels of SC cells, and represent the mean ± range from 2 experiments. **b**, Representative whole-cell TRPV4 current traces in SC (*left*) and shTRPV4 (*right*) cells exposed to 0.77 cP (pink) or 8 cP in the absence (blue) or presence (grey) of the TRPV4 inhibitor HC-067047 (HC). **c**, GCaMP6s activity in SC, shTRPV4 or shNHE1 cells at 0.77 or 8 cP. **d,e**, Confined migration speeds of SC and shTRPV4 (sequence 1) cells (**d**), or wild-type cells under GSK 2193874 (GSK2) or vehicle control treatment (**e**) at prescribed extracellular viscosities. Data are mean ± s.d. for *n*≥140 cells from 3 experiments. **f**, Effect of TRPV4 inhibition via GSK 2193874 (GSK2) in SUM159,

HOS and U87 cells or TRPV4 knockdown (sequence 1) in brain metastatic MDA-MB-231 cells (BrM2) on confined migration speeds at the prescribed viscosities. Data are mean ± s.d. for *n*≥83 cells from ≥2 experiments. **g**, Representative whole-cell TRPV4 current traces in SC (*left*) and shNHE1 (*right*) cells exposed to 0.77 cP (pink) or 8 cP in the absence (blue) or presence (grey) of the TRPV4 inhibitor HC-067047 (HC). **h**, GCaMP6s activity in SC or shRNA β1-integrin (shITGB1) cells at the prescribed viscosities. Data are mean ± s.d. for *n*≥32 cells from 2 experiments. Tests performed: one-way ANOVA followed by Tukey's multiple comparisons (**f** (only HOS)) and after log transformation of data (**d,e**), Kruskal-Wallis followed by Dunn's multiple comparison (**f** (except HOS), **h**). For gel source data, see Supplementary Fig. 1. Cell model: MDA-MB-231 unless otherwise indicated.

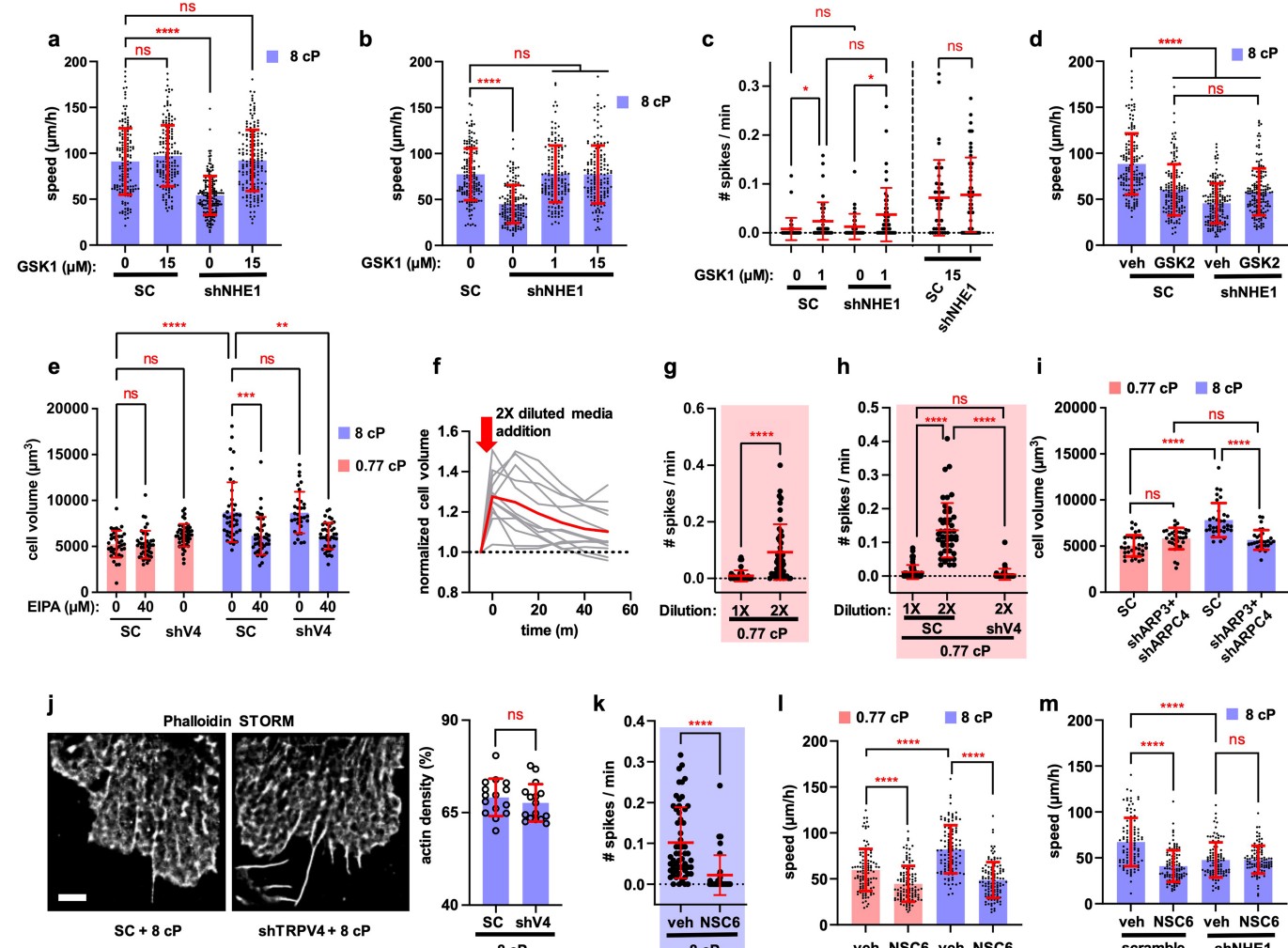

**Extended Data Fig. 7 | The roles of TRPV4 and ezrin in cell migration and calcium influx at elevated viscosities under isotonic and hypotonic conditions. a**, **b**, Confined migration speeds of SC and shNHE1 cells at 8 cP in the presence of vehicle control or indicated doses of the TRPV4 agonist GSK1016790A (GSK1). **c**, GCaMP6s activity in SC and shNHE1 cells on 2D at 0.77 cP in the presence of vehicle control or indicated doses of the TRPV4 agonist GSK1016790A (GSK1). **d**, The effect of the TRPV4 antagonist GSK 2193874 (GSK2) on the confined migration speeds of SC and shNHE1 cells at 8 cP. Data in (**a**–**d**) are mean ± s.d. for n≥32 cells from 3 experiments. **e**, Cell volume of Lifeact-GFP-expressing SC or shTRPV4 cells on 2D at 0.77 cP and 8 cP in the presence of vehicle control or EIPA. Data are mean ± s.d. for n≥32 cells from 2 independent experiments. **f**, Time-dependent volumes of Lifeact-GFP-tagged cells at 0.77 cP after addition of 2X hypotonic medium, which occurred between t = −5 min and t = 0 min. Data represent individual cells (grey) or the population average (red) at each time point. Volume of each cell is normalized to its corresponding value just before addition of hypotonic medium. Data were obtained from n = 12 cells from 2 experiments. **g**,**h**, GCaMP6s activity in

wild-type cells (**g**) or SC and shTRPV4 cells (**h**) on 2D at 0.77 cP in response to isotonic (1X) or hypotonic (2X) media. Data are mean ± s.d. for n≥38 cells from 3 experiments. **i**, Cell volume of Lifeact-GFP-expressing SC or dual ARP3/ARPC4 knockdown cells on 2D at 0.77 cP and 8 cP. Data are mean ± s.d. for n≥29 cells from 2 experiments. **j**, STORM reconstruction and density quantification of F-actin for n≥15 cells on substrates from 2 experiments. Scale bar: 2 μm. **k**, GCaMP6s activity in wild-type cells at 8 cP under ezrin inhibitor NSC668394 (NSC6) or vehicle control treatment. Data are mean ± s.d. for n≥50 cells from 3 experiments. **l**, **m**, The effect of ezrin inhibitor NSC668394 (NSC6) on confined migration speeds of wild-type cells (**l**) or SC and shNHE1 cells (**m**) at 8 cP. Data are mean ± s.d. for n≥107 cells from 2 (**l**) or 3 (**m**) experiments. Tests performed: Kruskal-Wallis followed by Dunn's multiple comparisons (**a**,**b**,**c** (for the 4 groups left of the dashed line), **d**,**e**,**h**,**i**,**m**), Mann Whitney (**c** (pair to the right of the dashed line), **g**,**k**), unpaired t-test (**j**), and one-way ANOVA followed by Tukey's multiple comparisons on log transformed data (**l**). Cell model: MDA-MB-231.

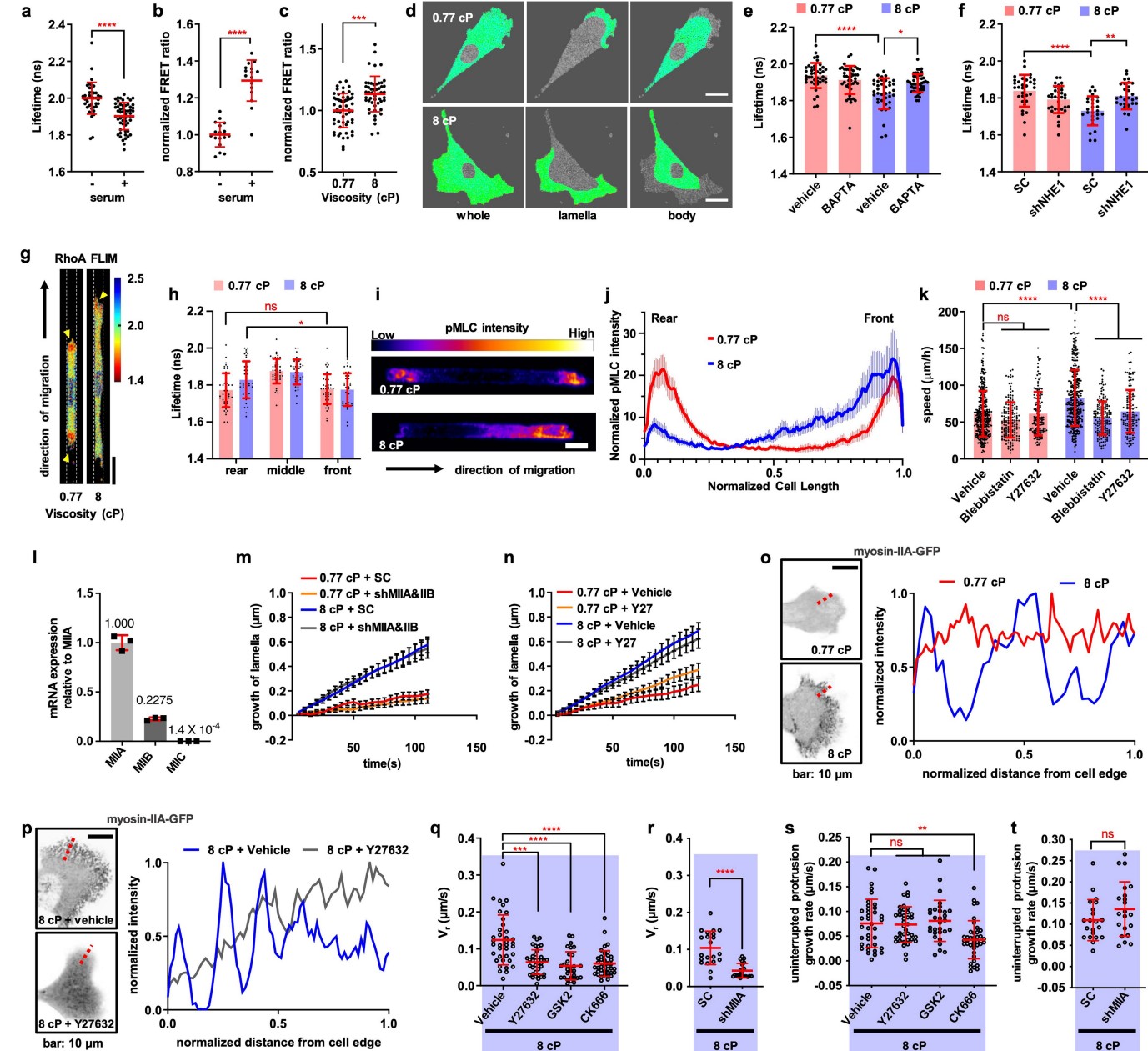

**Extended Data Fig. 8** | See next page for caption.

**Extended Data Fig. 8 | The effects of extracellular fluid viscosity on RHOA activity, and its contribution to myosin II-, ARP2/3- and TRPV4-regulated actin dynamics. a-c,** Lifetimes (**a**) or epi-fluorescent FRET ratios (**b,c**) of the RHOA activity biosensor in cells on 2D in serum-free (**a,b**) or serum-containing (10% FBS; **a-c**) media at the indicated viscosities. Data are mean ± s.d. for $n \geq 16$ cells from ≥2 experiments. **d,** Segmentation strategy of cells on 2D for confocal FLIM-FRET analysis. The entire cell without the nucleus is defined as "whole", while the "lamella" at cell edge is identified from DIC images. The "body" of the cell comprises of the remainder of the cell without the lamella and the nucleus. Scale bars: 20 µm. **e, f,** RHOA activity in wild-type cells in the presence of vehicle control or BAPTA (**e**) or in SC and shNHE1 cells (**f**) on 2D at prescribed viscosities. Data are mean ± s.d. for $n \geq 27$ cells from 3 experiments. **g,** RHOA lifetimes in confined MDA-MB-231 cells. Yellow arrowheads indicate regions of high activity. **h,** Spatial distribution of RHOA activity in different segments of confined MDA-MB-231 cells ($n \geq 38$) from 5 experiments. Data are mean ± s.d. **i,** Confocal images of cells inside confining channels immunostained for pMLC with intensity shown as a heatmap. Scale bar: 5 µm. **j,** Line-scan of pMLC intensity along normalized length of each cell migrating in confinement. Intensity is normalized to the lowest intensity along the scan. Data are moving average ± s.e.m. of $n = 14$ or 16 cells at 0.77 or 8 cP, respectively, pooled from 2 experiments. **k,** Confined migration speeds at prescribed viscosities in the presence of vehicle control, blebbistatin or Y27632. Data are mean ± s.d. for $n \geq 139$ cells from ≥3 experiments. **l,** Relative MIIA, MIIB and MIIC mRNA levels in wild-type cells. Data are normalized to the average levels of MIIA, and represent the mean ± s.d. from 3 experiments. **m,n,** Leading-edge lamella growth of Lifeact-GFP-tagged cells following dual MIIA/MIIB knockdown (**m**) or Y27632 treatment (**n**) on 2D relative to appropriate controls. Data are moving averages ± s.e.m. at each time point of $n \geq 20$ cells per condition from 3 experiments. *$P < 0.05$ for t ≥ 45 s between 0.77 versus 8 cP for SC (**m**) or vehicle control (**n**) samples. $P > 0.05$ for all time points between 0.77 and 8 cP for SC versus dual shMIIA/MIIB (**m**) and vehicle versus Y27632 (**n**). **o,p,** (*Left*) Confocal images showing the spatial localization of MIIA-GFP in cells migrating on 2D at 0.77 and 8 cP (**o**) or 8 cP only in the presence of vehicle control or Y27632 (**p**). (*Right*) Intensity profile along dashed red line in corresponding images. Intensity was normalized to the highest intensity along the scan. Scale bars: 10 µm. **q–t,** Retrograde actin flow rate (**q**) and uninterrupted protrusion growth rate (**s**) in wild-type cells expressing β-actin-mRFP-PAGFP on 2D at 8 cP in the presence of vehicle control, Y27632, GSK 2193874 (GSK2) or CK666. Similar measurements were made for SC and shMIIA cells (**r,t**). Data are mean ± s.d. for $n \geq 21$ cells from ≥2 experiments. Tests performed: Mann Whitney test (**a,b,r**), unpaired t-test (**c**) and after log transformation (**t**), Kruskal-Wallis followed by Dunn's multiple comparisons (**e,k,q,s**), one-way ANOVA followed by Tukey's multiple comparisons (**f**), one-way ANOVA followed by Tukey's between segments in each group (**h**), and two-way ANOVA followed by Tukey's multiple comparisons (**m,n**). Images in **i,o,p** are representative of 2 and in **g** of 5 biological replicas. Cell model: MDA-MB-231.

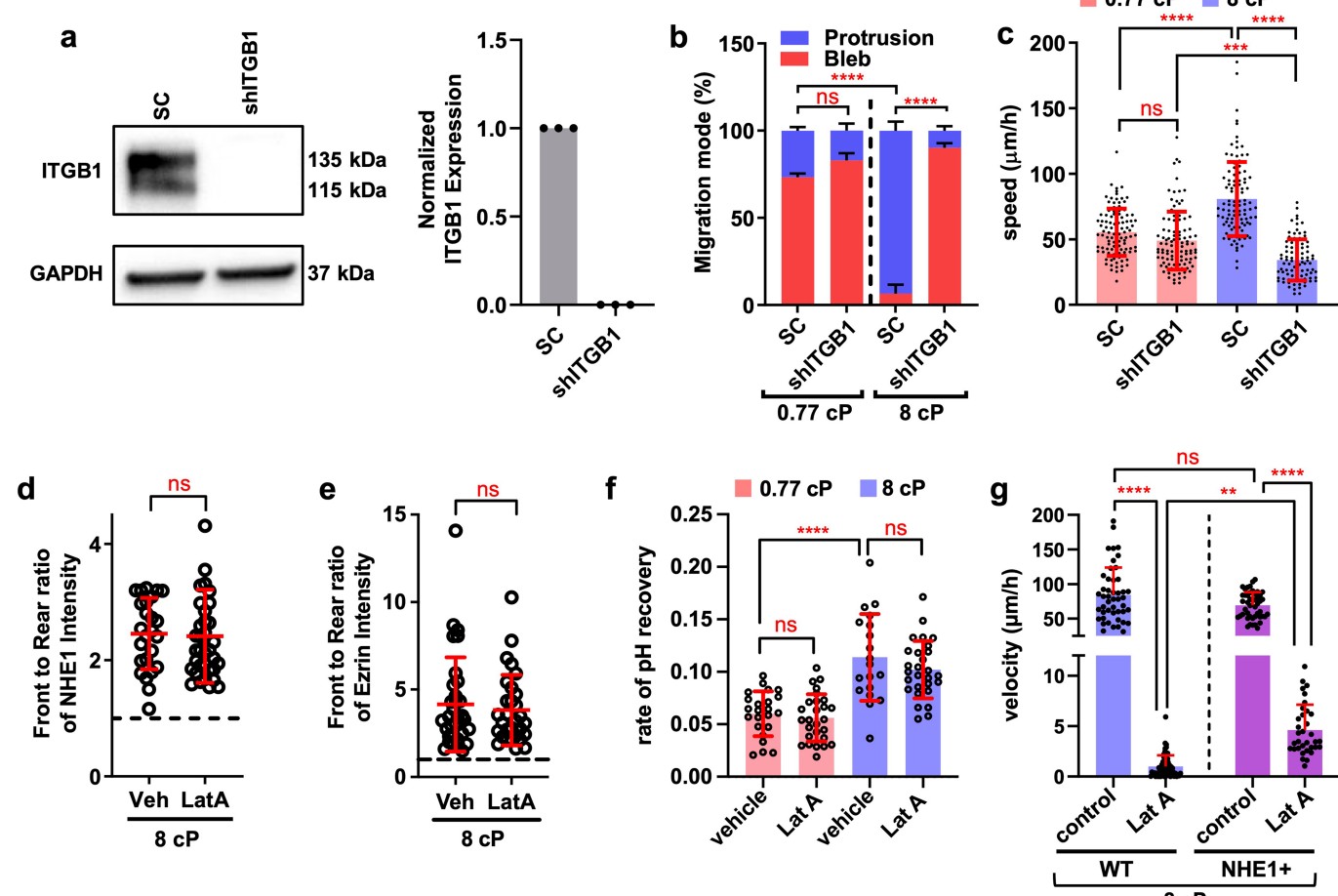

**Extended Data Fig. 9 | The role of β1-integrin in cell phenotype and migration at elevated viscosities as well as the effects of LatA on the spatial distribution of NHE1 and ezrin. a**, Representative western blot image of SC and shITGB1 (β1-integrin) cells (*left*) and their quantification (*right*) from 3 independent biological replicas. For gel source data, see Supplementary Fig. 1. **b**, Percentage of SC and shITGB1 cells migrating with blebbing versus protrusive phenotypes at 0.77 cP and 8 cP. Data are mean ± s.e.m. for $n{\geq}20$ cells per experiment from 3 independent experiments. **c**, Confined migration speeds of SC and shITGB1 cells at prescribed extracellular viscosities. Data are mean ± s.d. for $n{\geq}89$ cells from 3 experiments. **d**, **e**, Front to rear NHE1 (**d**) or ezrin (**e**) intensity ratio in confined cells migrating at 8 cP in the presence of

vehicle control or LatA (2 μM). Data are mean ± s.d. for $n{\geq}25$ cells from 2 experiments. **f**, Rate of pH recovery after intracellular acidosis due to $NH_4Cl$ pulse treatment of cells expressing pHRed. Data are mean ± s.d. from $n{\geq}19$ cells in each condition pooled from 3 experiments. **g**, Confined migration velocity of wild-type or NHE1-GFP-overexpressing (NHE1+) MDA-MB-231 cells ($n{\geq}35$) at 8 cP in the presence of vehicle control or Lat A from ≥3 experiments. The y axis is discontinued from 17–25 μm/h to highlight differences in velocity. Data are mean ± s.d. Tests performed: two-way ANOVA followed by Tukey's multiple comparisons (**b**), Kruskal-Wallis followed by Dunn's (**c**,**g**), unpaired t-test on log transformed data (**d**,**e**), and one-way ANOVA followed by Tukey's (**f**). Cell model: MDA-MB-231.

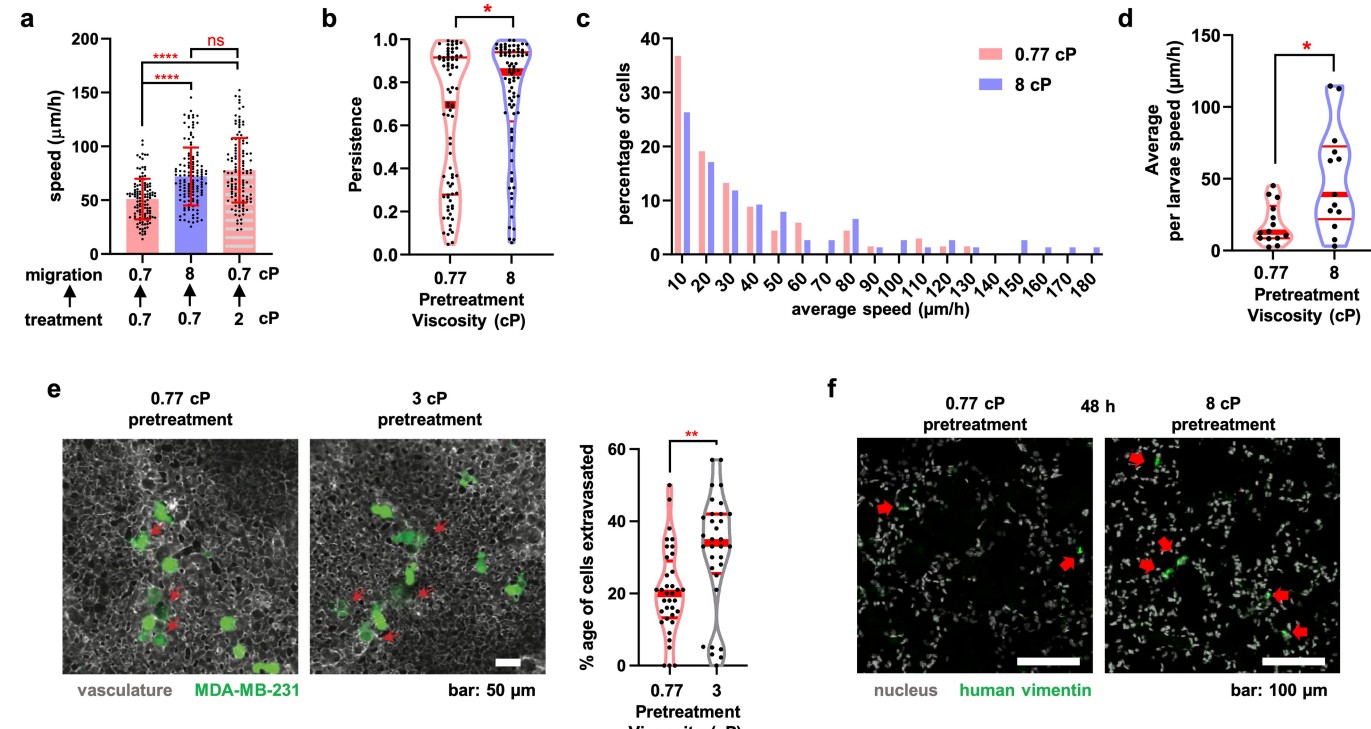

**Extended Data Fig. 10 | The effect of pre-conditioning at different viscosities on cell migration *in vitro* and *in vivo*. a**, Confined migration speeds of cells, pre-treated at 0.77 cP or 2 cP for 6 days, and resuspended in the prescribed "migration" viscosity in which motility was tracked. Data are mean ± s.d. for *n* = 131 cells from 3 experiments. **b**, Persistence values of cells moving through zebrafish ISVs. Red lines indicate median (thick) and quartiles (thin). **c**, Histogram of average cell speeds through zebrafish ISVs. Values were calculated for *n* = 77 and *n* = 81 cells pre-treated at 0.77 cP and 8 cP, respectively, from 3 experiments. **d**, Average speeds on a per larvae basis of cells moving through zebrafish ISVs. Values were calculated across 14 or 13 larvae injected with cells pre-conditioned at 0.77 or 8 cP, respectively, pooled from 3 experiments. Red lines indicate median (thick) and quartiles (thin). **e**, (*Left*)

Representative maximum intensity projections of confocal images showing CAM vasculature with extravasating cells. Cells were cultured for 6 days either at 0.77 or 3 cP, re-suspended in ice-cold PBS, and injected into the CAM vasculature. Red arrows indicate extravasated cells. Scale bar: 50 μm. (*Right*) Quantification of cell extravasation. Red lines indicate median (thick) and quartiles (thin) for ≥16 animals from 3 experiments. **f**, Confocal images of mouse lung sections 48 h post-injection showing human vimentin-positive metastatic colonies (red arrows). Scale bars: 100 μm. Tests performed: Kruskal-Wallis followed by Dunn's multiple comparisons (**a**), Mann Whitney (**b**), unpaired t-test (**e**), and unpaired t-test with Welch's correction (**d**). Images are representative of 3 (**e**) or 2 (**f**) independent biological replicas. Cell model: MDA-MB-231.

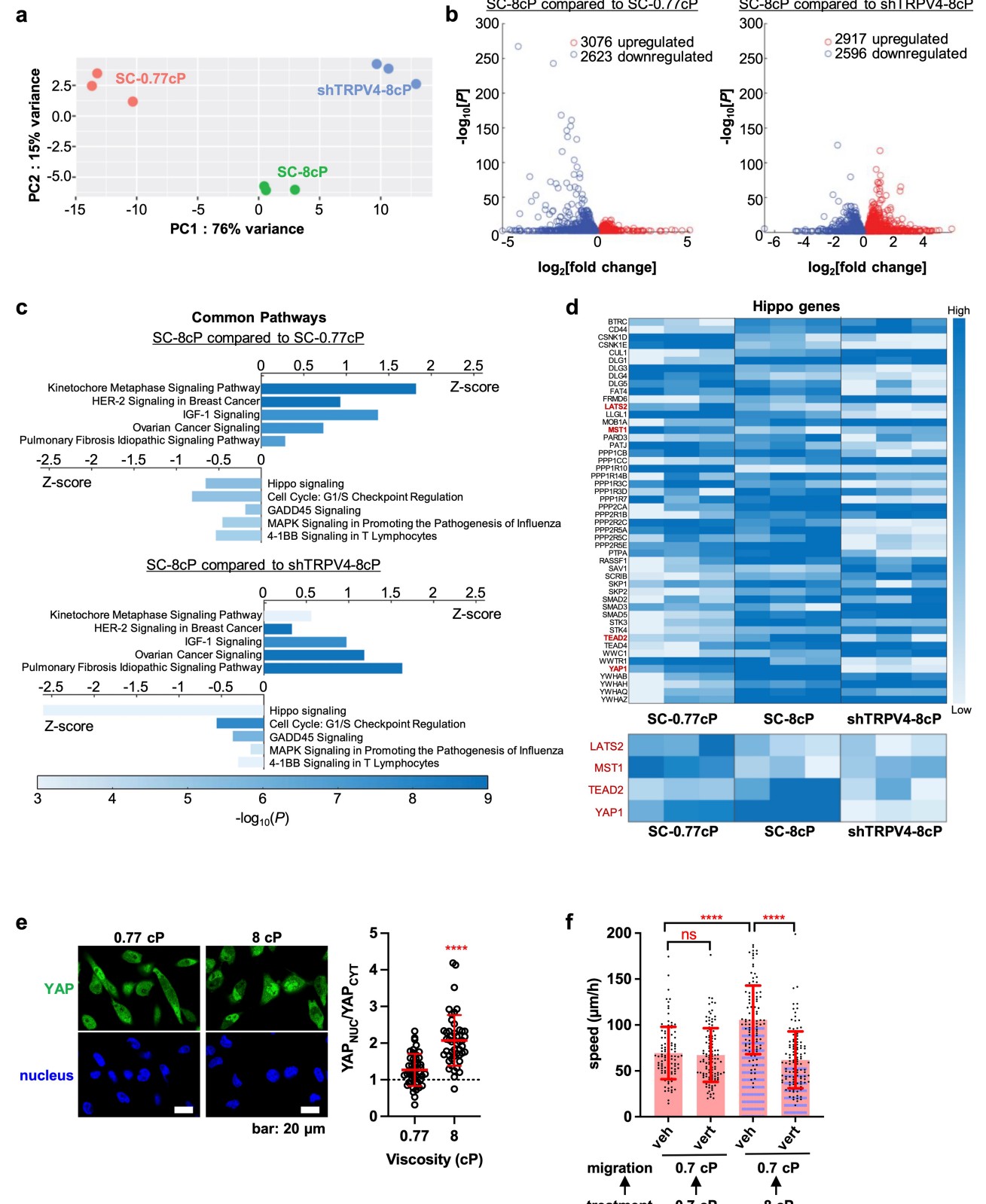

**Extended Data Fig. 11** | See next page for caption.

**Extended Data Fig. 11 | TRPV4 mediates mechanical response to fluid viscosity via transcriptional control of the Hippo pathway. a**, PCA plot of SC cells exposed to 0.77 cP (SC-0.77cP) or 8 cP (SC-8cP) and shTRPV4 cells subjected to 8 cP (shTRPV4-8cP) from 3 independent biological replicates. **b**, Volcano plots displaying DEGs with $P \leq 0.05$ in SC cells at 0.77 versus 8 cP, and between SC and shTRPV4 cells at 8 cP. Downregulated genes are in blue and upregulated in red. **c**, Ingenuity pathway analyses of paired samples shown in (**b**). Top 5 commonly upregulated and downregulated pathways are shown. **d**, Heatmap showing the relative expression levels of Hippo pathway genes identified in (**c**). **e**, (*Left*) Confocal images of cells on collagen-I-coated glass bottom dishes after 4 h of exposure to medium of prescribed viscosities and immunostained for YAP1 and Hoechst. Scale bars: 20 μm. (*Right*) Quantification of nuclear-to-cytosolic YAP1 ratio. Data are mean ± s.d. for $n \geq 46$ cells from 2 experiments. **f**, Confined migration speeds of cells, pre-conditioned at 0.77 or 8 cP in the presence of verteporfin or vehicle control for 6 days, and resuspended in the indicated "migration" viscosity without verteporfin in which their motility was tracked. Data are mean ± s.d. for $n \geq 107$ cells from 2 experiments. Tests performed: Mann Whitney (**e**) and Kruskal-Wallis followed by Dunn's multiple comparisons (**f**). Images are representative of 2 independent biological replicas (**e**). Cell model: MDA-MB-231.

# Reporting Summary

## Statistics

For all statistical analyses, confirm that the following items are present in the figure legend, table legend, main text, or Methods section.

| n/a | Confirmed | |
|---|---|---|
| ☐ | ☒ | The exact sample size ($n$) for each experimental group/condition, given as a discrete number and unit of measurement |
| ☐ | ☒ | A statement on whether measurements were taken from distinct samples or whether the same sample was measured repeatedly |
| ☐ | ☒ | The statistical test(s) used AND whether they are one- or two-sided<br>*Only common tests should be described solely by name; describe more complex techniques in the Methods section.* |
| ☐ | ☒ | A description of all covariates tested |
| ☐ | ☒ | A description of any assumptions or corrections, such as tests of normality and adjustment for multiple comparisons |
| ☐ | ☒ | A full description of the statistical parameters including central tendency (e.g. means) or other basic estimates (e.g. regression coefficient) AND variation (e.g. standard deviation) or associated estimates of uncertainty (e.g. confidence intervals) |
| ☐ | ☒ | For null hypothesis testing, the test statistic (e.g. $F$, $t$, $r$) with confidence intervals, effect sizes, degrees of freedom and $P$ value noted<br>*Give P values as exact values whenever suitable.* |
| ☒ | ☐ | For Bayesian analysis, information on the choice of priors and Markov chain Monte Carlo settings |
| ☒ | ☐ | For hierarchical and complex designs, identification of the appropriate level for tests and full reporting of outcomes |
| ☒ | ☐ | Estimates of effect sizes (e.g. Cohen's $d$, Pearson's $r$), indicating how they were calculated |

*Our web collection on statistics for biologists contains articles on many of the points above.*

## Software and code

Policy information about availability of computer code

| Data collection | Data were collected using standard features of NIS-Elements (Version 5.02.01 or Version 3.22.00), ImageJ (1.53f51), Axon p-Clamp (Version 10), Hamamatsu HCImage (v1.0, 2018), ZEN 2.3 SP1 FP3 (black), SymPhoTime 64 (Version 2.4) or ZEN 2.3 (blue edition). |
|---|---|
| Data analysis | Data were organized primarily using Microsoft Excel (Version 16.57) and Graphpad Prism (Version 7,8 or 9). Select analysis were performed in MATLAB (Version R2021b, 9.11.0.1809720), SigmaPlot (Version 10), NIS-Elements (Version 5.02.01), Picasso (Version 0.3.8), SymPhoTime 64 (Version 2.4), Imaris (Version 9.7.0), HISAT2 (Version 2.2.1), HTSEQ (Version 0.13.5) or DEseq2 (Version 1.30.1, using R 4.0 version), as described in Methods. The human GRCh38.p13 genome (hg38) was obtained from Ensembl (release 104). The code used for data analysis and stochastic actin network simulation is available on GitHub in a link provided by KK: https://github.com/KKLabJHU. The code for the two-phase model is available on GitHub in a link provided by SXS: https://github.com/sxslabjhu/ |

For manuscripts utilizing custom algorithms or software that are central to the research but not yet described in published literature, software must be made available to editors and reviewers. We strongly encourage code deposition in a community repository (e.g. GitHub). See the Nature Portfolio guidelines for submitting code & software for further information.

## Data

Policy information about availability of data

All manuscripts must include a data availability statement. This statement should provide the following information, where applicable:
- Accession codes, unique identifiers, or web links for publicly available datasets
- A description of any restrictions on data availability
- For clinical datasets or third party data, please ensure that the statement adheres to our policy

The main data supporting the results of this study are available within the paper and its Extended Data Figure file. All source data are provided with this paper.

# Field-specific reporting

Please select the one below that is the best fit for your research. If you are not sure, read the appropriate sections before making your selection.

☒ Life sciences          ☐ Behavioural & social sciences          ☐ Ecological, evolutionary & environmental sciences

For a reference copy of the document with all sections, see nature.com/documents/nr-reporting-summary-flat.pdf

# Life sciences study design

All studies must disclose on these points even when the disclosure is negative.

| | |
|---|---|
| Sample size | We indicated in supplementary information the exact sample size, number of replicates and p value for each experiment. No predetermination of sample size was done. Sample size was chosen based on the throughput of the technique used. Sample sizes were sufficient to show the same trends between the replicates performed for each experiment, and by statistical testing. For animal experiments, no statistical methods were used to calculate sample size and group size, and the sample size was determined based on experience of similar assays performed earlier. In each experiment multiple cells or animals were examined in parallel leading to sample sizes primarily of the order of tens to hundreds. |
| Data exclusions | Data were excluded for in vitro migration experiments based on pre-established criteria also mentioned in the Methods section: dividing or apoptotic cells were excluded from analysis. For mice experiments, animals in which the entire volume of cancer cell suspension was not successfully injected into the mouse tail vein were immediately excluded from the study. For zebrafish experiments, fish lacking human cells, as identified from microscopic screening ~1-4h post injection, were excluded from imaging. During analysis of cell trajectories in intersegmental vessels, only cells with a diameter of at least 10 μm were tracked to avoid cell fragments. Speeds and persistences were calculated only for cells remaining in the intersegmental vessels for at least 4 frames. |
| Replication | The exact number of replicates for each experiment has been indicated in the figure legends and supplementary information. Most experiments were repeated 3 or more times, with similar results observed each time. Select control experiments were repeated 2 times with consistent data across all replicates. |
| Randomization | For zebrafish experiments, fish from a given clutch were randomly divided into experimental groups prior to injection. For CAM extravasation assays, embryos were randomly divided into experimental groups prior to injection. Mice were randomized based on weight to maintain similar average weight across experimental groups. For all other experiments, cells were randomly distributed into experimental groups before imaging and analysis. |
| Blinding | RNA sequencing and data analysis and mice injections were performed in a blinded manner, without prior knowledge of cell pre-treatment conditions or identity of shRNA-mediated modifications. For remaining experiments researchers were not blinded as data collection and analysis were performed by the same individual assigning the groups. Wherever possible findings (e.g., cell migration speeds, cell volume) were analyzed in an unbiased manner by the use of automated Fiji and custom analysis codes in Matlab. |

# Reporting for specific materials, systems and methods

We require information from authors about some types of materials, experimental systems and methods used in many studies. Here, indicate whether each material, system or method listed is relevant to your study. If you are not sure if a list item applies to your research, read the appropriate section before selecting a response.

## Materials & experimental systems

| n/a | Involved in the study |
|---|---|
| ☐ | ☒ Antibodies |
| ☐ | ☒ Eukaryotic cell lines |
| ☒ | ☐ Palaeontology and archaeology |
| ☐ | ☒ Animals and other organisms |
| ☒ | ☐ Human research participants |
| ☒ | ☐ Clinical data |
| ☒ | ☐ Dual use research of concern |

## Methods

| n/a | Involved in the study |
|---|---|
| ☒ | ☐ ChIP-seq |
| ☐ | ☒ Flow cytometry |
| ☒ | ☐ MRI-based neuroimaging |

## Antibodies

| | |
|---|---|
| Antibodies used | Primary antibodies used for immunostaining were: anti-pMLC (Ser19) antibody (raised in rabbit; Cell Signaling; 3671; Lot# 6; 1:100 dilution); anti-Ki-67 antibody (raised in mouse; clone 8D5; Cell Signaling; 9449; Lot# 4; 1:800 dilution); anti-NHE1 antibody (raised in mouse; clone 54; Santa Cruz Biotechnology; sc-136239; Lot# H2021; 1:50 dilution); anti-Ezrin antibody (raised in rabbit; Cell Signaling; 3145S; Lot# 5; 1:200 dilution); anti-human Vimentin (raised in mouse; clone O91D3; BioLegend; Alexa Fluor 647 conjugated; 677807; Lot# B309436; 1:200 dilution); anti-YAP antibody (raised in mouse; clone 63.7; Santa Cruz Biotechnology; |

sc-101199; Lot# G2821; 1:50 dilution). Secondary antibodies used for immunostaining were: Alexa Fluor 488 goat anti-rabbit immunoglobulin G (IgG) H+L, (Invitrogen; A11034; Lot# 2256692; 1:200 dilution), Alexa Fluor 488 goat anti-mouse immunoglobulin G (IgG) H+L, (Invitrogen; A11029; Lot# 2179204; 1:200 dilution) or Alexa Fluor Plus 647 goat anti-mouse immunoglobulin G (IgG) H+L, (Invitrogen; A32728; Lot# WE322197; 1:100 dilution).
Primary antibodies used for western blotting were: anti-TRPV4 antibody (raised in mouse; clone 1B2.6; Millipore Sigma; MABS466; Lot# 3462069; 1:1000 dilution), anti-ARP3 (raised in mouse; FMS338; Abcam; ab49671; Lot# GR234096-12; 1:5000 dilution), anti-ARPC4 (raised in rabbit; Abcam; ab217065; Lot# GR312814-8; 1:2000 dilution), anti-Integrin beta-1 Antibody (raised in rabbit; Cell Signaling; 4706S; Lot#6; 1:1000 dilution) and anti-NHE1 (raised in mouse; clone 54; Santa Cruz Biotechnology; sc-136239; Lot# H2021; 1:200 dilution). GAPDH was used as a loading control (raised in rabbit; 14C10; Cell Signaling; 2118S; Lot# 14; 1:5000 dilution). Secondary antibodies used for western blotting were: anti-mouse IgG, HRP-linked antibody (Cell Signaling; 7076S; Lot# 36; 1:1000 dilution) and anti-rabbit IgG, HRP-linked antibody (Cell Signaling, 7074S; Lot# 29; 1:1000 dilution).

| Validation | Prior to purchasing, antibodies were validated by the manufacturer and available on their website:<br>https://media.cellsignal.com/coa/3671/6/3671-lot-6-coa.pdf<br>https://media.cellsignal.com/coa/9449/14/9449-lot-14-coa.pdf<br>https://datasheets.scbt.com/sc-136239.pdf<br>https://media.cellsignal.com/coa/3145/5/3145-lot-5-coa.pdf<br>https://www.biolegend.com/nl-nl/certificate-of-analysis (lot# B309436)<br>https://datasheets.scbt.com/sc-101199.pdf<br>https://www.merckmillipore.com/IN/en/product/Anti-Trpv4-Antibody-clone-1B2.6,MM_NF-MABS466#anchor_COA (lot# 3462069)<br>https://www.abcam.com/arp3-antibody-fms338-ab49671.pdf<br>https://www.abcam.com/arpc4-antibody-ab217065.pdf<br>https://media.cellsignal.com/coa/4706/6/4706-lot-6-coa.pdf<br>https://media.cellsignal.com/coa/2118/14/2118-lot-14-coa.pdf<br>Additionally, antibodies for western blotting were verified based on the appropriate molecular weight of the protein probed and for immunofluorescence by comparison of their cellular distribution to that provided in the manufacturer's datasheet. |

# Eukaryotic cell lines

Policy information about cell lines

| Cell line source(s) | MDA-MB-231, Human osteosarcoma cells, U87 and HEK cells were purchased from American Type Culture Collection (ATCC). Human dermal fibroblasts (GM05565) were purchased from the Coriell Institute. Human AOSMC was purchased from PromoCell. SUM159 cells were a gift from Denis Wirtz (Johns Hopkins University), brain metastatic MDA-MB-231 (BrM2) cells were from Joan Massagué (Memorial Sloan Kettering Cancer Center). Select cell lines were modified from the parental line, as described in Methods (e.g., development of cell lines with scramble, shRNA or live reporters). |
| Authentication | Cell lines were originally authenticated by ATCC, Coriell Institute or PromoCell, and were not further authenticated as part of this study. |
| Mycoplasma contamination | All cell lines were regularly tested by PCR and verified to be free of mycoplasma contamination. |
| Commonly misidentified lines (See ICLAC register) | No commonly misidentified cell lines were used. |

# Animals and other organisms

Policy information about studies involving animals; ARRIVE guidelines recommended for reporting animal research

| Laboratory animals | For zebrafish studies, fish larvae were obtained from established transgenic Tg(mpx:EGFP/flk:mCherry) stock lines housed at the NHGRI zebrafish core. Zebrafish were injected at 3 dpf and imaged between 3-4 dpf. Gonad differentiation to determine sex had not been completed by the time points used in these experiments. Fertilized White Leghorn chicken eggs were obtained from the University of Alberta Poultry Research Centre. For tail vein injection, five-to-seven-week-old female NOD-SCID Gamma (NSG) mice weighing 15-25 g were obtained from Johns Hopkins animal core facility. Animal rooms were maintained at 30-70% relative humidity and a temperature of 18-26ºC with a minimum of 10 room air changes per hour. Cages were changed once a week. Mice were fed a diet containing low fiber (5%), protein (20%) and fat (5-10%). |
| Wild animals | This study did not involve wild animals. |
| Field-collected samples | This study did not involve samples collected from the field. |
| Ethics oversight | Mice studies were conducted with all relevant ethical regulations outlined in protocols approved by the Johns Hopkins University Animal Care and Use Committee. Zebrafish studies were conducted under protocols approved by the National Cancer Institute and the National Institutes of Health Animal Care and Use Committee. Chick embryo studies were performed by following procedures approved by the University of Alberta Institutional Animal Care and Use Committee. |

Note that full information on the approval of the study protocol must also be provided in the manuscript.

# Flow Cytometry

## Plots

Confirm that:

☐ The axis labels state the marker and fluorochrome used (e.g. CD4-FITC).

☐ The axis scales are clearly visible. Include numbers along axes only for bottom left plot of group (a 'group' is an analysis of identical markers).

☐ All plots are contour plots with outliers or pseudocolor plots.

☐ A numerical value for number of cells or percentage (with statistics) is provided.

## Methodology

| | |
|---|---|
| Sample preparation | All cell suspensions were filtered through a 40 μm cell strainer to remove cell aggregates and clumps; no staining was performed. |
| Instrument | Sony SH800 was used for cell sorting (with 100 μm sorting chip, 488 and 561 nm laser lines). |
| Software | Data were collected with Sony Cell Sorter Software (Version 2.1.5). |
| Cell population abundance | Cell population abundance: Arp3-pmCherryC1, beta-actin-mRFP-PAGFP or NHE1-GFP cells were collected. For sorting the beta-actin-mRFP-PAGFP cells, the red channel was used. |
| Gating strategy | Doublets and debris were rejected based on FSC and SSC characteristics. |

☐ Tick this box to confirm that a figure exemplifying the gating strategy is provided in the Supplementary Information.

