## [Peer Review File · Nature]

Manuscript Title: Extracellular fluid viscosity enhances cell migration and cancer dissemination

Redactions – unpublished data

Reviewer Comments & Author Rebuttals

Reviewer Reports on the Initial Version:

Referees' comments:

Referee #1 (Remarks to the Author):

In this study by Bera et al., the authors build on previous work from the lab (Stroka et al., 2014, Zhao et al., 2019, Yankaskas et al., 2019, Zhao et al., 2021) and describe extracellular viscosity as a new mechanical cue, which appears essential in increasing cell migration abilities and how this could influence cancer progression.

More specifically, the authors aim at unravelling three important aspects:

- How does external viscosity lead to altered cell mechanics and phenotype and change in migration abilities ?
- What is the molecular pathway leading to increased migration in response to increased viscosity ?
- How does this influence cancer progression and metastasis ?

The authors rely on a microfluidic based confinement device associated with medium viscosity tuning. They conclude that external viscosity alters cortical actin leading to an opening of Na⁺/H⁺ transporter NHE1 and aquaporin AQ5 leading to an increase in cell volume. This cell swelling induces an activation of the mechanosensitive TRPV4 channel and Ca²⁺ entry leading to the activation of the RhoA/actomyosin contraction pathway. Further more, they show that this viscosity-induced mechanical changes are retained suggesting mechanical memory conservation. Later relying on in vivo experimental metastasis models, they suggest that cancer cells preconditioned in more viscous media are more metastatic than their normal counterparts.

While the work is overall well performed with large amounts of data, the manuscript would benefit from several improvements to strengthen the conclusions and convey a better message.

I would like to start my report by commenting on some claims in the abstract of this study, and thereby identify some limitations of the study. While I overall can't agree more that looking at the contribution of external viscosity to cell behavior and fate is important, I wonder why the authors decided to focus on viscosity changes that are likely to occur in interstitial fluids and have not considered other fluids such as blood and lymph whose contribution, with regards to cancer progression, is important. Second, the vast majority of the work, which is carefully documented, focuses on very particular structures, which are protrusive regions of the cell, i.e. leading edge in the abstract. It is unclear if these effects would apply to cells that are not protrusive, which is likely to happen in various situations such as tumor masses subjected to interstitial fluids. Third, while the latter effects look at almost-instantaneous effect of viscosity on cell motility, the last part

of the abstract and thus manuscript aims at linking such phenotypes to a relevant pathological situation, metastasis. However, to do so, the authors coin a different approach, i.e. memory of viscosity, which, in my opinion, cannot be a simple extrapolation of changes that occur instantaneously on migrating cells. Although very interesting, phenotypes of cells that are exposed to long-term viscosity are likely to be very different and include additional transcriptional regulation for example. Apart from TRPV4, the authors have not looked at whether effects described in Fig.1-3 (and associated supplemental material) are at play in this situation of viscosity memory. Moreover, it remains unclear why the authors have decided to focus on i) intravascular migration of cells in that context, whose relevance to metastasis needs to be demonstrated, and on ii) experimental metastasis in mice that is likely to involve many other mechanisms than increased cell motility. Assessing tumor invasion/intravasation (or other cell motility phenotypes) in realistic in vivo situations (intravital imaging) could have been a more suitable option to directly link the in vitro description of the effect of viscosity on cell motility.

Along this line, I have identified additional and more specific pitfalls :

- While the mechanoresponsive pathway starting from NHE1/AQP5 opening and leading to RhoA-mediated contractility and improved migration abilities is elegantly demonstrated with strong data supporting the claim, the transmission or clutch conveying the external viscous mechanical cue into an internal cortical cell response is far less clear and strong data to support the claims are missing. In particular, the cytoskeleton reorganization data in the first part of the manuscript are sometimes confusing and it is unclear how actin response to extracellular pressure induces the subsequent mechanoresponse. Overall, I feel this important point relies more on data extrapolation and should be addressed.

- How external viscosity is sensed by cells, within protrusive structures, and conveyed through "mechanical loading" to actin remodeling needs to be identified.

- How similar is the immediate response to long-term "memory" viscosity needs to be documented carefully as it is likely that mechanisms differ.

- Most of the in vitro experiments model invasion-like process and suggest the importance of altered mechanics in response to increase in viscosity in a switch of migration phenotype, which is typically seen during primary tumor invasion. After cancer cell bloodborne dissemination, extravasation is followed by tumor cell colonization of niches which are often perivascular (see the large amount of data on brain, lungs, liver or bone metastasis), which questions the relevance of the in vivo experimental metastasis model chosen. Correlating the extensive amount of data obtained in Fig.1-3 would require a better adapted in vivo evaluation (such as intravital imaging of tumor invasion for example)

Below are the specific comments on the manuscript:

Introduction

L44 : It should be mentioned that migration is also important in developmental process, this even more important in the cancer context where EMT and cell stemness, 2 important developmental pathway are hijacked by cancer cells.

L50 : While the authors mention that viscosity is driven by augmented macromolecules presence, it should be mentioned that this is usually driven by lymphatics drainage failure which is an important marker of cancer progression and altered tumor mechanics (see the work from Rakesh Jain for instance)

L.59 : "how do cells sense the physical cue of elevated, yet physiologically relevant, extracellular viscosity?" This is indeed an important question that would require specific experiments to identify the viscosity sensing mechanism at the plasma membrane.

L.70 : "We also extend our findings to the in vivo setting by showing that breast cancer cells pre-exposed to elevated viscosity exhibit faster motility through intersegmental vessels (ISVs) in

zebrafish, and a higher ability to establish lung metastases in mice after tail vein injection" As discussed earlier (abstract comments) and later, it remains unclear what is the rationale of looking at intravascular migration of long-term viscosity treatments without looking at direct effect on metastasis (extravasation, outgrowth).

Results

- I.97-99 : how universal is such an effect ? Would immune cells be sensitive to it as well since they are the ones exposed massively to interstitial fluids? Is the viscosity-dependent effect on motility dependent on motility types? The different cells were not tested for physiologically relevant viscosities. Intermediate range found in interstitial fluid flow (1-3 cP, as suggested in the introduction) should also be tested to support the relevance of the role of viscosities
 - I.103 : how do the authors make sure that viscosity is tunable in 3D environments ? (spheroids?) What is the link between confinement-based motility and 3D spheroids dissociation ? Is there an effect on migration directionality in the spheroid migration assay ?
 - I.139-140 : Mechanical loading ? Why focusing on the protrusive structures while cells in tumors aren't necessarily protrusive when exposed to IFP ? How long are cells incubated for? Have the authors performed additional transcriptomic analysis (and compared with long-term viscosity experiments) ?
 - Would a gradient in cP act as a mechanotaxis cue ? The microfluidics device could be used to generate such gradients taking advantage of the opposed inlets to assess the potential role of such cue. That would be similar to the molecular gradients found in close to tumors lacking proper lymphatic drainage which have been shown to act as directional cues.
 - I.144 : This reviewer does not have expertise to judge modelisation. However, when the authors claim that "These modeling data further verify that elevated extracellular viscosity is sufficient to alter the actin network at leading-edge" : why only at the leading edge ?
 - The analysis of lamella extension does not account for the increase of cellular volume (which is later clearly demonstrated). How can authors rule out that this increase is not a secondary effect from an overall volume increase ? This should be clarified.
 - In PA-beta-actin GFP dataset, I don't get what is the size of the photoactivation zone. The methods mention a point for activation ? What is the size ? The activated region seems quite big compared to the overall size of cells. A line scan perpendicular to the main polarization axis would be more convincing.
 - Arp3 signal intensities were quantified without any normalizing staining. How can the authors rule out that the increased intensities are due to a real accumulation of Arp2/3 and not an artifactual accumulation due to membrane ruffling leading to overlays of membrane/actin cortex ?
 - The authors have used Paxillin-GFP to monitor the effect of viscosity on FAs. Paxillin is known to be only transiently recruited to FA and absent from early talin enriched FA or late tensin enriched FA. These results should be confirmed with the proper integrin staining controls.

 - I.166 : there are no images to appreciate the effect, is this the case in confined cells ?
 - The analysis of the Arp3/FA phenotypes are lacking a strong conclusion and a transition to the following paragraph might ease the reader to get the rationale.
 - Extended Fig 5d was more relevant to the general model being demonstrated than Fig 2e as the first one concerns cells in confinement and the second cells simply in a 2D culture configuration. I would suggest exchanging these 2 panels.
 - Along the same line, Fig 2g and Extended Fig 5g seem to be the exact same dataset but with the y axis being "speed" and the other "velocity". Unless there is a clear reason for that, which should appear in the captions of the panels, there is no need for data duplication.
- Major comment: At this stage, it remains unclear what links the extracellular viscosity to increased actin remodeling? How specific to migration/protrusive structures is such an effect?
- I.180 : Why looking only at NHE1 ? Why not assessing other channels ?
 - I.203 : Have the authors considered looking at membrane tension ? Activation of stretch-activated channels ?
 - I.211 : channels are restricted to MDA-231 cells, have you considered other channels that would

not be 231-specific ?

- I.219-222 : TRPV4 is the key MIC activated by viscosity – does it apply to other cells types than SUM159 and BrM2 ? What is the effect of TRPV4 depletion on actin remodeling ?
- I.226 : “Taken together, we postulate that elevated viscosity promotes NHE1-dependent cell swelling, which leads to calcium influx through the MIC TRPV4” -again, what is the viscosity sensing mechanism ? Why would TRPV4 only act downstream of NHE1 ? Has NHE1 silencing been tested in other cell lines ?
- I.248 : it is not clear to this reviewer why would cells require to first increase their volume in response to viscosity and then activate membrane channels ?
- L260-263 : the authors postulate from ezrin inhibition that block Ca²⁺ flashes at 8cP and phenocopies siNHE1 that elevated extracellular viscosity leads to the formation of a denser actin network at the leading edge. If true, there should be a polarization of ezrin toward the edge as cell migrate in 8cP medium ? Is this the case ? This should lead the a polarized entry of Ca²⁺ mainly from the leading edge ? Is that the case ? Authors might consider using more subtil Ca²⁺ biosensors such as PKC β -C2 for local Ca²⁺ entry instead of GCAMP which only highlights global Ca²⁺ response,
- In Fig.2n is it epifluorescence or confocal microscopy ? If epifluorescence, how can the authors rule out artifactual accumulation from membrane bending ?
- The authors have shown that NMIIA is required but not NMIIB. What about NMIIC ? This should be tested as NMIIC was shown to (see Surcel et al., PNAS 2015 and Cancer Research 2019). This is particularly relevant in a context of cancer cell invasion and metastasis.
- In Extended Fig 8e, the authors used Latrunculin A, a known actin depolymerizing agent, which halts cell migration without affecting NHE1 polarization. They conclude that the OEM is not sufficient to support cell migration. However, how would the authors explain that NHE1 polarization which relies on a denser actin network at the front is not affected by LatA, while the rest of the actin cytoskeleton required for cell migration is ? This dataset is a major counter-argument against the model proposed : Extracellular Viscosity → denser actin network → Ezrin/NHE polarization → TRPV4 activation This should be clarified : How is Ezrin polarized in such situation ?
- While I understand that a drug mediated approach is tricky given that the upstream and downstream effectors rely on actin, a drug independent approach might be more relevant : for instance using laser ablation on the denser actin network (using a relevant marker such as LifeAct) at the front as the cell starts polarizing.
- Major comment, I.344 : The model is mostly based on migrating cells with a leading edge – would such effect apply to cells that are subjected to viscosity as this happens within a tumor mass (no leading edge, no protrusive phenotype, etc..) -> what would be the trigger then ? What is the viscosity-sensing mechanism ?
- I.358 : cells that are exposed for 6 days to medium with tuned viscosities very likely massively tune their transcriptional program in response to it – have the authors considered such effect ? Have they looked at expression levels of the targets (TRPV4, etc..) ? Plus, the whole concept of memory plays against the instantaneous mechanical loading concept – how can the authors reconcile such an effect? Ideally, the authors should consider recapitulating most of their approaches (actin, etc..) in cells exposed for longer times...
- Major comment : The authors aim at studying the migration of arrested cancer cells within the ISVs as a model of cancer cells migrating in the stroma (around the primary tumor or after extravasation). How could hematogenous dissemination model anything similar to such situation (described in fig.1-3), with shear stress that are several orders or magnitude higher in the vasculature compared to any interstitial fluid? How would the authors discriminate between a migrating cell and a cell arresting by vessel occlusion and deforming as blood flow is pushing the back of the cell. This is quite difficult to judge from still image, but this jiggling is what is observed with occluded cells pushed by high shear stress blood flow.
More specifically
- I.366 : what is the relevance of looking at intravascular migration of cells ? Is this specific to ISVs ? What is the effect of long-term viscosity on cell mechanics ? Have the authors considered subjecting cells to tuned intravascular viscosities, which is doable in ZF. What about extravasation

in that model ? Metastasis ? Other vascular regions ? Finally, are the authors really injecting cells in the dorsal aorta rather than in the duct of Cuvier ?

- I;374 : again, it is of utmost importance to assess transcriptional regulation imposed by long-term viscosity.

- I.380 : "These data are consistent with the higher migratory/invasive potential of pre-conditioned cells observed in vitro and in zebrafish". How can the authors link both phenotypes ? I don't see the consistency here.

- I.396 : the extension to a pro-metastatic phenotype is based on experiments that drastically differ from the 1st part of the Ms – instantaneous effect should be considered and assessed over invasion, growth and intravasation mechanisms. As for metastasis, since this is a long-term effect, either the authors carefully assess each phenotype (Fig.1-3) or they try to tune viscosity of vascular fluids or they assess transcriptomic regulation imposed by viscosity.

- The authors use mouse experimental metastasis with tail vein injection to monitor lung colonization. While the data suggesting a cytoskeletal memory effect seems quite convincing at first glance, the relevance of the experimental model is highly questionable and severely tones down the validity of the data. The blood flow dissemination, and subsequent outgrowth, drastically differs from stroma invasion (with shear stress that are several orders of magnitude higher). Furthermore, it is now well demonstrated that after extravasation, cancer cells will colonize niches that are nearby blood vessels with little need for high migration capacities. Is there any difference in cell division abilities in the lungs observed in 0.77 vs 8cP pre-conditioned cells ?

- The use of actin as a normalizing control in HLINE experiments might not be the most relevant as extracellular pressure acts on actin cytoskeleton and there is no data supporting the unaltered expression of actin in response to the 8cP culture in the current manuscript

Additional comments :

- Overall while there are many data quantified, relevant representative images are missing so the reader can appreciate the change in phenotypes.

- All data should be displayed with SD, if the dataset successfully passed relevant statistical tests based on raw values, then it should not matter and it would be statically cleaner.

- Microscopy modalities used should be written in the caption to appreciate data quality.

- SC stands very likely for scramble control but is never defined. It should be properly written at least in figure captions.

- MICs/OEM should be defined

Referee #2 (Remarks to the Author):

The manuscript "Counterintuitive effect of viscosity on enhancing motility and metastasis" by Bera, Kiepas, Godet, Li, Mehta, Ifemembi, Paul, Sen, Serra, Lee, Zhang, Boen, Mistriotis, Gilkes, Tanner, Valverde, Sun, and Konstantopolous is an elegant and rigorous exploration of a novel phenomenon driving cellular behavior in response to increased extracellular fluid viscosity. Striking experiments on 2D surfaces and in 3D confining microchannels demonstrate that cells respond dynamically and robustly to extracellular fluid viscosity by, among other responses, increasing their migration speed, migration mode, lamellipodia growth, and tumor escape. This is found to be modulated by ARP2/3 activity, NHE1-dependent activation of the ion channel TRPV4, and subsequent RhoA signaling-induced contractility. The authors extend these findings in vivo by analyzing the migration of pre-treated cancer cells in both zebrafish and mouse models.

Overall, this work is likely of immediate interest to a wide swath of the cell biology community, and of urgent interest to the mechanobiology community. Observations on mechanical memory of fluid viscosity has the potential to provide new avenues of mechanical interventions into cell behavior that can be explored in a number of applications, as the authors touch upon in the discussion. In the context of this work, the authors have provided ample evidence, in painstaking detail, for the mechanism driving this novel phenomenon. It is also of note that they have demonstrated the physiological relevance of fluid viscosity in two separate model organisms,

further underscoring the universal nature of their findings. Some small suggestions are made below, most of which can be addressed in the discussion.

Minor revisions/comments

An interesting emerging observation in cancer mechanobiology, usually in the context of YAP translocation or tropomyosin activation, has been that many cancer cells become less mechanosensitive to their surroundings. Here, every cancer line presented is strongly sensitive to viscosity. Have the authors found any cell lines that are not susceptible to viscosity changes? Tangential to this, the hAOSMCs were exposed to 9.5 cP instead of 8 cP, suggesting that different cell lines have different threshold levels of viscosity response (for 231s in Fig. 1A this appears to be 3 cP). Does this threshold have any tissue- or disease-specific physiological relevance?

Several works have speculated that cell migration in confinement can be related to the total mass of fluid the cell needs to displace through the exit reservoir (which normally assumes an impermeable seal between cell and channel wall). In this case, by increasing the viscosity, the total mass of the fluid is also increasing (although only negligibly- the 0.6% methylcellulose solution would only increase the fluid density by 0.6%!). Can these two models be united? This is briefly touched upon in the discussion. The total fluid mass to be 'pushed' by the cell only increases to a small degree here, but the cellular response is quite robust. Does this imply that fluid viscosity is far more important than simple mass?

Numerous assays in this work reveal a higher degree of cell polarization in confinement in response to elevated viscosity, including the distribution of myosin II, ARP2/3 localization, and RhoA activity. Accordingly, was an increase in directionality (i.e. a reduction in 'reversals' of cells in confined channels) observed in high viscosity environments? (This is shown in EFig 9 in zebrafish- just curious if it was also observed in vitro) The denser actin network and lamellipodial stability would also theoretically contribute to this.

In general, a framework is presented in which at high viscosities, bleb-based migration and its supporting signaling pathways give way to mesenchymal migration. This is true for the majority of cells, but Efig 2b it appears that ~20% of migrating cells still utilize blebs. Of course it can be challenging to binarize a process that can exhibit some degree of overlap, but I'm wondering if the blebs present in high viscosity environments are morphologically different than those at physiological viscosity.

Is it possible or illuminating to track the dynamics of switching from low viscosity to high viscosity by introducing high viscosity media once cells are already in the channels? If high viscosity media is supplied to cells that are already moving in a 0.77 cP channel, can you visualize the blebs disappearing? Related to this- from the methods section, it seems as if cells are plated into the cell-seeding channel, then allowed to enter the confining channel in the absence of any chemoattractant or flow. While it appears as if all analyses were performed on cells that had already entered the channels, do the authors have any observations on whether cells more likely to enter confinement as a function of viscosity?

Given the 5-fold or greater increase in FA number shown in Efig 3f, and the slight yet significant increase in FA surface area per cell (Efig 3i), is it safe to say that in high viscosity cells, the individual focal adhesion puncta are much smaller? Does the model in Efig 5e take this into account?

To what degree do the ISVs recapitulate the confining microenvironments used in the in vitro work? The diameters shown in Fig. 4d are mostly around 10 μm , suggesting a cross-sectional area closer to the 10 μm wide channels than the 3.5 μm wide channels. Two cell lines were analyzed in 10 μm channels, but do the MDA-MB-231 cells also show this enhanced speed in 10 μm channels

after viscosity pretreatment?

Edit suggestions:

Although this group's work on microchannels is widely known in the field, a diagram of the microchannel chip setup might be helpful to readers less familiar with microchannel confinement strategies.

The word 'viscosity' in the title is potentially confusing, as a number of works have used this simple term to describe substrate viscoelasticity (e.g. Charrier/Janmey et al Nat. Comm. 2018). Perhaps 'fluid viscosity' would be more descriptive without being too wordy.

In Efig 3a, I'm having trouble understanding what the red triangle represents, although it's referred to as 'instant of viscosity applied'. Does this imply that the X axis supposed to be time?

The model in Fig 2f appears to be an excellent prediction of the data shown in Fig 2g. Putting the axes on the same range would emphasize this (or even combining them into a single graph!)

Presumably OEM refers to Osmotic Engine Model, but this is not introduced as an acronym prior to its use on page 11.

Page 9 Line 256: I think it's meant to say 'Fig. 2o' instead of 'Fig. 2n'

Efig 5e-f could be labelled FA and Actin in the figure for clarity.

Typo on Page 18 line 559: Liefact vs. Lifeact

Typo in the axis of Efig 2J

Referee #3 (Remarks to the Author):

A. Kiepas, I. Godet, Yizeng Li, P. Mehta and colleagues, a strong group of investigators led by Dr Konstantopoulos of Johns Hopkins, report novel and interesting findings on how extracellular viscosity can enhance cells' motility and metastatic capacity of malignantly transformed cells. These results are novel and interesting, at a general level of interest, yet underdeveloped in some aspects.

Results and putting-it-all-together are curated well, but in need of some fixing and clarifying.

The cellular machinery of the cell lines examined is clarified reasonably well.

A central element identified is the TRPV4 ion channel.

There are several TRPV4-related aspects that the authors need to address by conducting additional experiments, which will also benefit the entire mechanistic contribution of the paper, which is overall convincing.

1) TRPV4 has been examined in the context of tumorigenesis and metastasis, with one authoritative review here

Yu et al. Cell Death and Disease (2019) 10:497

<https://doi.org/10.1038/s41419-019-1708-9>

Fig 2 of this paper highlights an important property of TRPV4 in human cancers. Whereas most examined cancers show increased expression of TRPV4, the common cancers, prostate

adenocarcinoma and esophagus carcinoma have significantly reduced TRPV4 expression. There are a number of cell lines that need to be included in a revised Fig. 1b (see below), but these two particular types of cancer do need to be examined, whether they follow the logic the authors have determined, or whether there is a non-universal role for TRPV4, depending on tumor type.

Glioma should also be studied

<https://pubmed.ncbi.nlm.nih.gov/29928875/>

because migration and invasiveness are particularly prominent features of gliomas. What is important in the glioma context is that they rather express AQP4, not AQP5. Understanding glioma cells' migratory capacity in extracellular matrix of watery vs increased viscosity will help us understand whether the TRPV4-dependent response to increased viscosity has to be coupled to AQP5, or whether other AQPs can fill in as well.

2) TRPV4 has been established as a mechanotransducer in endothelial cells that transduces within 4msec after the onset of shear stress (Don Ingber's group), also by recent diligently conducted studies from Dr Kate Poole's lab - these studies need to be referenced and taken into account experimentally.

<https://pubmed.ncbi.nlm.nih.gov/20725677/>

<https://pubmed.ncbi.nlm.nih.gov/28135189/>

<https://pubmed.ncbi.nlm.nih.gov/30984749/>

<https://pubmed.ncbi.nlm.nih.gov/33537292/>

Questions arise to the contribution to TRPV4 functioning of beta1 integrins, CD98, and also vinculin, and they need to be addressed by the authors.

TRPV4 - vinculin

<https://pubmed.ncbi.nlm.nih.gov/30674675/>

3) Fig. 3I depicts TRPV4 as a swelling-activated calcium-permeable channel downstream of viscosity-mechanical sensing machinery.

Have the authors considered whether TRPV4 might be playing multiple roles in response to increased viscosity, e.g as a more direct and upstream mechanotransducer as well ?

helpful ref.

<https://pubmed.ncbi.nlm.nih.gov/15952033/>

4) loss-of-function of TRPV4 experiments are well conducted, however, enhanced function of TRPV4 can be updated based on a recent insight.

Lysophosphatidylcholine (LPC) has been found an endogenous activator of TRPV4, via direct activation through a C-terminal binding site for the glycerophospholipid. LPC is also a relevant component of tumor microenvironments. Therefore, this should be tested.

[https://www.gastrojournal.org/article/S0016-5085\(21\)00576-X/fulltext](https://www.gastrojournal.org/article/S0016-5085(21)00576-X/fulltext)

In addition, TRPV4 can be sensitized by proteolytic signaling via PAR-2 activation. Tumor microenvironments do have robust proteolytic activity.

This should also be accommodated experimentally.

<https://pubmed.ncbi.nlm.nih.gov/17124270/>

5) TRPV4 activation in response to flow has been postulated in the context of signaling via the primary cilium.

PMIDs

25143588

19790068

18719094

18024594

15753126

The authors need to explore whether primary cilia are (critically) relevant for tumor cells' response to increased viscosity.

6) re link TRPV4-RhoA signaling,

Dr Charlotte Sumner's work (out of Hopkins - !) needs to be referenced

<https://pubmed.ncbi.nlm.nih.gov/33664271/>

The following paper also needs to be referenced and explicitly commented on

<https://pubmed.ncbi.nlm.nih.gov/33230171/>

Going through the ms., minor construction sites.

- EC fluid viscosity also important for inflammation, cellular motility in development
- intro line 64 "holistic" - inappropriate choice
- stay at a general level of understanding and comprehension, avoid using jargon or code-talking abbreviations:

- What is "MIC", what is "OEM" ?

DUANU - don't use abbreviations nobody understands.

First sentence of the Discussion: "apodictic influence" - non-helpful starter sentence of this all-important section.

Referee #4 (Remarks to the Author):

In this manuscript, Bera and colleagues report how an increase in medium viscosity induces cell migration and dissect the pathway that is involved in this phenomenon. They show that this effect can be observed in many transformed and untransformed cells. It involves polymerization of branched actin by the Arp2/3 complex as well as myosin-mediated contractility. It also involves mechanical transduction of viscosity through the sodium proton antiporter NHE1, cell swelling, Ca²⁺ spikes and the cation channel TRPV4. Using mouse and zebrafish xenografts, the authors show that high viscosity can be used in vitro to reprogram cells for enhanced migration and metastasis formation in vivo.

I found this paper original and interesting. To me it is a good candidate for publication in Nature. Despite its complexity, this manuscript should be of interest to a general audience for one reason or another: mechanotransduction of a new physical parameter, role in tumor cell dissemination, as a means to increase cell migration, as a way to access the gene expression program of mesenchymal motility ... Even though it is not the first time that this effect of increased viscosity is reported in the scientific literature, this paper analyzes in depth the phenomenon, which is certainly not very well known. It is part of a stream of current studies showing the importance of physical parameters for almost all cell functions and in particular cell migration. Overall the data are convincing, although some points need to be verified before I can recommend publication without reservation.

General Comments

1 Are branched actin networks the primary sensor of medium viscosity ? The authors did a good job of positioning the NHE1 contribution upstream of the TRPV4 contribution (Fig.2). One can expect cortical branched actin to be further upstream, since branched actin polymerization is enhanced both in vitro and in vivo when the pushing force that is generated by branched actin encounters resistance, as it is properly discussed in the manuscript. However, this point is not experimentally addressed. It is easy for the authors to demonstrate this point with the system they have in hands. Would they still observe Ca²⁺ spikes, increase in cell volume and TRPV4 currents when Arp2/3 is depleted ? If yes, do these responses depend on the dose of CK-666 ? This Arp2/3 dependence is particularly important, since myosin-induced contractility is also required and in the mind of most readers, mesenchymal migration based on branched actin is antagonistic to amoeboid bleb-based migration that depends on contractility. Here both pushing

and pulling forces might well be required, but Arp2/3 should definitely be upstream. That would be consistent with the lamellipodium and cortical branched actin being mechanical sensors, as observed for sensing substrate stiffness or cell density in a monolayer (Oakes ... Gardel, PNAS 2018; Molinie ... Gautreau, Cell Res 2019).

2 This manuscript does not only address the required machinery. It also shows that exposure to viscous medium for 6 days commit cells into a motile status that is sustained even when high viscosity is not maintained (Fig.4). This important result shows that cell motility corresponds to a particular gene expression program that I believe would be important to characterize here by transcriptomics. While this may seem like an unnecessary extension of the manuscript, it would actually benefit from the detailed characterization of the conditions (pretreatment with high viscosity, TRPV4 depletion) as well as the in vivo consequences to provide a resource for the further dissection of this phenomenon that this manuscript should trigger upon publication. Molecular machines involved in the process should be well expressed, perhaps even induced by high viscosity, but not down-regulated... This can be verified for the here identified players (Arp2/3, NHE1, TRPV4), but a global transcriptomics analysis would provide one more reason to cite this study in the future.

3 Organization and clarity of the manuscript can probably be improved. To me, the part on integrin beta1 is less required than the rest. It is already clear that viscosity-induced migration is of the mesenchymal type. The requirement for Arp2/3 is enough to make the point. This part can be removed if space is needed. The discussion indeed recapitulates most of the points, but it is long and does not provide much new information. To me, the discussion is not very consistent with Nature's style.

Specific Points

4 I agree that myosin-mediated contractility is not only a means to migrate through a bleb-based mechanism, but also a mechanism by which it promotes mesenchymal migration. This point is quite critical since it explains why both branched actin and myosin mediated contractility are required, despite the well-established antagonism between mesenchymal and amoeboid types of migration. In addition to ROCK inhibition (ED Fig.7o), the authors should show that MIIA depletion also reduces the actin retrograde flow, and emphasize this point much more in the manuscript, perhaps by including these experiments in the main figures. A seminal reference on the topic is Wilson... Theriot Nature 2010. I believe that it should be cited.

5 This manuscript does not adhere to what has become a standard in the field. shRNA mediated depletion is not systematically validated for efficiency (mRNA levels and/or WB) and for the fact that the phenotype is not due to off-target. It leaves the room for potential wrong conclusions. At least the main phenotype should always be validated with two independent sequences.

Arp2/3: WB, targeting of 2 different subunits, so I guess it is OK.

NHE1: WB, a single shRNA sequence. Not OK.

AQP5: no WB, a single shRNA sequence. Not OK.

TRPV4: mRNA levels, WB, 2 independent sequences, so it is probably good. But it is not clearly stated that one sequence was used for the main figure and the other one for the extended data.

Myosins: no WB, single sequence. Not OK.

ITGB1: mRNA levels, WB, but a single shRNA sequence. Not OK.

It is up to the authors to include these additional controls, probably in Extended Data, while not convoluting too much the manuscript.

6 Important abbreviations such as MICs or OEM are not defined.

7 I did not assess the accuracy of statistical tests (for lack of time) and mathematical modeling

(for lack of expertise).

Author Rebuttals to Initial Comments:

Authors' Responses to Editor's and Reviewers' Remarks

First and foremost, we would like to thank the editor and the reviewers who have meticulously read our manuscript and provided us with insightful comments. We also greatly appreciate the invitation to submit a revised manuscript, which addresses all of the reviewers' concerns. Starting on the next page, we have pasted each of the reviewers' comments followed by our responses.

To address the reviewers' remarks, we performed a series of **25 new experiments** (each performed in triplicate unless shown otherwise), which resulted in the generation of **33 figure panels presented in our revised manuscript**. We also generated an additional **13 figure panels that we only included in the rebuttal letter for the reviewers' evaluation**. However, we chose not to include them in our revised manuscript due to an overwhelming amount of data already present. The list of experiments is shown below:

1. Extravasation in the chick embryo model after MDA-MB-231 pretreatment at 3 cP (**in response to Rev. # 1**).
2. Measurement of membrane tension at elevated viscosities and under inhibition of Arp2/3, NHE1 and TRPV4 (**Rev. # 1**).
3. Comparison of actin network density in shTRPV4 and scramble control cells (**Rev. # 1 and Rev. #3**)
4. Evaluation of calcium spikes and cell volume after Arp2/3 inhibition (**Rev. # 1 and Rev. #3 and Rev. #4**)
5. Migration of SUM159 and brain metastatic MDA-MB-231 cells under inhibition of NHE1 (**Rev. # 1**)
6. Evaluation of memory formation after treatment of shNHE1 cells with 8 cP media (**Rev. # 1**)
7. Integrin β 1 staining to show colocalization with paxillin GFP (**Rev. # 1**)
8. Evaluation of relative levels of myosin A, B and C (**Rev. # 1**)
9. Evaluation of NHE1 activity after actin abrogation (**Rev. # 1**)
10. Evaluation of ezrin polarization after cell treatment with LatA at elevated viscosities (**Rev. # 1**)
11. Transwell assays to test tendency of cells to move towards or away from viscosity (**Rev. # 1**)
12. RNA sequencing (**Rev. # 1 and Rev. #4**).
13. Subcellular localization of YAP (**Rev. # 1**)
14. Evaluation of memory formation after treatment of cells with 8 cP media under the presence of YAP inhibitor (**Rev. # 1**)

15. Evaluation of morphology of cellular blebs at two different viscosities (**Rev. # 2**)
16. Evaluation of entry time of cells into narrow channels (**Rev. # 2**)
17. Further characterization of non-cancerous human fibroblast cell migration at 3 cP (**Rev. # 2**)
18. Evaluation of persistence of cells at 8 cP (**Rev. # 1 and Rev. # 2**)

19. Migration characterization of human osteosarcoma cells and U87 glioma cells at elevated viscosities with or without TRPV4 inhibition (**Rev. # 3**)
20. Measurements of calcium activity in U87 glioma cells (**Rev. # 3 and Rev. #1**)
21. Evaluation of calcium spikes in integrin β 1 KD cells (**Rev. # 3**)
22. Evaluation of calcium levels in response to a new TRPV4 activator 18:1 Lyso phosphocholine (**Rev #3**).
23. Evaluation of motility rescue in shNHE1 cells at 8 cP by treatment with a new TRPV4 activator, 18:1 Lyso phosphocholine (**Rev. # 3**)

24. Migration with two separate shRNA sequences of NHE1 (**Rev. # 4**)
25. Evaluation of actin flow after myosin IIA depletion using shRNA (**Rev. # 4**)

In addition, we have highlighted the editorial changes in our revised manuscript in red.

REFEREE #1:

In this study by Bera et al., the authors build on previous work from the lab (Stroka et al., 2014, Zhao et al., 2019, Yankaskas et al., 2019, Zhao et al., 2021) and describe extracellular viscosity as a new mechanical cue, which appears essential in increasing cell migration abilities and how this could influence cancer progression.

More specifically, the authors aim at unravelling three important aspects:

- How does external viscosity lead to altered cell mechanics and phenotype and change in migration abilities?
- What is the molecular pathway leading to increased migration in response to increased viscosity?
- How does this influence cancer progression and metastasis?

The authors rely on a microfluidic based confinement device associated with medium viscosity tuning. They conclude that external viscosity alters cortical actin leading to an opening of Na⁺/H⁺ transporter NHE1 and aquaporin AQ5 leading to an increase in cell volume. This cell swelling induces an activation of the mechanosensitive TRPV4 channel and Ca²⁺ entry leading to the activation of the RhoA/actomyosin contraction pathway. Furthermore, they show that this viscosity-induced mechanical changes are retained suggesting mechanical memory conservation. Later relying on *in vivo* experimental metastasis models, they suggest that cancer cells preconditioned in more viscous media are more metastatic than their normal counterparts.

While the work is overall well performed with large amounts of data, the manuscript would benefit from several improvements to strengthen the conclusions and convey a better message.

Authors' response: We thank the reviewer who concluded that our work was “**well performed with large amount of data**”. Importantly, we followed their advice and performed a series of new experiments to meticulously address all of their insightful points.

Comment #1: I would like to start my report by commenting on some claims in the abstract of this study, and thereby identify some limitations of the study. While I overall can't agree more that looking at the contribution of external viscosity to cell behavior and fate is important, I wonder why the authors decided to focus on viscosity changes that are likely to occur in interstitial fluids and have not considered other fluids such as blood and lymph whose contribution, with regards to cancer progression, is important.

Authors' Response: Per reviewer's suggestion, we generalized our wording in the abstract (please see Line 34) and through our manuscript. Along these lines, we wish to respectfully point out that we examined a range of **patho-physiologically relevant** viscosities, which are encountered not only in the lymph and interstitial fluids (≤ 3 cP) but also in the blood (5-8 cP).

Comment #2: Second, the vast majority of the work, which is carefully documented, focuses on very particular structures, which are protrusive regions of the cell, i.e. leading edge in the abstract. It is unclear if these effects would apply to cells that are not protrusive, which is likely to happen in various situations such as tumor masses subjected to interstitial fluids.

Authors' Response: We appreciate the reviewer's remark for our “**carefully documented**” work. First, we wish to point out that motile cells, such as invading tumor cells, display front-to-rear polarity. As such, in many cases *in vivo*, including cell entry into a confining microenvironment or

during the intravasation process, the cell front represents the edge which is first exposed to a biochemical or biophysical cue.

We also wish to bring to their attention our findings where 3D spheroids (which recapitulate aspects of a primary tumor mass) were kept on collagen-coated dishes and submerged in media of specified viscosities. As such, the medium used in these assays exposes cells in the spheroids (especially, the cells in the periphery of the spheroids) to specified viscosities. Our data reveal that elevated extracellular fluid viscosity increases tumor cell dissemination from 3D tumor spheroids (**Fig. 1d,e**), which is in accord with our migration data on 2D surfaces and inside confining microchannels. Consistent with our signaling pathway, TRPV4 inhibition markedly delays cell dissemination from 3D spheroids (**Fig. 2m**).

Comment #3: Third, while the latter effects look at almost-instantaneous effect of viscosity on cell motility, the last part of the abstract and thus manuscript aims at linking such phenotypes to a relevant pathological situation, metastasis. However, to do so, the authors coin a different approach, i.e. memory of viscosity, which, in my opinion, cannot be a simple extrapolation of changes that occur instantaneously on migrating cells. Although very interesting, phenotypes of cells that are exposed to long-term viscosity are likely to be very different and include additional transcriptional regulation for example. Apart from TRPV4, the authors have not looked at whether effects described in Fig.1-3 (and associated supplemental material) are at play in this situation of viscosity memory. Moreover, it remains unclear why the authors have decided to focus on i) intravascular migration of cells in that context, whose relevance to metastasis needs to be demonstrated, and on ii) experimental metastasis in mice that is likely to involve many other mechanisms than increased cell motility. Assessing tumor invasion/intravasation (or other cell motility phenotypes) in realistic *in vivo* situations (intravital imaging) could have been a more suitable option to directly link the *in vitro* description of the effect of viscosity on cell motility.

Authors' Response: Cells *in vivo* are subjected to a wide range of viscosity values for different time periods in distinct regions of the body (i.e., interstitial fluids, blood circulation etc). { REDACTED}
We herein wished to develop a comprehensive understanding of how exposure to different viscosities would affect metastasis, which is a process that is not developed instantaneously. That is why, we looked at the **cell memory to viscosity**.

Please note that it is at minimum challenging to precisely alter the viscosity *in vivo* in order to study the biological effects of such an alteration on single cells. Using our extensive *in vitro* characterization, we identified that the TRPV4 mechanosensor is a key transducer of viscosity-induced cell responses. Importantly, we demonstrate that TRPV4-depleted cells fail to form viscous memory *in vitro* and display reduced metastatic potential *in vivo*. We also wish to point out that TRPV4 is not the most upstream molecule in the signaling cascade of events that we deciphered (**Fig. 3l**; please also see our response to **comment #4**). However, TRPV4 is the molecule which converts the biophysical signal from actin loading and NHE1-dependent cell volume regulation into a biochemical response via calcium signaling. { REDACTED}

Regarding the reviewer's remark on our *in vivo* assays, we respectfully point out that first, the zebrafish model enabled us to **extend our *in vitro* findings** on the enhanced motility in response to cell pre-conditioning at elevated viscosities (8 cP versus 0.77 cP) to **the *in vivo* setting**. Second, using a mouse model, we demonstrated that cell pre-conditioning to elevated viscosities (8 cP versus 0.77 cP) results in markedly elevated, TRPV4-dependent experimental metastasis. Third, we employed the **chick embryo model** to demonstrate that cell pre-conditioning at 3 cP versus 0.77 cP leads to elevated tumor cell extravasation (**new Extended Data Fig. 9e**). Fourth, we wish to conclude our response by pointing out important technical limitations:

i) MDA-MB-231 cells do not establish primary tumors in the chick embryo model, and as such, we cannot study cancer metastasis in this model. That is why, our new assays focused on MDA-MB-231 cell extravasation, which is a key step in the metastatic cascade of events (and also requested by this reviewer).

ii) Regarding the suggested intravital imaging of MDA-MB-231 cells, we {REDACTED} not yet developed a mammary imaging window that can be transplanted onto the mammary fat pads of mice. As such, the reviewer's proposed experiment requires the

development of new methodology.

iii) The precise spatiotemporal alteration of viscosity *in vivo* is not feasible, which further prevents us from performing intravital imaging of cancer cells subjected to elevated viscosity acutely.

In sum, we have provided new *in vitro* (microfluidic assays) and *in vivo* (chick embryo model) data to address the reviewer's insightful questions.

Along this line, I have identified additional and more specific pitfalls :

Comment #4: - While the mechanoresponsive pathway starting from NHE1/AQP5 opening and leading to RhoA-mediated contractility and improved migration abilities is **elegantly demonstrated with strong data** supporting the claim, the transmission or clutch conveying the external viscous mechanical cue into an internal cortical cell response is far less clear and strong data to support the claims are missing. In particular, the cytoskeleton reorganization data in the first part of the manuscript are sometimes confusing and it is unclear how actin response to extracellular pressure induces the subsequent mechanoresponse. Overall, I feel this important point relies more on data extrapolation and should be addressed.

Authors' Response: We thank the reviewer for their kind remarks regarding the elucidation of viscosity-mediated mechanoresponsive pathway. Per reviewer's recommendation, we performed new experiments to further support our model. As we reported in our original submission, the **actin network is the primary sensor of extracellular fluid viscosity**, which is supported by both experimental and stochastic modeling data. Specifically, our stochastic model reveals that actin loading in response to elevated viscosity is capable of altering the dynamic and architectural properties of the network. These model predictions were verified first by using photoactivatable fluorescent probes which show slower retrograde actin flow at 8 cP, and second by super resolution imaging which reveals a dense and highly branched network at elevated viscosities. Our findings on actin loading in response to elevated force are in line with prior work in reconstituted systems by Fletcher's group in *Cell* (2016) and in cellular systems by Sixt's group in *Cell* (2017).

In our initial submission, we wrote that Arp2/3-dependent actin loading in response to elevated viscosity triggers NHE1-dependent swelling which in turn activates TRPV4 and enhances calcium influx that leads to increased RhoA-dependent cell contractility.

Using FLIM imaging of Flipper-TR, we now demonstrate that membrane tension is increased upon acute cell exposure to elevated viscosity (**new Fig. 2g,h**). Importantly, **new data** also reveal that inhibition of Arp2/3 function abrogates viscosity-mediated volume increase (**new Extended Data Fig. 6i**), membrane tension increase (**new Fig. 2h**) and calcium spikes in cells (**Fig. 2i**), providing further evidence to support the concept that the actin network is the sensor of extracellular fluid viscosity.

Consistent with the concept that the branched actin network is upstream of TRPV4 activation, new STORM imaging data reveal that TRPV4 silencing has no effect on actin remodeling at elevated viscosities (**new Extended Data Fig. 6j**). Furthermore, TRPV4 knockdown prevents channel activation without altering cell volume (**new Extended Data Fig. 6e**) or membrane tension (**new Fig. 2h**). We further demonstrate that NHE1 inhibition blocks the viscosity-induced cell volume and membrane tension increases as well as TRPV4-dependent calcium spikes (**new Figs. 2h, i, and Extended Data Fig. 6e**).

Taken together, these data conclusively establish that the branched actin network is the sensor of extracellular fluid viscosity, which initiates a cascade of events. In this cascade of events, Arp2/3-dependent actin loading in response to elevated viscosity triggers NHE1-dependent swelling, which leads to an increase in membrane tension that triggers TRPV4 activation and calcium influx, thereby resulting in RhoA-dependent cell contractility and enhanced cell motility.

Comment #5: - How external viscosity is sensed by cells, within protrusive structures, and conveyed through “mechanical loading” to actin remodeling needs to be identified.

Authors’ Response: We respectfully refer the reviewer to our detailed response to **comment #4** above. More specifically, our findings on actin loading in response to elevated force are in line with prior work with actin in reconstituted systems by Fletcher’s group in *Cell* (2016) and in cellular systems by Sixt’s group in *Cell* (2017), a view also shared by **reviewer #4**.

Comment #6: - How similar is the immediate response to long-term “memory” viscosity needs to be documented carefully as it is likely that mechanisms differ.

Authors’ Response: We herein demonstrate that TRPV4-depleted cells fail to form viscous memory *in vitro* and display reduced metastatic potential *in vivo*. Although TRPV4 is not the most upstream molecule in the signaling cascade of viscosity-mediated events (**Fig. 3I**), TRPV4 is the molecule which converts the biophysical signal from actin loading and NHE1-dependent cell volume regulation into a biochemical response via calcium signaling. {REDACTED} Please also note that we have added new data from RNAseq assays (**new Extended Data Fig. 10a-d**), which were requested in a subsequent comment. The new data point to the TRPV4-dependent modulation of the Hippo pathway as a key event in the acquisition of long-term memory.

Comment #7: - Most of the *in vitro* experiments model invasion-like process and suggest the importance of altered mechanics in response to increase in viscosity in a switch of migration phenotype, which is typically seen during primary tumor invasion. After cancer cell bloodborne

dissemination, extravasation is followed by tumor cell colonization of niches which are often perivascular (see the large amount of data on brain, lungs, liver or bone metastasis), which questions the relevance of the *in vivo* experimental metastasis model chosen. Correlating the extensive amount of data obtained in Fig.1-3 would require a better adapted *in vivo* evaluation (such as intravital imaging of tumor invasion for example)

Authors' Response: We respectfully point out that first, the zebrafish model enabled us to **extend our *in vitro* findings** on the enhanced motility in response to tumor cell pre-conditioning at elevated viscosities (8 cP versus 0.77 cP) to **the *in vivo* setting**. Second, using a mouse model, we demonstrated that cell pre-conditioning to elevated viscosities (8 cP versus 0.77 cP) results in markedly elevated, TRPV4-dependent experimental metastasis. Third, we employed the **chick embryo model** to demonstrate that cell pre-conditioning at 3 cP versus 0.77 cP leads to elevated tumor cell extravasation (**new Extended Data Fig. 9e**). Fourth, we wish to point out the following important experimental limitations:

i) MDA-MB-231 cells do not establish primary tumors in the chick embryo model, and as such, we cannot study cancer metastasis in this model. That is why, our new experiments (**new Extended Data Fig. 9e**) focused on MDA-MB-231 cell extravasation, which is a key step in the metastatic cascade of events.

ii) Regarding the suggested intravital imaging of MDA-MB-231 cells, we {CO not yet developed a mammary imaging window that can be transplanted onto the mammary fat pads of mice. As such, the reviewer's proposed experiment requires the

development of new methodology.

iii) The precise spatiotemporal alteration of viscosity *in vivo* is not feasible, which further prevents us from performing intravital imaging of cancer cells subjected to elevated viscosity acutely.

In sum, we have provided essential *in vivo* data using three different models (zebrafish, chick embryo, and mouse models) showing that elevated viscosity enhances cell motility, tumor cell extravasation and metastasis *in vivo*, which verify and extend our detailed *in vitro* findings.

Below are the specific comments on the manuscript:

Introduction

Comment #8: L44 : It should be mentioned that migration is also important in developmental process, this even more important in the cancer context where EMT and cell stemness, 2 important developmental pathway are hijacked by cancer cells.

Authors' Response: We thank the reviewer for this suggestion, and we have appropriately revised the relevant statement to incorporate their remark (see **Lines 51-52**).

Comment #9: L50 : While the authors mention that viscosity is driven by augmented macromolecules presence, it should be mentioned that this is usually driven by lymphatics drainage failure which is an important marker of cancer progression and altered tumor mechanics (see the work from Rakesh Jain for instance)

Authors' Response: We thank the reviewer for this suggestion, and we have appropriately revised the relevant statement to incorporate their insightful comment (see **Lines 59-60**).

Comment #10: L.59 : "how do cells sense the physical cue of elevated, yet physiologically

relevant, extracellular viscosity?” This is indeed an important question that would require specific experiments to identify the viscosity sensing mechanism at the plasma membrane.

Authors' Response: We respectfully refer the reviewer to our detailed response to their comment #4 above.

Comment #11: L.70 : “We also extend our findings to the *in vivo* setting by showing that breast cancer cells pre-exposed to elevated viscosity exhibit faster motility through intersegmental vessels (ISVs) in zebrafish, and a higher ability to establish lung metastases in mice after tail vein injection” As discussed earlier (abstract comments) and later, it remains unclear what is the rationale of looking at intravascular migration of long-term viscosity treatments without looking at direct effect on metastasis (extravasation, outgrowth).

Authors' Response: We respectfully refer the reviewer to our detailed response to their (similar comment #7 above. Regarding the reviewer's remark on extravasation, we wish to reiterate the new experiments we carried out using the chick embryo model (**new Extended Data Fig. 9e**). In summary, we have employed three different models (zebrafish, chick embryo, and mouse models) to demonstrate that elevated extracellular viscosity acts as a novel biophysical cue to enhance cancer cell motility through ISVs (**Fig. 4d-f** and **Extended Data Fig. 9b-d**), extravasation (**new Extended Data Fig. 9e**) and lung colonization (**Fig. 4g-n** and **Extended Data Fig. 9f**) *in vivo*. These *in vivo* results verify and extend our detailed *in vitro* findings.

Results

Comment #12: - I.97-99 : how universal is such an effect ? Would immune cells be sensitive to it as well since they are the ones exposed massively to interstitial fluids? Is the viscosity-dependent effect on motility dependent on motility types? The different cells were not tested for physiologically relevant viscosities. Intermediate range found in interstitial fluid flow (1-3 cP, as suggested in the introduction) should also be tested to support the relevance of the role of viscosities

Authors' Response: We hope that the reviewer appreciates the wealth of the data that we have provided in our manuscript, which includes numerous normal (human fibroblasts, human aortic smooth muscle cells) and cancerous (MDA-MB-231, BrM2 and SUM159 breast cancer cells, human osteosarcoma (HOS) cells, and U87 glioblastoma cells) cell lines. Importantly, we wish to keep our work focused on cancer cell migration and metastasis. Immune cells can be tested in follow up papers by Dr. Matthieu Piel or Dr. Michael Sixt, who are leading experts in the area of immune cell migration.

Comment #13: - I.103 : how do the authors make sure that viscosity is tunable in 3D environments? (spheroids?) What is the link between confinement-based motility and 3D spheroids dissociation? Is there an effect on migration directionality in the spheroid migration assay?

Authors' Response: As we have now stated in the figure legends and the *Methods* section of the manuscript, the 3D spheroids were kept on collagen-coated dishes and submerged in media of specified viscosities. As such, the media used in these assays, which has been thoroughly characterized in **Extended Data Fig. 1a,b,d**, exposes cells in the spheroids (especially, the cells in the periphery of the spheroids) to specified viscosities. Our data reveal that elevated

extracellular fluid viscosity increases tumor cell dissemination from 3D tumor spheroids *in vitro* (Fig. 1d,e), which is in accord with our migration data on 2D surfaces and inside confining microchannels. Consistent with our signaling pathway (Fig. 3I), TRPV4 inhibition markedly delays cell dissemination from 3D spheroids (new Fig. 2m).

In our manuscript, we used **complementary *in vitro*** (2D migration assays, confining channels and 3D spheroids) **and *in vivo* models** to demonstrate that elevated viscosity potentiates different steps of the metastatic cascade. The 3D spheroid assays reveal that elevated viscosity promotes cell dissemination from 3D tumor spheroids, whereas 2D and microfluidic assays establish that viscosity enhances migration in different microenvironments. These *in vitro* data coupled with our *in vivo* results establish the significance of extracellular fluid viscosity as an important biophysical cue, which alters cancer cell migration and metastasis.

Comment #14: - l.139-140 : Mechanical loading? Why focusing on the protrusive structures while cells in tumors aren't necessarily protrusive when exposed to IFP? How long are cells incubated for? Have the authors performed additional transcriptomic analysis (and compared with long-term viscosity experiments)?

Authors' Response: Regarding the “mechanical loading” remark, we wish to respectfully draw the reviewer’s attention to the fact that cells, exposed to elevated viscosity, experience higher mechanical resistance when they attempt to expand their cytoplasmic boundary because of the increased force upon the cell edge. This directly stems from Newton’s law of viscosity where an increase in fluid viscosity leads to an increase in shear stress. For example, it is easier to stir a spoon in a cup of water compared to that in a jar of honey, as honey has higher viscosity than that of water.

Elevated viscosity at any growing edge of the cell will impose increased mechanical loading on the growing actin network. Prior work has shown that mechanical loads either applied by a cantilever on a cell-free reconstituted actin network or by increasing membrane tension of a cell, alter properties of the actin network. Our stochastic model does not focus on the leading edge necessarily, and as such, can be applied to any growing edge of actin. However, in many cases *in vivo*, such as cell entry into a confining microenvironment or during the intravasation process, the cell “front” represents the edge which is first exposed to a biochemical or biophysical cue. We also refer you to our detailed response to **comment #2** above where we discuss the effect of extracellular fluid viscosity on tumor cell dissemination from 3D spheroids immersed in fluids of different viscosities.

Lastly, following the reviewer’s suggestion, we performed RNAseq experiments (**new Extended Data Fig. 10a-d**). {REDACTED}

Comment #15: - Would a gradient in cP act as a mechanotaxis cue? The microfluidics device could be used to generate such gradients taking advantage of the opposed inlets to assess the potential role of such cue. That would be similar

to the molecular gradients found in close to tumors lacking proper lymphatic drainage which have been shown to act as directional cues.

{REDACTED}

Comment #16: - I.144 : This reviewer does not have expertise to judge modelisation. However, when the authors claim that “These modeling data further verify that elevated extracellular viscosity is sufficient to alter the actin network at leading-edge” : why only at the leading edge?

Authors' Response: The stochastic model successfully predicts the effect of extracellular fluid viscosity on a growing network of actin. As verified experimentally by STORM imaging of the actin cytoskeleton, elevated viscosity (8 cP) causes cells to exhibit a dense, highly branched network within a growing edge (**Fig. 1i,j**). We wish to emphasize that our stochastic model does not focus on the leading edge necessarily, and as such, can be applied to **any growing edge of actin**.

Comment #17: - The analysis of lamella extension does not account for the increase of cellular volume (which is later clearly demonstrated). How can authors rule out that this increase is not a secondary effect from an overall volume increase? This should be clarified.

Authors' Response: The reviewer raises an excellent point. Please note that we see a $\geq 60\%$ increase in lamella area, whereas we observe a volume increase of $\sim 40\%$. If the increase in lamella were solely due to increase in cell volume, the change in area would be proportional to the square of the increase in dimensions due to uniform volume increase. Each length should increase approximately by cube root of 40% which is 3.419%. As such, the area should increase by square of 3.419% which is 11.7%. However, the increase in lamella is more than 60%, which cannot be accounted for only by volume increase. Importantly, we show using super resolution imaging that the actin structure in lamella is markedly different in cells subjected to baseline versus elevated viscosities.

Comment #18: In PA-beta-actin GFP dataset, I don't get what is the size of the photoactivation zone. The methods mention a point for activation? What is the size? The activated region seems quite big compared to the overall size of cells. A line scan perpendicular to the main polarization axis would be more convincing.

Authors' Response: It is a point activation whose size is determined by the resolution of the imaging technique. Using point spread function approximation, for the confocal imaging conditions we employed, we calculated the point size to be **~ 175 nm** in diameter (full width at half maximum). However, due to the very high mobility of the G-actin molecules tagged with photoactivated GFP, they diffuse out of the photoactivation zone during the time of photoactivation and before the capture of the first image. As such a “seemingly” larger area of activation appears in the image.

Regarding retrograde flow analysis, a line scan perpendicular to the cell membrane was indeed drawn and the kymograph was considered. These important points raised by the reviewer are now explained in the Methods section of our revised manuscript.

Comment #19: Arp3 signal intensities were quantified without any normalizing staining. How can the authors rule out that the increased intensities are due to a real accumulation of Arp2/3 and not an artifactual accumulation due to membrane ruffling leading to overlays of membrane/actin cortex ?

Authors' Response: Please note that these images were acquired using a confocal microscope where the z section was kept constant throughout the experimental conditions. Thus, the signal was captured from regions of equal thickness across all groups. This important point has now been clarified in the figure legend of our revised manuscript.

Comment #20: The authors have used Paxillin-GFP to monitor the effect of viscosity on FAs. Paxillin is known to be only transiently recruited to FA and absent from early talin enriched FA or late tensin enriched FA. These results should be confirmed with the proper integrin staining controls.

Authors' Response: Several studies have shown that integrin β_1 and paxillin co-localize throughout epithelial cells at adhesion sites, including cell-matrix adhesions formed in 3D environments (PMID: 26548801). In our MDA-MB-231 cells expressing paxillin-GFP, we also see a high degree of co-localization (~80%) between paxillin and integrin β_1 (**Fig. 1.3**). Please note that integrin β_1 may also be present in endocytic vesicles during intracellular trafficking to new adhesion sites (PMID: 22222055). In view of prior work and data shown in **Fig. 1.3**, we are confident that the results presented for paxillin-GFP accurately represent the state of adhesions in MDA-MB-231 cells. We chose not to include **Fig. 1.3** in our revised manuscript because it is already rich in data, and inclusion of additional data would render this article overwhelming.

Comment #21: l.166 : there are no images to appreciate the effect, is this the case in confined cells?

Authors' Response: Yes, we confirm that it is in confined cells. Both the relevant text and figure legend have been updated to clarify this, and representative images have now been included in the revised manuscript (**Extended Data Fig. 4k**).

Comment #23: The analysis of the Arp3/FA phenotypes are lacking a strong conclusion and a transition to the following paragraph might ease the reader to get the rationale.

Authors' Response: We thank the reviewer for this important suggestion. The text in the revised manuscript has been modified following the reviewer's remark (please see **Lines: 153-155**).

Comment #24: Extended Fig 5d was more relevant to the general model being demonstrated than Fig 2e as the first one concerns cells in confinement and the second cells simply in a 2D culture configuration. I would suggest exchanging these 2 panels.

Authors' Response: Following the reviewer's suggestion, we switched the position of the relevant panels in our revised manuscript.

Comment #25: Along the same line, Fig 2g and Extended Fig 5g seem to be the exact same dataset but with the y axis being "speed" and the other "velocity". Unless there is a clear reason for that, which should appear in the captions of the panels, there is no need for data duplication.

Authors' Response: Yes, these are the same dataset. Following the reviewer's remark, the "old" Extended Data Fig. 5g have been removed from our revised manuscript.

Comment #26: Major comment: At this stage, it remains unclear what links the extracellular viscosity to increased actin remodeling? How specific to migration/protrusive structures is such an effect?

Authors' Response: We respectfully refer the reviewer to our detailed responses to your comments #2 and #4 above.

Comment #27: I.180 : Why looking only at NHE1 ? Why not assessing other channels?

Authors' Response: Prior work based on Osmotic Engine Model (Stroka et al. 2014) has shown that NHE1 is the major ion transporter involved in confined cell migration. Importantly, we herein show that viscosity-mediated cell swelling (**Fig. 2e and Extended Data Fig. 5d**) {REDACTED} are completely abolished by NHE1, suggesting its prominent role in this process. Regarding the participation of other ion channels, we used either shRNA or pharmacological tools to test the potential contributions of Piezo1, Piezo2, TRPC1, TRPC3, TRPC5, TRPC6, TRPV6, TRPM3 and TRPM7, but according to our results (**Extended Data Fig. 5n-q**) none of them participates in the response to high viscous solutions. Only TRPV4 is critical to this process.

Comment #28: I.203 : Have the authors considered looking at membrane tension? Activation of stretch-activated channels?

Authors' Response: We thank the reviewer for their insightful suggestion. In our revised manuscript, we used quantitative FLIM microscopy and Flipper-TR to demonstrate that membrane tension is increased upon acute cell exposure to elevated viscosity (**new Fig. 2g,h**). TRPV4 silencing fails to suppress the viscosity-mediated increase in membrane tension (**new Fig. 2h**). In marked contrast, inhibition of Arp2/3 or NHE1 function abrogate the viscosity-mediated increase in membrane tension (**new Fig. 2h**) as well as the downstream calcium flashes (**Fig. 2i and new Fig. 2n**). Collectively, these data reveal the role of actin remodeling and NHE1-mediated cell volume increase in triggering TRPV4-mediated calcium flashes mediated by an increase in membrane tension.

We also wish to point out the central role of TRPV4 in this process, as inhibition of other TRP channels (TRPC1, TRPC3, TRPC5, TRPC6, TRPV6, TRPM3, TRPM7, TRPM8 and TRPP2) using 2 APB had no effect in confined cell migration at 8 cP.

Comment #29: I.211 : channels are restricted to MDA-231 cells, have you considered other channels that would not be 231-specific ?

Authors' Response: We wish to stress that none of these channels are specific for MDA-MB-231 cells. In fact, they are expressed almost ubiquitously throughout the body. The rationale to select these channels is that they are the main mechano/osmosensitive ion channels involved in the generation of Ca^{2+} signals. In the revised version of our manuscript, we show that HOS, SUM159 and **U87 glioblastoma cells** also respond to elevated viscosity via a TRPV4-dependent pathway (**new Extended Data Fig. 5w**). We further include new data showing that U87 glioblastoma cells activate TRPV4 in response to elevated viscosity (**Fig. 1.4**). We chose not to include **Fig. 1.3** in our revised manuscript because it is already rich in data, and inclusion of additional data would render this article overwhelming.

Comment #30: I.219-222 : TRPV4 is the key MIC activated by viscosity – does it apply to other cells types than SUM159 and BrM2 ?

Authors' Response: As stated in our response to **comment #29** above, **U87 glioblastoma cells** activate TRPV4 in response to high viscosity (**Fig. 1.4**). We have also generated new data showing that TRPV4 inhibition blocks the motility of all tested cancer cell types at elevated (8 cP), but not baseline, viscosities (**new Extended Data Fig. 5w**). This finding holds true not only for MDA-MB-231, SUM159 and BrM2 cells but also for **human osteosarcoma (HOS)** and **U87 glioblastoma cells**. Taken together, our data point to TRPV4 as the main cation channel activated by high viscosity in different cell types.

Comment #31: What is the effect of TRPV4 depletion on actin remodeling?

Authors' Response: Following the reviewer's important suggestion, we performed new experiments using STORM imaging and now show that TRPV4 silencing has no effect on actin remodeling (**new Extended Data Fig. 6j**). These new data are consistent with our signaling model, which has identified actin network remodeling upstream of TRPV4 activation.

Comment #32: I.226 : "Taken together, we postulate that elevated viscosity promotes NHE1-dependent cell swelling, which leads to calcium influx through the MIC TRPV4" -again, what is the viscosity sensing mechanism ? Why would TRPV4 only act downstream of NHE1? Has NHE1 silencing been tested in other cell lines?

Authors' Response: We respectfully refer the reviewer to our detailed responses to **comments #4** and **#31** above. In brief, the actin network **is the sensor of extracellular fluid viscosity**. Arp2/3-dependent actin loading in response to elevated viscosity triggers NHE1-dependent swelling, which increases membrane tension, activates TRPV4 and enhances calcium influx that leads to increased RhoA-dependent cell contractility.

Inhibition of Arp2/3 function abrogates viscosity-mediated volume increase (**new Extended Data Fig. 6i**), membrane tension increase (**new Fig. 2h**) and TRPV4-dependent calcium spikes in cells (**new Fig. 2n**), providing further support to the concept that the actin network is the sensor of extracellular fluid viscosity.

Consistent with the concept that the branched actin network is upstream of TRPV4 activation, new STORM imaging data reveal that TRPV4 silencing has no effect on actin remodeling at elevated viscosities (**new Extended Data Fig. 6j**). Furthermore, TRPV4 knockdown prevents channel activation without altering cell volume (**new Extended Data Fig. 6e**) or membrane tension (**new Fig. 2h**).

As an orthogonal approach, we now show that cell swelling, caused by hypotonic media, also induces calcium flashes in cells. Such flashes were suppressed in shTRPV4 cells (**new Extended Data Fig. 6h**).

We further demonstrate that NHE1 inhibition blocks the viscosity-induced cell volume (**Fig. 2e, Extended Data Fig. 5d, 6e**) and membrane tension increases (**new Figs. 2h**) as well as TRPV4-dependent calcium spikes (**Figs. 2i**).

Collectively, we demonstrate that the branched actin network acts as a mechanical sensor of extracellular fluid viscosity, which initiates a cascade of events. In this cascade of events, NHE1 mediates cell swelling, which leads to an increase in membrane tension that triggers TRPV4 activation and calcium influx, resulting in RhoA-dependent cell contractility and enhanced cell motility.

In response to the reviewer's suggestion, we generated new data showing that NHE1 inhibition markedly suppresses the motility of BrM2 and SUM159 cells at elevated viscosities (**new Extended Data Fig. 5h**).

Comment #33: I.248 : it is not clear to this reviewer why would cells require to first increase their volume in response to viscosity and then activate membrane channels ?

Authors' Response: We respectfully refer the reviewer to our detailed responses to **comments #4, #31** and **#32** above. Moreover, we wish to point out that in light of our background and published work on mechanosensitive ion channels (e.g., Zhao et al., *Science Advances* 2019,

2021a, 2021b), we initially thought of TRPV4 as one of the first responding molecules under elevated viscous loads. However, as data were generated, we clearly saw that NHE1 was upstream of TRPV4 activation. Knockdown of NHE1 or Arp2/3 blocked the viscosity-induced increases in cell volume (**Fig. 2e, Extended Data Figs. 5d, 6e, new Extended Data Fig. 6i**) and membrane tension (**new Fig. 2h**) as well as the activation of TRPV4 (**Fig. 2i and new Fig. 2n**), whereas knockdown of TRPV4 prevented channel activation without altering actin branching (**new Extended Data Fig. 6j**), membrane tension (**new Fig. 2h**) or cell volume (**Extended Data Fig. 6e**). Altogether, we herein demonstrate that actin reorganization acts as a mechanical sensor of extracellular fluid viscosity, which initiates a cascade of events. Specifically, Arp2/3-dependent actin loading in response to elevated viscosity triggers NHE1-dependent swelling, which leads to an increase in membrane tension that triggers TRPV4 activation and calcium influx, thereby resulting in RhoA-dependent cell contractility and enhanced cell motility.

Comment #34: L260-263 : the authors postulate from ezrin inhibition that block Ca²⁺ flashes at 8cP and phenocopies siNHE1 that elevated extracellular viscosity leads to the formation of a denser actin network at the leading edge. If true, there should be a polarization of ezrin toward the edge as cell migrate in 8cP medium ? Is this the case?

Authors' Response: Yes, we confirm that we see higher polarization of ezrin at the leading edges of cells at 8 cP, as shown in **Fig. 2o, p**.

Comment #35: This should lead the a polarized entry of Ca²⁺ mainly from the leading edge? Is that the case? Authors might consider using more subtil Ca²⁺ biosensors such as PKC β -C2 for local Ca²⁺ entry instead of GCAMP which only highlights global Ca²⁺ response,

Authors' Response: The reviewer is correct in their appreciation that, most likely there is a polarized entry of Ca²⁺. We have previously reported that the Ca²⁺ signal generated by TRPV4 can either act locally (Doñate-Macian *Nat Comm* 2018) or away from where it is generated (Fernández-Fernandez et al *Pflugers Arch* 2008). A relevant statement has been added to our revised manuscript (please see **Lines: 207-211**).

Comment #36: In Fig.2n is it epifluorescence or confocal microscopy? If epifluorescence, how can the authors rule out artifactual accumulation from membrane bending?

Authors' Response: Old Fig. 2n is now labeled as **Fig. 2o**, and it is a confocal image. We have now revised the figure legends to indicate this.

Comment #37: The authors have shown that NMIIA is required but not NMIIB. What about NMIIC ? This should be tested as NMIIC was shown to (see Surcel et al., PNAS 2015 and Cancer Research 2019). This is particularly relevant in a context of cancer cell invasion and metastasis.

Authors' Response: We appreciate the reviewer's insightful remark about NMIIC. Please note that qPCR evaluation of the three myosin isoforms reveals that the RNA levels of NMIIC are >7000 times lower than those of NMIIA (**new Extended Data Fig. 7j**). Thus, we predict that the potential contribution of NMIIC in our system would be negligible, if any, due to such low abundance of NMIIC. A relevant statement has made in our revised manuscript (**Lines 291-292**).

Comment #38: In Extended Fig 8e, the authors used Latrunculin A, a known actin depolymerizing agent, which halts cell migration without affecting NHE1 polarization. They conclude that the OEM is not sufficient to support cell migration. However, how would the authors explain that NHE1 polarization which relies on a denser actin network at the front is not affected by LatA, while the rest of the actin cytoskeleton required for cell migration is? This dataset is a major counter-argument against the model proposed: Extracellular Viscosity → denser actin network → Ezrin/NHE polarization → TRPV4 activation This should be clarified : How is Ezrin polarized in such situation ?

Authors' Response: As shown by Stroka et al. in *Cell* (2014), NHE1 polarization, once established in confinement at a baseline viscosity, is not lost after LatA treatment. However, LatA treatment prevents NHE1 repolarization to the opposite pole in response to a hypotonic shock (Stroka et al. in *Cell*, 2014). New data with NH₄Cl assay indicate that NHE1 activity is unaltered after cell treatment with LatA (2 μM) following cell exposure to 8 cP (**new Extended Data Fig. 8f**). Similarly to data acquired with NHE1 (*Cell*, 2014), we herein show that LatA treatment does not alter ezrin polarization (**new Extended Data Fig. 8e**).

Comment #39: While I understand that a drug mediated approach is tricky given that the upstream and downstream effectors rely on actin, a drug independent approach might be more relevant: for instance using laser ablation on the denser actin network (using a relevant marker such as LifeAct) at the front as the cell starts polarizing.

Authors' Response: We have published data using laser ablation (Mistriotis et al. *J Cell Bio* 2019), and we can assure you that it is not only low throughput but also causes complete rupture of the cell cortex. Instead, we now provide new data with a wide range of Arp2/3 inhibitor concentrations (1-100 μM) showing their effect on TRPV4-mediated calcium signaling (**new Fig. 2n**).

Comment #40: Major comment, I.344 : The model is mostly based on migrating cells with a leading edge – would such effect apply to cells that are subjected to viscosity as this happens within a tumor mass (no leading edge, no protrusive phenotype, etc..) -> what would be the trigger then? What is the viscosity-sensing mechanism?

Authors' Response: We respectfully refer the reviewer to our detailed response to **comment #13** above where 3D spheroids (which recapitulate aspects of a primary tumor mass) were kept on collagen-coated dishes and submerged in media of specified viscosities. As such, the medium used in these assays exposes cells in the spheroids (especially, the cells in the periphery of the spheroids) to specified viscosities. Our data reveal that elevated extracellular fluid viscosity increases tumor cell dissemination from 3D tumor spheroids (**Fig. 1d,e**), which is in accord with our migration data on 2D surfaces and inside confining microchannels. Consistent with our signaling pathway (**Fig. 3I**), TRPV4 inhibition markedly suppresses cell dissemination from 3D spheroids (**Fig. 2m**). Lastly, we respectfully note that motile cells, such as invading tumor cells, display front-to-rear polarity. As such, in many cases *in vivo*, including cell entry into a confining microenvironment or during the intravasation process, the cell front represents the edge which is first exposed to a biochemical or biophysical cue.

Comment #41: I.358 : cells that are exposed for 6 days to medium with tuned viscosities very

likely massively tune their transcriptional program in response to it – have the authors considered such effect? Have they looked at expression levels of the targets (TRPV4, etc..) ? Plus, the whole concept of memory plays against the instantaneous mechanical loading concept – how can the authors reconcile such an effect? Ideally, the authors should consider recapitulating most of their approaches (actin, etc..) in cells exposed for longer times...

Authors' Response: We regret to say that we do not entirely agree with the reviewer's remark that "*the whole concept of memory plays against the instantaneous mechanical loading concept*". It may be that exposure to elevated viscosity triggers both short- and long-term cellular responses, which may be the result of the same or different effector molecules. It is well documented that mechanobiological memory may modify disease progression (PMID: 21508987). However, what is crystal clear from our experiments is that both short- and long-term responses share a common, key element: TRPV4. Please see **Lines 354-376** in our revised manuscript.

We herein demonstrate that TRPV4-depleted cells fail to form viscous memory *in vitro* and display reduced metastatic potential *in vivo*. Although TRPV4 is not the most upstream molecule in the signaling cascade of viscosity-mediated events (**Fig. 3I**), TRPV4 is the molecule which converts the biophysical signal from actin loading and NHE1-dependent cell volume regulation in to a biochemical response via calcium signaling.

{REDACTED}.

Again, given how rich in data our manuscript is, we strongly feel that inclusion of additional data would render this article overwhelming.

Comment #42: Major comment: The authors aim at studying the migration of arrested cancer cells within the ISVs as a model of cancer cells migrating in the stroma (around the primary tumor or after extravasation). How could hematogenous dissemination model anything similar to such situation (described in fig.1-3), with shear stress that are several orders or magnitude higher in the vasculature compared to any interstitial fluid? How would the authors discriminate between a migrating cell and a cell arresting by vessel occlusion and deforming as blood flow is pushing the back of the cell. This is quite difficult to judge from still image, but this jiggling is what is observed with occluded cells pushed by high shear stress blood flow.

More specifically

- I.366 : what is the relevance of looking at intravascular migration of cells? Is this specific to ISVs? What is the effect of long-term viscosity on cell mechanics? Have the authors considered subjecting cells to tuned intravascular viscosities, which is doable in ZF. What about extravasation in that model? Metastasis? Other vascular regions? Finally, are the authors really injecting cells in the dorsal aorta rather than in the duct of Cuvier ?

Authors' Response: The reviewer has made a very similar remark before. In brief, we wish to respectfully convey that in this manuscript, we wished to extend our *in vitro* findings on cell migration to the *in vivo* setting. We chose the zebrafish model and ISVs simply because their dimensions are of the same order as our microchannels *in vitro*. Thus, the zebrafish model enabled us to **extend our *in vitro* findings** on the enhanced motility in response to cell pre-conditioning at elevated viscosities (8 cP versus 0.77 cP) to **the *in vivo* setting**. Using a mouse model, we demonstrated that cell pre-conditioning to elevated viscosities (8 cP versus 0.77 cP) results in markedly elevated experimental metastasis that is dependent on TRPV4 expression. During this process, tumor cells are subjected to the shear stresses of the mouse vasculature. However, the shear stress stimulus is rather irrelevant. This manuscript identifies extracellular fluid viscosity as a novel, physiologically relevant biophysical cue, which can alter cell migration and metastasis *in vivo*. Regarding tumor cell extravasation, we employed a third *in vivo* model,

the **chick embryo model**, to establish that cell pre-conditioning at 3 cP versus 0.77 cP leads to elevated tumor cell extravasation (**new Extended Data Fig. 9e**).

Given how rich in data our manuscript is, we strongly feel that inclusion of additional data would render this article overwhelming, and would not add anything significant.

Comment #43: - l;374 : again, it is of utmost importance to assess transcriptional regulation imposed by long-term viscosity.

Authors' Response: Following the reviewer's suggestion, we performed RNAseq experiments, which are now included in **new Extended Data Fig. 10a-d**.

Comment #44: - l.380 : "These data are consistent with the higher migratory/invasive potential of pre-conditioned cells observed in vitro and in zebrafish". How can the authors link both phenotypes? I don't see the consistency here.

Authors' Response: The increased migration observed in 2D and confining microchannels *in vitro* as well as in zebrafish ISVs *in vivo* are consistent with increased tumor cell extravasation observed in chick embryos (**new Extended Data Fig. 9e**) and lung colonization in mice. All these data consistently point towards a link between higher migratory phenotype and invasive potential, which has been established by us (Yankaskas et al., *Nature Biomedical Engineering*, 2019) and numerous others.

Comment #45: - l.396 : the extension to a pro-metastatic phenotype is based on experiments that drastically differ from the 1st part of the Ms – instantaneous effect should be considered and assessed over invasion, growth and intravasation mechanisms. As for metastasis, since this is a long-term effect, either the authors carefully assess each phenotype (Fig.1-3) or they try to tune viscosity of vascular fluids or they assess transcriptomic regulation imposed by viscosity.

- The authors use mouse experimental metastasis with tail vein injection to monitor lung colonization. While the data suggesting a cytoskeletal memory effect seems quite convincing at first glance, the relevance of the experimental model is high questionable and severely tones down the validity of the data. The blood flow dissemination, and subsequent outgrowth, drastically differs from stroma invasion (with shear stress that are several orders of magnitude higher). Furthermore, it is now well demonstrated that after extravasation, cancer cells will colonize niche that are nearby blood vessels with little need for high migration capacities. Is there any difference in cell division abilities in the lungs observed in 0.77 vs 8cP pre-conditioned cells ?

Authors' Response: We wish to respectfully refer the reviewer to our detailed responses to their comments #3, #6, #7, #11, #13, #41 and #42 above.

Regarding the reviewer's new question about the potential effects of cell division, we wish to note that besides the 3-week timepoint in which we assessed distant colonization, we used the 48 h timepoint to minimize cell division-related factors. At 48 h, pre-conditioning of tumor cells to elevated viscosity results in markedly increased lung colonization. Furthermore, new data from intravital imaging in a chick embryo model (**new Extended Data Fig. 9e**) clearly reveal that single cell extravasation is increased.

Comment #46: - The use of actin as a normalizing control in HLINE experiments might not be the most relevant as extracellular pressure acts on actin cytoskeleton and there is no data supporting the unaltered expression of actin in response the 8cP culture in the current manuscript.

Authors' Response: We wish to respectfully point out that only DNA was isolated from tissue specimens, and the actin primer used identifies DNA encoding actin in both mouse and human cells. As such, these quantities are independent of mRNA levels. Importantly, the bulk of the DNA is comprised of mouse DNA which constitutes the cells from the lung parenchyma of the mouse which were never experimentally subjected to elevated viscosities. Moreover, new RNAseq data analysis reveals that actin mRNA levels remain unchanged in all groups tested. Lastly, *h*LINE results are in line with the data obtained from quantification of cells colonizing the lung tissue as assessed by confocal imaging.

Additional comments :

Comment #47: - Overall while there are many data quantified, relevant representative images are missing so the reader can appreciate the change in phenotypes.

Authors' Response: We have modified the text and the figures keeping this in mind. Importantly, new images have been added to our revised manuscript (please see **new Extended Data Figs. 4k, 6j, 9e** and **new Fig. 2g**).

Comment #48: - All data should be displayed with SD, if the dataset successfully passed relevant statistical tests based on raw values, then it should not matter and it would be statically cleaner.

Authors' Response: We have followed the reviewer's advice and all data are now displayed with SD except for those experiments where the population average per individual experiment is plotted.

Comment #49: - Microscopy modalities used should be written in the caption to appreciate data quality.

Authors' Response: We thank the reviewer for this important suggestion. The text in the revised manuscript has been modified following the reviewer's remark.

Comment #50: - SC stands very likely for scramble control but is never defined. It should be properly written at least in figure captions.

Authors' Response: Apologies for the oversight. The text has been modified in the revised manuscript following the reviewer's suggestion.

Comment #51: - MICs/OEM should be defined

Authors' Response: The text has been modified in the revised manuscript following the reviewer's suggestion.

REFEREE #2:

The manuscript “Counterintuitive effect of viscosity on enhancing motility and metastasis” by Bera, Kiepas, Godet, Li, Mehta, Ifemembi, Paul, Sen, Serra, Lee, Zhang, Boen, Mistriotis, Gilkes, Tanner, Valverde, Sun, and Konstantopolous is an elegant and rigorous exploration of a novel phenomena driving cellular behavior in response to increased extracellular fluid viscosity. Striking experiments on 2D surfaces and in 3D confining microchannels demonstrate that cells respond dynamically and robustly to extracellular fluid viscosity by, among other responses, increasing their migration speed, migration mode, lamellipodia growth, and tumor escape. This is found to be modulated by ARP2/3 activity, NHE1-dependent activation of the ion channel TRPV4, and subsequent RhoA signaling-induced contractility. The authors extend these findings in vivo by analyzing the migration of pre-treated cancer cells in both zebrafish and mouse models.

Overall, this work is likely of immediate interest to a wide swath of the cell biology community, and of urgent interest to the mechanobiology community. Observations on mechanical memory of fluid viscosity has the potential to provide new avenues of mechanical interventions into cell behavior that can be explored in a number of applications, as the authors touch upon in the discussion. In the context of this work, the authors have provided ample evidence, in painstaking detail, for the mechanism driving this novel phenomena. It is also of note that they have demonstrated the physiological relevance of fluid viscosity in two separate model organisms, further underscoring the universal nature of their findings. Some small suggestions are made below, most of which can be addressed in the discussion.

Authors’ response: We thank the reviewer for their enthusiasm and insightful remarks, which helped us improve our manuscript. Below, as well as in our revised manuscript, we address each of their remarks.

Minor revisions/comments

Comment #1: An interesting emerging observation in cancer mechanobiology, usually in the context of YAP translocation or tropomyosin activation, has been that many cancer cells become less mechanosensitive to their surroundings. Here, every cancer line presented is strongly sensitive to viscosity. Have the authors found any cell lines that are not susceptible to viscosity changes? Tangential to this, the hAOSMCs were exposed to 9.5 cP instead of 8 cP, suggesting that different cell lines have different threshold levels of viscosity response (for 231s in Fig. 1A this appears to be 3 cP). Does this threshold have any tissue- or disease-specific physiological relevance?

Authors’ Response: In our manuscript, we tested seven (7) different cancerous and non-cancerous cell types. While all cell lines responded to elevated viscous loads with increased migration velocity, we are aware of the different reports claiming that cancer cells become less sensitive to their surroundings. In fact, we have also recently described this kind of attenuation to fluid shear stress (i.e., HT1080 fibrosarcoma cells exhibit reduced TRPM7 expression and activation compared to fibroblasts, which renders them less sensitive to fluid shear; Yankaskas et. al., *Science Advances* 2021).

In response to the reviewer’s question, we generated new data showing that human fibroblast cells exposed to elevated (3 cP) relative to baseline (0.77 cP) viscosity also migrate faster (**Fig. 2.1; please see next page**). At this point, we cannot tell if there is a cell type insensitive to high viscous loads. Please note that we chose not to include **Fig. 2.1** in our revised manuscript because it is already rich in data, and inclusion of additional data would render this article overwhelming. We hope the reviewer agree with our decision.

Fig. 2.1: Migration speeds of human fibroblasts in confining channels at 0.77 and 3 cP. Each red line represents the median obtained from ≥ 83 cells from 2 independent experiments.

Comment #2: Several works have speculated that cell migration in confinement can be related to the total mass of fluid the cell needs to displace through the exit reservoir (which normally assumes an impermeable seal between cell and channel wall). In this case, by increasing the viscosity, the total mass of the fluid is also increasing (although only negligibly- the 0.6% methylcellulose solution would only increase the fluid density by 0.6%!). Can these two models be united? This is briefly touched upon in the discussion. The total fluid mass to be ‘pushed’ by the cell only increases to a small degree here, but the cellular response is quite robust. Does this imply that fluid viscosity is far more important than simple mass?

Authors’ Response: The reviewer raises an insightful point. Indeed, we believe that these two models work together. To this end, we refer you to **Lines 157-158**: “In confinement, tumor cells migrate by taking up and discharging water at the cell poles, and also by pushing the column of water ahead of them, which generates hydraulic resistance. Elevated extracellular viscosity increases the hydraulic resistance at the leading edge of confined cells. The two-phase model of cell migration, which treats the cytosol that is essentially water and the actin network as two fluid phases interacting with each other, predicts that high hydraulic resistance promotes cell migration via water uptake”. Accordingly, we demonstrate that hydraulic resistance due to elevated extracellular fluid viscosity results in increased cell volume (**Fig. 2e, Extended Data Fig. 5d, 6e**), presumably via water uptake as evidenced by NHE1 depletion or inhibition, which is a key constituent of the osmotic engine model.

Comment #3: Numerous assays in this work reveal a higher degree of cell polarization in confinement in response to elevated viscosity, including the distribution of myosin II, ARP2/3 localization, and RhoA activity. Accordingly, was an increase in directionality (i.e. a reduction in ‘reversals’ of cells in confined channels) observed in high viscosity environments? (This is shown in EFig 9 in zebrafish- just curious if it was also observed in vitro) The denser actin network and lamellipodial stability would also theoretically contribute to this.

Authors’ Response: Please note that the persistence or directionality of cells is very high in confining (3.5 μm narrow) channels even at baseline (0.77 cP) viscosity. Nevertheless, it is further increased with elevated (8 cP) viscosity (**Fig. 2.2**). We chose not to include **Fig. 2.2** in our revised manuscript since we feel that inclusion of this additional data would render this article overwhelming.

Fig. 2.2: Persistence of MDA-MB-231 cells at prescribed viscosities in confining channels. Data are mean \pm SD for >150 cells from 2 independent experiments.

Comment #4: In general, a framework is presented in which at high viscosities, bleb-based migration and its supporting signaling pathways give way to mesenchymal migration. This is true for the majority of cells, but Efig 2b it appears that ~20% of migrating cells still utilize blebs. Of course it can be challenging to binarize a process that can exhibit some degree of overlap, but I'm wondering if the blebs present in high viscosity environments are morphologically different than those at physiological viscosity.

Authors' Response: The reviewer is correct in their speculation. Data analysis reveals that the blebs detected at elevated viscosities are smaller than those at 0.77 cP (**Fig. 2.3**). This difference can be attributed to the denser actin cortex and higher cell membrane tension (**new Fig. 2g,h**) observed at elevated relative to baseline viscosities. Again, we chose not to include **Fig. 2.3** in our revised manuscript since we feel that inclusion of these additional data would render this article overwhelming.

Fig. 2.3: Blebbing regions (highlighted in red in the bottom image) were manually segmented to compare their cumulative areas to the total cell area of each Lifeact-GFP-tagged MDA-MB-231 cell migrating in confinement at 0.77 or 8 cP. Data are mean \pm SD from ≥ 15 cells from 3 independent experiments.

Comment #5: Is it possible or illuminating to track the dynamics of switching from low viscosity to high viscosity by introducing high viscosity media once cells are already in the channels? If high viscosity media is supplied to cells that are already moving in a 0.77 cP channel, can you visualize the blebs disappearing?

Authors' Response: The reviewer makes an excellent suggestion. However, it is technically challenging, or to be precise unfeasible, to replace 0.77 cP medium with 8 cP in the middle of the experiment due to the difficulty in flowing this thicker medium (8 cP) through confining microchannels.

Comment #6: Related to this- from the methods section, it seems as if cells are plated into the cell-seeding channel, then allowed to enter the confining channel in the absence of any chemoattractant or flow. While it appears as if all analyses were performed on cells that had already entered the channels, do the authors have any observations on whether cells more likely to enter confinement as a function of viscosity?

Authors' Response: Following the reviewer's suggestion, we quantified cell entry times at elevated (8 cP) and baseline (0.77 cP) viscosities, and found them to be faster at 8 cP (**new**

Extended Data Fig. 1I). These data are in line with the higher levels of tumor cell extravasation observed *in vivo* (**new Extended Data Fig. 9e**). A relevant statement has made in our revised manuscript (please see **Line: 96**).

Comment #7: Given the 5-fold or greater increase in FA number shown in Efig 3f, and the slight yet significant increase in FA surface area per cell (Efig 3i), is it safe to say that in high viscosity cells, the individual focal adhesion puncta are much smaller?

Authors' Response: Yes, we indeed see a large number of small FAs at the cell front in response to elevated viscosity (**Extended Data Fig 4j**).

Comment #8: Does the model in EFig 5e take this into account?

Authors' Response: This is an excellent point. Following the reviewer's suggestion, we modified our model where the parameter η_{st} is proportional to the number of FAs and the average size of each FA throughout the cell length. Our model predictions are in excellent agreement with experimental data (**Fig. 2f** and **Extended Data Fig. 5f**).

Comment #9: To what degree do the ISVs recapitulate the confining microenvironments used in the *in vitro* work? The diameters shown in Fig. 4d are mostly around 10 μm , suggesting a cross-sectional area closer to the 10 μm wide channels than the 3.5 μm wide channels. Two cell lines were analyzed in 10 μm channels, but do the MDA-MB-231 cells also show this enhanced speed in 10 μm channels after viscosity pretreatment?

Authors' Response: We agree with the reviewer that the diameters of the ISVs in the zebrafish model are wider than our confining channels. Because elevated extracellular fluid viscosity promotes faster migration in a wide array of experimental models (i.e., wound healing assay, 2D migration, confined migration and cell dissociation from 3D spheroids), there is no doubt in our mind that this would be the case in wider channels *in vitro*, as shown in the *in vivo* setting.

Edit suggestions:

Comment #10: Although this group's work on microchannels is widely known in the field, a diagram of the microchannel chip setup might be helpful to readers less familiar with microchannel confinement strategies.

Authors' Response: We have included a schematic of the microfluidic device in our revised manuscript (please see **Extended Data Fig. 1c**).

Comment #11: The word 'viscosity' in the title is potentially confusing, as a number of works have used this simple term to describe substrate viscoelasticity (e.g. Charrier/Janmey et al Nat. Comm. 2018). Perhaps 'fluid viscosity' would be more descriptive without being too wordy.

Authors' Response: We appreciate the reviewer's excellent point. The title of our revised manuscript has been appropriately revised.

Comment #12: In Efig 3a, I'm having trouble understanding what the red triangle represents, although it's referred to as 'instant of viscosity applied'. Does this imply that the X axis supposed to be time?

Authors' Response: We wish to note that the X axis indicates the distance that filaments grew over time. To reach steady state, the simulation was allowed to run for a time interval (6 s) during which the network grew only against membrane tension from 0 to ~1250 nm. From that point onwards, indicated by the red arrowheads, the network grew against membrane tension plus viscous forces. We have clarified this point in the figure legend of our revised manuscript.

Comment #13: The model in Fig 2f appears to be an excellent prediction of the data shown in Fig 2g. Putting the axes on the same range would emphasize this (or even combining them into a single graph!)

Authors' Response: We have implemented the reviewer's suggestion in our revised manuscript (please see **Fig. 2f**).

Comment #14: Presumably OEM refers to Osmotic Engine Model, but this is not introduced as an acronym prior to its use on page 11.

Authors' Response: We have modified the relevant statements in our revised manuscript following the reviewer's suggestion.

Comment #15: Page 9 Line 256: I think it's meant to say 'Fig. 2o' instead of 'Fig. 2n'

Authors' Response: We thank the reviewer for catching this typo. We have ensured that all figure panels are referenced correctly in our revised manuscript.

Comment #16: EFig 5e-f could be labelled FA and Actin in the figure for clarity.

Authors' Response: This change has been made in our revised manuscript.

Comment #17: Typo on Page 18 line 559: Liefact vs. Lifeact

Authors' Response: We thank the reviewer for catching this typo, which has now been corrected in our revised manuscript.

Comment #18: Typo in the axis of Efig 2J

Authors' Response: This typo has been fixed. In closing, we thank the reviewer for their meticulous reading and insightful suggestions.

REFEREE #3:

A. Kiepas, I. Godet, Yizeng Li, P. Mehta and colleagues, a strong group of investigators led by Dr Konstantopoulos of Johns Hopkins, report **novel and interesting findings** on how extracellular viscosity can enhance cells' motility and metastatic capacity of malignantly transformed cells. **These results are novel and interesting, at a general level of interest**, yet underdeveloped in some aspects. Results and putting-it-all-together are curated well, but in need of some fixing and clarifying.

The cellular machinery of the cell lines examined is clarified reasonably well. A central element identified is the TRPV4 ion channel. There are several TRPV4-related aspects that the authors need to address by conducting additional experiments, which will also benefit the entire mechanistic contribution of the paper, which is **overall convincing**.

Authors' response: We thank the reviewer for their enthusiasm and insightful remarks, which helped us improve our manuscript. Below, as well as in our revised manuscript, we address each of their remarks.

Comment # 1a. 1) TRPV4 has been examined in the context of tumorigenesis and metastasis, with one authoritative review here Yu et al. Cell Death and Disease (2019) 10:497 <https://doi.org/10.1038/s41419-019-1708-9> Fig 2 of this paper highlights an important property of TRPV4 in human cancers. Whereas most examined cancers show increased expression of TRPV4, the common cancers, prostate adenocarcinoma and esophagus carcinoma have significantly reduced TRPV4 expression. There are a number of cell lines that need to be included in a revised Fig. 1b (see below), but these two particular types of cancer do need to be examined, whether they follow the logic the authors have determined, or whether there is a non-universal role for TRPV4, depending on tumor type.

Authors' response: We read with interest the review that you brought to our attention by Yu et. al., which claims that TRPV4 expression is reduced in two cancers (prostate adenocarcinoma and esophagus carcinoma) based on data obtained from oncomine (Fig. 2 in their article). Interestingly, the papers cited in their Table 3, which supposedly support this observation, do not show any data related to TRPV4 expression. In contrast to the claim by Yu et. al., there are other reviews that claim TRPV4 is increased in esophageal cancer (PMID: 32182937). Considering the discrepancies regarding TRPV4 expression in those cancers, and the very high workload to address the numerous, but very interesting, points of all 4 reviewers, we decided not to perform experiments with these two specific types of cancer cells.

Comment # 1b. Glioma should also be studied <https://pubmed.ncbi.nlm.nih.gov/29928875/> because migration and invasiveness are particularly prominent features of gliomas. What is important in the glioma context is that they rather express AQP4, not AQP5. Understanding glioma cells' migratory capacity in extracellular matrix of watery vs increased viscosity will help us understand whether the TRPV4-dependent response to increased viscosity has to be coupled to AQP5, or whether other AQPs can fill in as well.

Authors' response: We have followed the reviewer's very interesting suggestion and performed experiments with **U87 glioblastoma cells**. We demonstrate that elevated fluid viscosity (8 cP) generates TRPV4-mediated increases in intracellular Ca^{2+} in U87 cells (**Fig. 3.1**) and increases their migration speed in a TRPV4-dependent manner (**New Extended Data Fig. 5w**). We chose

not to include **Fig. 3.1** in our revised manuscript because it is already rich in data, and inclusion of additional data would render this article overwhelming.

Comment # 2. TRPV4 has been established as a mechanotransducer in endothelial cells that transduces within 4msec after the onset of shear stress (Don Ingber's group), also by recent diligently conducted studies from Dr Kate Poole's lab - these studies need to be referenced and taken into account experimentally.

<https://pubmed.ncbi.nlm.nih.gov/20725677/>

<https://pubmed.ncbi.nlm.nih.gov/28135189/>

<https://pubmed.ncbi.nlm.nih.gov/30984749/>

<https://pubmed.ncbi.nlm.nih.gov/33537292/>

Questions arise to the contribution to TRPV4 functioning of beta1 integrins, CD98, and also vinculin, and they need to be addressed by the authors. TRPV4 – vinculin
<https://pubmed.ncbi.nlm.nih.gov/30674675/>

Authors' response: Following the reviewer's insightful remark, we tested the role of β 1-integrin in mediating Ca^{2+} spikes in response to elevated fluid viscosity. We have now included these new data, which show that β 1-integrin knockdown does not affect the generation of Ca^{2+} spikes in cells exposed to elevated viscosity (8 cP) (**new Extended Data Fig. 5y**). In contrast, depleting either NHE1 or TRPV4 essentially abolishes Ca^{2+} spikes (**Fig. 2i**). While the references provided superbly describe the different modes of TRPV4 activation, we hope that the reviewer will understand that it is impossible to include them due to the length limitations imposed by *Nature*. The references seem also to be beyond the scope of the current study, which describes cell response to the novel physical cue of fluid viscosity.

Comment # 3. Fig. 3I depicts TRPV4 as a swelling-activated calcium-permeable channel downstream of viscosity-mechanical sensing machinery.

Have the authors considered whether TRPV4 might be playing multiple roles in response to increased viscosity, e.g as a more direct and upstream mechanotransducer as well ? helpful ref. <https://pubmed.ncbi.nlm.nih.gov/15952033/>

Authors' response: We wish to respectfully point out that many years ago, we showed that high viscous fluids activate TRPV4 in native and cultured epithelial cells as well as in heterologous expression systems (Andrade, *J Cell Biol* 2005; Fernandes, *J Cell Biol*, 2008; Lorenzo *PNAS* 2008). TRPV4 activation by high viscosity shares many characteristics with TRPV4 activation by cell swelling under hypotonic shock (Liedtke *Cell* 2000; Arniges *J Biol Chem* 2004; Fernandez-

Fernandez Pflugers 2008). TRPV4 was critical for the initiation and maintenance of the oscillatory Ca^{2+} signals in response to high viscosity in all these cell models.

In light of our background and our recently published work on mechanosensitive ion channels (e.g., Zhao et al., *Science Advances* 2019 and 2021, Yankaskas et al., *Science Advances* 2021), we initially thought that TRPV4 is one of the first responding molecules under elevated viscous loads. However, as data were generated, we clearly saw that NHE1 was upstream of TRPV4 activation. Knockdown of NHE1 or Arp2/3 blocked viscosity-induced increases in cell volume (**Fig. 2e**, **Extended Data Fig. 5d, 6e**, **new Extended Data Fig. 6i**) and membrane tension (**new Fig. 2h**) as well as the activation of TRPV4 (**Fig. 2i** and **new Fig. 2n**), whereas knockdown of TRPV4 prevented channel activation without altering actin branching (**new Extended Data Fig. 6j**), membrane tension (**new Fig. 2h**) or cell volume (**Extended Data Fig. 6e**). We also observed no additive effect upon inhibiting both TRPV4 and NHE1. Altogether, we confirmed the model we proposed more than 10 years ago, that TRPV4 is key to oscillatory Ca^{2+} signals in response to high viscosity; however, we have now added a fundamental element to the picture. Actin reorganization acts as a mechanical sensor of extracellular fluid viscosity, which initiates a cascade of events: Arp2/3-dependent actin loading in response to elevated viscosity triggers NHE1-dependent swelling, which leads to an increase in membrane tension that triggers TRPV4 activation and calcium influx, thereby resulting in RhoA-dependent cell contractility and enhanced cell motility.

Comment # 4. Loss-of-function of TRPV4 experiments are well conducted, however, enhanced function of TRPV4 can be updated based on a recent insight.

Lysophosphatidylcholine (LPC) has been found an endogenous activator of TRPV4, via direct activation through a C-terminal binding site for the glycerophospholipid. LPC is also a relevant component of tumor microenvironments. Therefore, this should be tested. [https://www.gastrojournal.org/article/S0016-5085\(21\)00576-X/fulltext](https://www.gastrojournal.org/article/S0016-5085(21)00576-X/fulltext) In addition, TRPV4 can be sensitized by proteolytic signaling via PAR-2 activation. Tumor microenvironments do have robust proteolytic activity. This should also be accomodated experimentally. <https://pubmed.ncbi.nlm.nih.gov/17124270/>

Authors' response: We thank the reviewer for pointing out LPC as a new TRPV4 agonist. Although it is not entirely clear to us the relationship between LPC and high viscosity, following the reviewer's suggestion, we produced new data showing that LPC generates TRPV4-dependent Ca^{2+} signals in MDA-MB-231 breast cancer cells and partially recovers migration velocity in shNHE1 cells exposed to elevated viscosity (8 cP) (**Fig. 3.2, please see next page**). We believe that these data reinforce the hypothesis that TRPV4 is a key element in cell response to elevated viscosity. Due to time and human resources limitations to produce a comprehensive response to all reviewers' comments, we decided not to test other possible modulators of TRPV4 that are not directly related to high viscosity in the context of cancer biology. We also chose not to include **Fig. 3.2** in our revised manuscript because it is already rich in data, and inclusion of additional data would render this article overwhelming. We hope the reviewer agree with our decision.

Comment # 5. TRPV4 activation in response to flow has been postulated in the context of signaling via the primary cilium.

PMIDs

25143588

19790068

18719094

18024594

15753126

The authors need to explore whether primary cilia are (critically) relevant for tumor cells' response to increased viscosity.

Authors' response: The reviewer raises an important point. However, we would like to respectfully point out that there are several papers claiming that cancer cells lose their primary cilium, including MDA-MB-231 cells:

<https://www.ncbi.nlm.nih.gov/pmc/articles/PMC2942739/>

https://faseb.onlinelibrary.wiley.com/doi/abs/10.1096/fasebj.2019.33.1_supplement.815.7

<https://pubmed.ncbi.nlm.nih.gov/25997440/>

Comment # 6. re link TRPV4-RhoA signaling, Dr Charlotte Sumner's work (out of Hopkins - !) needs to be referenced <https://pubmed.ncbi.nlm.nih.gov/33664271/> The following paper also needs to be referenced and explicitly commented on <https://pubmed.ncbi.nlm.nih.gov/33230171/>

Authors' Response: We have cited these articles in our revised manuscript (please see Line 262).

- EC fluid viscosity also important for inflammation, cellular motility in development

Authors' Response: We have modified our Introduction to include the reviewer's insightful remark (please see **Line 51**).

- intro line 64 "holistic" - inappropriate choice

Authors' Response: This wording has been deleted from our revised manuscript.

- stay at a general level of understanding and comprehension, avoid using jargon or code-talking abbreviations:
 - What is "MIC", what is "OEM" ?
- DUANU - don't use abbreviations nobody understands.

Authors' Response: We have defined the relevant abbreviations in our revised manuscript following the reviewer's suggestion.

First sentence of the Discussion: "apodictic influence" - non-helpful starter sentence of this all-important section.

Authors' Response: This wording has been deleted from our revised manuscript.

REFEREE #4:

In this manuscript, Bera and colleagues report how an increase in medium viscosity induces cell migration and dissect the pathway that is involved in this phenomenon. They show that this effect can be observed in many transformed and untransformed cells. It involves polymerization of branched actin by the Arp2/3 complex as well as myosin-mediated contractility. It also involves mechanical transduction of viscosity through the sodium proton antiporter NHE1, cell swelling, Ca²⁺ spikes and the cation channel TRPV4. Using mouse and zebrafish xenografts, the authors show that high viscosity can be used in vitro to reprogram cells for enhanced migration and metastasis formation in vivo.

I found this paper original and interesting. To me it is a good candidate for publication in Nature. Despite its complexity, this manuscript should be of interest to a general audience for one reason or another: mechanotransduction of a new physical parameter, role in tumor cell dissemination, as a means to increase cell migration, as a way to access the gene expression program of mesenchymal motility ... Even though it is not the first time that this effect of increased viscosity is reported in the scientific literature, this paper analyzes in depth the phenomenon, which is certainly not very well known. It is part of a stream of current studies showing the importance of physical parameters for almost all cell functions and in particular cell migration. Overall the data are convincing, although some points need to be verified before I can recommend publication without reservation.

Authors' Response: We thank the reviewer for their enthusiasm and insightful remarks, which helped us improve our manuscript. Below, as well as in our revised manuscript, we address each of their remarks.

General Comments

Comment #1: Are branched actin networks the primary sensor of medium viscosity? The authors did a good job of positioning the NHE1 contribution upstream of the TRPV4 contribution (Fig.2). One can expect cortical branched actin to be further upstream, since branched actin polymerization is enhanced both in vitro and in vivo when the pushing force that is generated by branched actin encounters resistance, as it is properly discussed in the manuscript. However, this point is not experimentally addressed. It is easy for the authors to demonstrate this point with the system they have in hands. Would they still observe Ca²⁺ spikes, increase in cell volume and TRPV4 currents when Arp2/3 is depleted? If yes, do these responses depend on the dose of CK-666?

Authors' Response: Following the reviewer's suggestion, we performed a series of new experiments to conclusively establish that the branched actin network is the sensor of extracellular fluid viscosity. Specifically, inhibition of Arp2/3 function abrogates viscosity-mediated increases in cell volume (**new Extended Data Fig. 6i**), membrane tension (**new Fig. 2h**) and TRPV4-dependent calcium spikes in MDA-MB-231 cells (**new Fig. 2n**). Consistent with the concept that the branched actin network is upstream of TRPV4 activation, new STORM imaging data reveal that TRPV4 silencing has no effect on actin remodeling at elevated viscosities (**new Extended Data Fig. 6j**). Furthermore, TRPV4 knockdown prevents channel activation without altering cell volume (**new Extended Data Fig. 6e**) or membrane tension (**new Fig. 2h**). We further demonstrate that NHE1 inhibition blocks the viscosity-induced increases in cell volume, membrane tension and TRPV4-dependent calcium spikes (**new Figs. 2h, n, and Extended Data Fig. 6e**). Taken together, these data establish the branched actin network as a mechanical sensor of extracellular fluid viscosity, which initiates a cascade of events. In this cascade of events, NHE1

localization to the plasma membrane mediates cell swelling, which leads to an increase in membrane tension that triggers TRPV4 activation and calcium influx, resulting in RhoA-dependent cell contractility and enhanced cell motility.

Comment #2: This Arp2/3 dependence is particularly important, since myosin-induced contractility is also required and in the mind of most readers, mesenchymal migration based on branched actin is antagonistic to amoeboid bleb-based migration that depends on contractility. Here both pushing and pulling forces might well be required, but Arp2/3 should definitely be upstream. That would be consistent with the lamellipodium and cortical branched actin being mechanical sensors, as observed for sensing substrate stiffness or cell density in a monolayer (Oakes ... Gardel, PNAS 2018; Molinie ... Gautreau, Cell Res 2019).

Authors' Response: This insightful point has been emphasized in a relevant statement added to our revised manuscript (please see **Lines 324-326**).

Comment #3: This manuscript does not only address the required machinery. It also shows that exposure to viscous medium for 6 days commit cells into a motile status that is sustained even when high viscosity is not maintained (Fig.4). This important result shows that cell motility corresponds to a particular gene expression program that I believe would be important to characterize here by transcriptomics. **While this may seem like an unnecessary extension of the manuscript**, it would actually benefit from the detailed characterization of the conditions (pretreatment with high viscosity, TRPV4 depletion) as well as the in vivo consequences to provide a resource for the further dissection of this phenomenon that this manuscript should trigger upon publication. Molecular machines involved in the process should be well expressed, perhaps even induced by high viscosity, but not down-regulated... This can be verified for the here identified players (Arp2/3, NHE1, TRPV4), but a global transcriptomics analysis would provide one more reason to cite this study in the future.

Authors' Response: Following the reviewer's suggestion, we performed RNAseq experiments (**new Extended Data Fig. 10a-d**).

Comment #4: Organization and clarity of the manuscript can probably be improved. To me, the part on integrin beta1 is less required than the rest. It is already clear that viscosity-induced migration is of the mesenchymal type. The requirement for Arp2/3 is enough to make the point. This part can be removed if space is needed. The discussion indeed recapitulates most of the points, but it is long and does not provide much new information. To me, the discussion is not very consistent with Nature's style.

Authors' Response: We appreciate the constructive criticism of the reviewer. Although we kept the data on β 1-integrin in order to address an important point raised by Reviewer #3, we have implemented the remaining suggestions of this referee in the revised version of our manuscript.

Specific Points

Comment #5: I agree that myosin-mediated contractility is not only a means to migrate through a bleb-based mechanism, but also a mechanism by which it promotes mesenchymal migration. This point is quite critical since it explains why both branched actin and myosin mediated contractility are required, despite the well-established antagonism between mesenchymal and amoeboid types of migration. In addition to ROCK inhibition (ED Fig.7o), the authors should show

that MIIA depletion also reduces the actin retrograde flow, and emphasize this point much more in the manuscript, perhaps by including these experiments in the main figures. A seminal reference on the topic is Wilson... Theriot Nature 2010. I believe that it should be cited.

Authors' Response: The reviewer is correct in their speculation. Indeed, following the reviewer's suggestion, we performed new experiments which reveal that actin retrograde flow, but not the uninterrupted protrusion growth rate, is reduced in shMIIA cells relative to scramble controls at elevated viscosities (**new Extended Data Fig. 7p,r**). This new finding is now stressed in our revised manuscript, as suggested by the reviewer (please see **Lines 324-326**). Moreover, we have cited the suggested landmark article (please see **Line 325**).

Comment #6: This manuscript does not adhere to what has become a standard in the field. shRNA mediated depletion is not systematically validated for efficiency (mRNA levels and/or WB) and for the fact that the phenotype is not due to off-target. It leaves the room for potential wrong conclusions. At least the main phenotype should always be validated with two independent sequences.

Arp2/3: WB, targeting of 2 different subunits, so I guess it is OK.
NHE1: WB, a single shRNA sequence. Not OK.

Authors' Response: Following the reviewer's suggestion, we generated new data using two different sequences for shNHE1 (**new Extended Data Fig. 5g**). Of note, in all other experiments, NHE1 was depleted using both shNHE1 sequences, as clarified in the Materials and Methods of our revised manuscript.

Comment #7: AQP5: no WB, a single shRNA sequence. Not OK.

Authors' Response: We wish to respectfully note that the coordinated action of AQP5 and NHE1 in confined migration under baseline viscosity has been previously reported, albeit with siRNA sequences (Stroka et al. *Cell* 2014). Although we present data using a single shRNA sequence, as correctly pointed out by the reviewer, these data confirm previously published results at baseline viscosity (Stroka et al. *Cell* 2014) obtained with siRNA sequences and extend them to the high viscosity regime. As such, we believe that inclusion of these data is appropriate. If the reviewer disagrees with our assessment, we will be happy to eliminate these peripheral data from the final version of our manuscript. Unfortunately, the AQP5 antibody that we used in our *Cell* (2014) article has been discontinued. Although we tested various AQP5 antibodies from different vendors, we were unable to obtain the WBs presented by the manufacturers. That is why, we only show successful knockdown via qPCR in this manuscript.

TRPV4: mRNA levels, WB, 2 independent sequences, so it is probably good. But it is not clearly stated that one sequence was used for the main figure and the other one for the extended data.

Authors' Response: We thank the reviewer for bringing this point to our attention. This point has now been clarified in our revised manuscript.

Myosins: no WB, single sequence. Not OK.

Authors' Response: The reviewer raises an important point. However, we wish to respectfully bring to their attention that we have tested different shMIIA and shMIIB sequences in our recent publications (Zhao et. al. *Science Advances* 2019, Mistriotis et al. *J Cell Bio* 2019, and Wisniewski et al. *Science Advances* 2021). We wish to emphasize that shMIIA and shMIIB stable MDA-MB-231 cells used in this manuscript have been characterized in Zhao et. al. *Science Advances* 2019.

Comment #8: ITGB1: mRNA levels, WB, but a single shRNA sequence. Not OK. It is up to the authors to include these additional controls, probably in Extended Data, while not convoluting too much the manuscript.

Authors' Response: We have now included the shRNA sequence of ITGB1 in the Materials & Methods section of our revised manuscript. We respectfully point out that there are numerous publications where this specific sequence was used to generate shRNA-mediated knockdown of β 1-integrin (e.g., PMID: 26515603, 29904280, 26528856). The original sequence was designed using the commercial online platform provided by Dharmacon (<http://www.dharmacon.com/>) as previously described in PMID: 26515603. Given that this sequence has been previously tested and used by others, and time and human resources limitations to produce a comprehensive response to all reviewers' comments, we decided not to test another shRNA sequence.

Comment #9: Important abbreviations such as MICs or OEM are not defined.

Authors' Response: The authors thank the reviewer for pointing out these issues. We have now properly modified the text to address these points.

Comment #10: I did not assess the accuracy of statistical tests (for lack of time) and mathematical modeling (for lack of expertise).

Authors' Response: The authors verified all statistical tests and math modeling data.

Reviewer Reports on the First Revision:

Referees' comments:

Referee #1 (Remarks to the Author):

The authors have now carried on a significant number of additional experiments for which they need to be congratulated. Several of the issues that I had raised have been addressed. There are a few issues that remain, and these are mostly located in my initial comments that the authors have summarized as comment 3. These emerged when I could not agree fully with the relevance and interpretation of the experiments performed *in vivo* and when I could not understand how authors pictured that the instantaneous effects of viscosity on tumor cells (beautifully described) can be extrapolated to the very complex event of tumor metastasis which involves a long series of very complex steps (some that are fully independent of migration and some that are independent of the tumor cells themselves).

i) it remains unclear how viscosity-induced changes are similar to changes occurring in a situation of memory to viscosity linked to metastasis, and not to tumor invasion. How similar, at a transcriptional level for example, would these cells be? I appreciate the impact observed on the Hippo pathway but such effect has not been functionally tested in the long-term effects observed when looking at metastasis.

ii) it remains unclear, although I appreciated the *in vivo* validation, what "intravascular migration of tumor cells" in zebrafish tells us about metastasis and how useful this mechanism can be linked to metastasis.

iii) it remains unclear why authors have not tested metastasis in this same animal model where extravasation and metastasis can be probed easily, it thus remains unclear why the authors have added an additional experimental model (chicken) to do so. Ideally, the authors would link the intravascular migration phenotypes observed in zebrafish to later stages of metastasis (in zebrafish as well), or remove this information/model system as it is misleading and somehow irrelevant to tumor metastasis.

iv) it remains unclear why the authors focus on these steps of metastasis which, as said, has multiple likely origins, and not on tumor invasion programs that can be probed using intravital imaging. Intravital imaging of breast tumors has been done by several labs over the world (Van Rheenen, Condeelis, Timpson, etc..) and would have been more in line with the main core of the manuscript.

Referee #2 (Remarks to the Author):

The authors have comprehensively addressed all of the minor revisions and edits included in my original comments, and I appreciate the inclusion of illuminating results that would not be appropriate for the final manuscript given its length.

This reviewer remains convinced that this work will be of immediate interest to a wide swath of the cell biology community, and of urgent and remarkable interest to the mechanobiology community. Observations on mechanical memory of fluid viscosity has the potential to provide new avenues of mechanical interventions into cell behavior that can be explored in a number of applications, as the authors touch upon in the discussion. In the context of this work, the authors have provided ample evidence, in painstaking detail, for the mechanism driving this novel phenomenon. It is also of note that they have demonstrated the physiological relevance of fluid viscosity in (now three) separate model organisms, further underscoring the universal nature of their findings.

Referee #3 (Remarks to the Author):

The authors have diligently prepared a response to critique.
At my end, the ms can now move forward, my points have been addressed to my satisfaction.
This is a very strong piece of work. As such, it represents a truly new chapter - it is well-rounded and ready-to-go.
Next chapters will be coming up, also based on their work, but the authors have done their job now.

Referee #4 (Remarks to the Author):

The manuscript has been revised satisfactorily. Even the suggestions, which I gave while precising they were not compulsory (for example, analysis of gene expression changes upon exposure to high viscosity), were followed. These and the other comments from other reviewers considerably improved the manuscript. The original manuscript was already very interesting and original. I believe that the revised manuscript will become a seminal paper of the field. I highly recommend its publication in Nature and congratulate the authors for this excellent work.